EMBO
Molecular Medicine

# A fetal oncogene NUAK2 is an emerging therapeutic target in glioblastoma

Hanhee Jo [1,2,3], Sarah Munoz [1,3], Aneesh Dalvi [1], Wenqi Yang[1], Elizabeth Morozova[1] & Stacey M Glasgow [1,2] ✉

## Abstract

**Glioblastoma Multiforme (GBM) is a highly malignant brain cancer with limited effective therapies. Neurodevelopmental pathways have been implicated in glioma formation, with key neurodevelopmental regulators being re-expressed or co-opted during glioma tumorigenesis. Here we identified a serine/threonine kinase, NUAK family kinase 2 (NUAK2), as a fetal oncogene in mouse and human brains. We found robust expression of NUAK2 in the embryonic brain that decreases throughout postnatal stages and then is re-expressed in malignant gliomas. However, the role of NUAK2 in GBM tumorigenesis remains unclear. We demonstrate that CRIPSR-Cas9 mediated NUAK2 deletion in GBM cells results in suppression of proliferation, while overexpression leads to enhanced cell growth in both in vitro and in vivo models. Further investigation of the downstream biological processes dysregulated in the absence of NUAK2 reveals that NUAK2 modulates extracellular matrix (ECM) components to facilitate migratory behavior. Lastly, we determined that pharmaceutical inhibition of NUAK2 is sufficient to impede the proliferation and migration of malignant glioma cells. Our results suggest that NUAK2 is an actionable therapeutic target for GBM treatment.**

**Keywords** Extracellular Matrix; Fetal Oncogene; Glioblastoma; Neurodevelopment; NUAK2
**Subject Categories** Cancer; Neuroscience

## Introduction

Glioblastoma (GBM) is the most common and lethal brain tumor (Aldape et al, 2019; Deorah et al, 2006). These devastating tumors exhibit widespread invasion throughout the brain, are highly proliferative, and are resistant to chemotherapy and radiotherapy (Stupp et al, 2009; Van Meir et al, 2010; Xu et al, 2020b) making them exceedingly difficult to treat (Konishi et al, 2012; Louis et al, 2021; McDonald et al, 2011; Milano et al, 2010; Omuro, 2013; Weller et al, 2015). Even with the current standard of care,

including surgical resection, radiation, and chemotherapy, the prognosis for glioblastoma is dismal, with a median survival rate of 15 months (Weller et al, 2015; Verdugo et al, 2022). Therefore, identifying new and efficient molecular targets is crucial for the development of therapeutic strategies.

Neurodevelopmental signaling pathways and transcriptional cascades have been implicated in glioma tumor initiation, maintenance, and progression (Curry and Glasgow, 2021; Mehta, 2018; Sojka and Sloan, 2024). The growing literature defining roles for these developmental genes in tumorigenesis has revealed a subclass of oncogenes called fetal oncogenes, which are predominantly expressed during embryonic development and cancer, but their expression is nominal in adult tissues (Cao et al, 2023; West et al, 2018). The minimal expression of fetal oncogenes in normal tissue can be exploited to allow for more precise targeting of cancer cells with marginal off-target effects on normal cells or neurotoxicity.

Tumor progression is controlled by molecular mechanisms triggered by multiple signaling pathways, often through the activation of regulatory kinases (Nakada et al, 2020; Schlessinger, 2000). Kinase activity directly affects the activation/inactivation of downstream effectors, which are crucial for the initiation of many biological phenomena, such as cell growth, proliferation, and apoptosis (Fleuren et al, 2016). Mutations and alterations in several kinase signaling cascades have been associated with glioma tumorigenesis, leading to inhibition of apoptosis, cellular proliferation, and tissue invasion (Aiello and Stanger, 2016; Balachandran and Narendran, 2023; Ma et al, 2010; Monk and Holding, 2001). The abnormal expression or activity of kinases, which can be specific to cancerous cells, represents an attractive target for glioma therapy (Adjei, 2005; Stitzlein et al, 2024).

NUAK family kinase 2 (NUAK2), also known as sucrose non-fermenting (SNF-1)-like kinase (SNARK), is a serine/threonine kinase of the AMP-activated protein kinase family. In developing mice, NUAK2 expression is found in the neural folds, and NUAK2 knockout mice show neural tube closure defects, including exencephaly (Hirano et al, 2006; Ohmura et al, 2012). Similarly, loss-of-function mutations of NUAK2 in humans result in anencephaly, a severe form of neural tube closure failure (Bonnard et al, 2020). In both mice and humans, these neural tube defects are linked to defective regulation of cytoskeletal proteins (Bonnard et al, 2020; Ohmura et al, 2012). Roles for NUAK2 in several

[1]Neurobiology Department, School of Biological Sciences, University of California San Diego, La Jolla 92093 CA, USA. [2]Neurosciences Graduate Program, University of California San Diego, La Jolla 92093 CA, USA. [3]These authors contributed equally: Hanhee Jo, Sarah Munoz. ✉E-mail: sglasgow@ucsd.edu

non-CNS cancers have been reported, with its expression being highly correlated with tumor progression and poor prognosis in patients (Li et al, 2021; Namiki et al, 2015, 2011; Tang et al, 2017; Wang et al, 2024; Fu et al, 2022). However, there is limited knowledge of the role of NUAK2 in GBM.

In this study, we find that NUAK2 is a fetal oncogene whose expression is low in juvenile and adult brains, but high in developing brains and glioblastoma patients. In GBM cells, we show that NUAK2 deletion leads to attenuation of proliferation and migration, while overexpression enhances these processes. Modulation of NUAK2 expression in in vivo models of malignant glioma mimics these results. Importantly, pharmaceutical inhibition of NUAK2 exhibits significant effects in mitigating glioma progression. Therefore, NUAK2 is a potential actionable target for the treatment of GBM.

# Results

## NUAK2, a fetal oncogene, is associated with poor prognosis in GBM patients

NUAK2 plays a crucial role in brain development and the formation of non-CNS solid tumors, with its expression or mutations leading to various abnormalities (Bonnard et al, 2020). To investigate whether NUAK2 functions as a fetal oncogene in the CNS, we first examined publicly accessible RNA-sequencing (RNA-seq) data from the BrainSpan Atlas for the developing human brain (http://brainspan.org), which profiles up to 16 cortical and subcortical structures throughout the entire span of human brain development. Our analysis revealed that NUAK2 mRNA expression is significantly elevated during early developmental stages, declining gradually and remaining silent after birth in human brains (Fig. 1A). In contrast, RNA-sequencing data from The Cancer Genome Atlas (TCGA) indicate that NUAK2 expression is markedly elevated in GBM patients, while normal brain tissues display only minimal expression levels (Fig. 1B). Additionally, analysis of the Chinese Glioma Genome Atlas (CGGA) databases (Fig. 1C) and TCGA (Fig. 1D) revealed a correlation between NUAK2 levels and glioma tumor grade, where high-grade gliomas exhibited greater NUAK2 expression than low-grade gliomas (Fig. 1C,D); implying that NUAK2 may play a role in brain tumor malignancy.

To assess the relationship between NUAK2 expression and overall survival in human gliomas, we analyzed TCGA and CGGA datasets comparing patients with high expression of NUAK2 versus patients expressing low levels of NUAK2. Analysis of overall patient 50% survival rates revealed that elevated NUAK2 levels are strongly associated with reduced survival rates in the CGGA (Fig. 1E). We similarly observed a strong association with reduced survival in NUAK2[high] compared to NUAK2[low] patients in the TCGA cohort that represents both high- and low-grade gliomas (Fig. 1F). We further analyzed the datasets based on glioma subtypes including GBM, astrocytoma, and oligodendroglioma finding that NUAK2[high] expressing patients have poor survival in these populations (Fig. 1F). Together with our gene expression analysis, these findings suggest that NUAK2 functions as a fetal oncogene and demonstrates an explicit relationship between tumor progression and NUAK2 expression in GBM.

To further confirm our analysis, we examined Nuak2 expression in mice across different ages using publicly available datasets from EMBL's European Bioinformatics Institute of developing mouse brain transcriptomes (Cardoso-Moreira et al, 2019). Similar to humans, Nuak2 mRNA expression in mice peaks during early development and declines postnatally (Fig. 1G). Our immunoblot and qPCR analyses from embryonic and postnatal mouse tissues further confirmed this trend, with high Nuak2 levels in developing mouse brains and substantially reduced expression after birth (Fig. 1H,I). Together, these findings classify Nuak2 as a fetal oncogene in both humans and mice, and demonstrate its strong association with GBM prognosis and tumor progression.

## NUAK2 is critical for GBM cell proliferation

To understand the role of NUAK2 in GBM, we investigated the impacts of loss-of-function (LOF) and gain-of-function (GOF) studies in GBM cells. mRNA and protein expression analysis across four GBM cell lines (U87, LN229, U251, and LN319) revealed varying NUAK2 levels, with U251 and LN319 showing high expression, LN229 showing moderate levels, while U87 cells exhibited nominal NUAK2 expression (Fig. 2A,B). To determine whether NUAK2 is essential for promoting glioblastoma cell growth, we used a CRISPR-Cas9 system to silence NUAK2 in U251 cells, which express relatively higher levels of NUAK2 expression compared to the other three glioma cell lines (Fig. 2A,B). Successful NUAK2 deletions were confirmed by both immunoblot and qPCR in three independent U251-NUAK2-CRISPR clones (Fig. 2C,D). To determine the effects of NUAK2 deletion on proliferation, we used immunocytochemistry for Ki67, a known marker of proliferation, and performed an EdU incorporation assay. Anti-Ki67 staining revealed significantly reduced cell proliferation in NUAK2-CRISPR (CR) cells (Fig. 2E) and reduced incorporation of EdU (Fig. 2F). Additionally, MTT assay and colony formation assays showed reduced growth of glioblastoma cells and suppressed formation of colonies in the absence of NUAK2 (Fig. 2G,H).

To determine if NUAK2 is sufficient to drive proliferation in glioma cells, we overexpressed NUAK2 via lentiviral transduction in the two glioma cell lines with low NUAK2 endogenous expression, U87 and LN229. Stable NUAK2-overexpressing (N2OE) cell lines were generated and validated using immunoblotting and immunocytochemistry (Fig. 3A,B). MTT proliferation and colony formation assays showed that NUAK2 overexpression significantly enhanced cell proliferation (Fig. 3C,D) in both U87 and LN229 cells. To determine if NUAK2 can drive proliferation in proliferation in a glioma stem cell (GSC) context, we assessed the effects of NUAK2 overexpression in two low NUAK2-expressing GSC lines, GSC11 and GSC23 (Fig. 3E). Overexpression was confirmed by immunoblotting (Fig. 3F). To examine proliferation, we performed spheroid assays finding increased proliferation (Fig. 3G) and viability (Fig. 3H) in both GSC11 and GSC23 cells upon NUAK2 overexpression. Additionally, transwell migration assays revealed that NUAK2 overexpression enhanced cell migration of both GSC lines (Fig. 3I,J). Together, these findings indicate that NUAK2 promotes proliferation and migration in glioma cell lines and GSCs, supporting its critical role in GBM progression.

## Silencing NUAK2 impedes GBM cell growth in orthotopic xenograft models

We next evaluated the effect of NUAK2 deletion in vivo. We employed a mouse xenograft GBM model in which U251 NUAK2-WT and

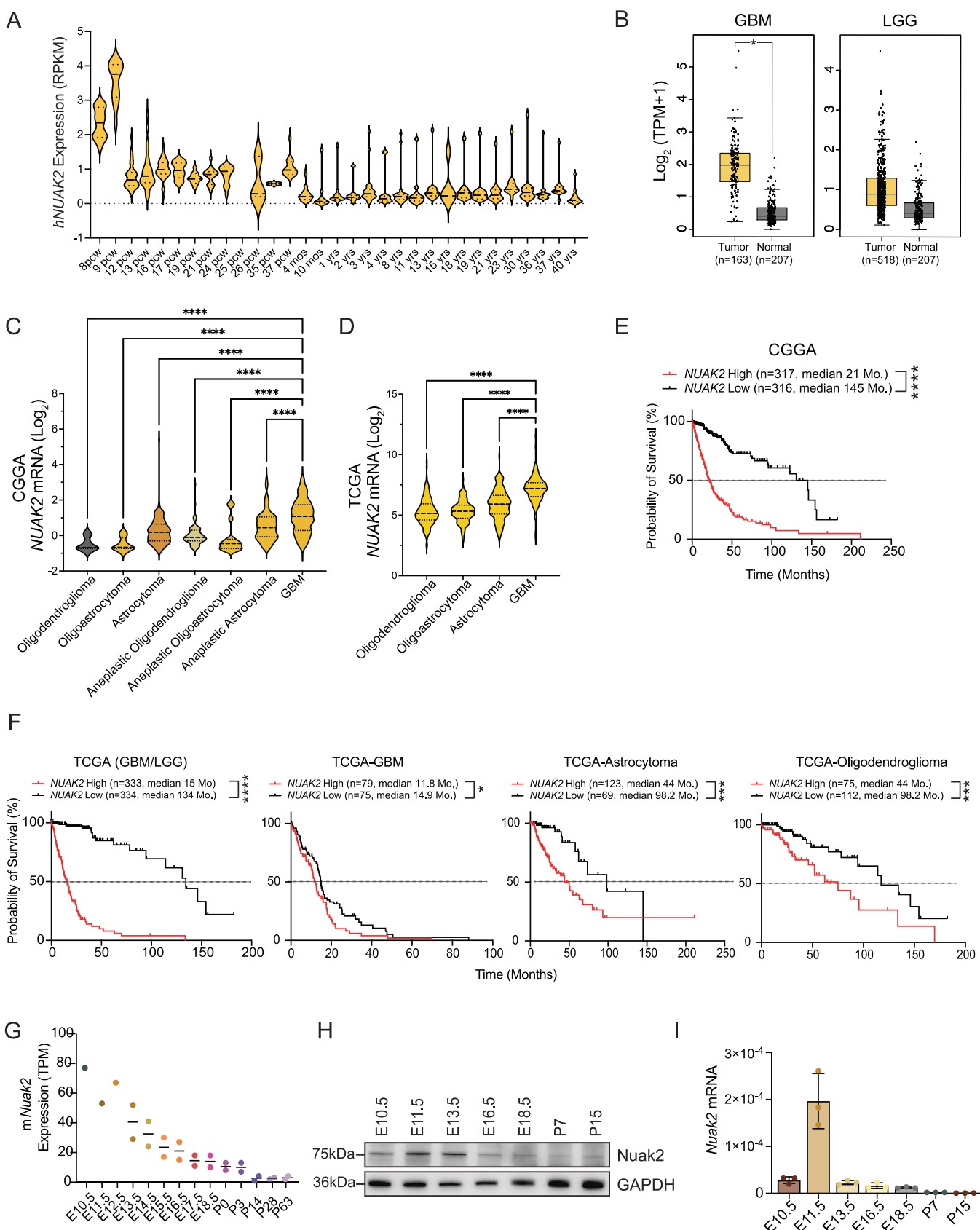

◄ **Figure 1. A fetal oncogene NUAK2 is associated with poor prognosis in GBM patients.**

(A) RPKM-normalized NUAK2 mRNA expression of various human brain regions from eight post-conception weeks (pcw) to 40 years of age. Data were obtained from the BrainSpan Atlas. $n = 5$–27 samples. (B) Normalized NUAK2 mRNA expression (log2(TPM + 1)) of TCGA GBM ($n = 163$) or low-grade glioma (LGG) ($n = 518$) and GTEx non-tumor ($n = 207$) samples ($*p = 0.01$; Statistical significance was determined by one-way ANOVA; exact $p$ value is reported in Appendix Table S3. Box plots display the median (center line), interquartile range (box: 25th to 75th percentiles), and whiskers representing the minimum and maximum values. Data were obtained from GEPIA (http://gepia.cancer-pku.cn/). (C) NUAK2 mRNA expression across glioma subtypes showing the highest expression in GBM in the CGGA dataset ($****p < 0.0001$; Statistical significance was determined by one-way ANOVA followed by Tukey's multiple comparisons test; exact $p$ value is reported in Appendix Table S3). Data were represented as mean ± SD. (D) NUAK2 mRNA expression across glioma subtypes showing the highest expression in GBM in the TCGA dataset ($****p < 0.0001$; Statistical significance was determined by one-way ANOVA followed by Tukey's multiple comparisons test; exact $p$ value is reported in Appendix Table S3). Data were represented as mean ± SD. (E) Kaplan–Meier survival analysis from CGGA of high (21 days; $n = 317$) and low (145 days; $n = 316$) NUAK2 expressers shows high NUAK2 expression is correlated with worse survival outcomes ($****p < 0.001$; Statistical significance was determined by log-rank (Mantel–Cox) test). Exact $p$ value is reported in Appendix Table S3. (F) Kaplan–Meier survival analysis from TCGA across glioma subtypes. The first plot represents high (15 months; $n = 333$) and low (134 months; $n = 334$) NUAK2 expressers from GBM and low-grade gliomas (LGG), showing high NUAK2 expression is correlated with worse survival outcomes. The second plot represents survival in high (11.8 months: $n = 79$) and low (14.9 months: $n = 75$) NUAK2 expressing GBM patients. The third plot shows NUAK2 high (44 months; $n = 123$) and low (98.2 months; $n = 69$) survival for astrocytomas. The fourth plot shows NUAK2 high (44 months; $n = 75$) and low (98.2 months; $n = 112$) survival in oligodendroglioma patients. ($****p < 0.0001$, $***p < 0.001$, $*p = 0.02$; Statistical significance was determined by log-rank (Mantel–Cox) test). Exact $p$ value is reported in Appendix Table S3. (G) TPM-normalized NUAK2 mRNA expression of mouse forebrain or hindbrain, ranging from embryonic day 10.5 to postnatal day 63. Data was obtained from EMBL's European Bioinformatics Institute (EMBL-EBI; https://www.ebi.ac.uk/). (H) Representative western blot of NUAK2 protein expression in whole wildtype (WT) embryonic brain tissue across seven stages of development. GAPDH was used as a loading control. E represents embryonic day and P represents postnatal. (I) Representative qRT-PCR of NUAK2 mRNA expression in wild-type embryonic brain tissues across developmental stages. Data were normalized to GAPDH ($n = 3$). Data were represented as mean ± SD. Source data are available online for this figure.

NUAK2-CR3 cells were intracranially injected into BALB/c nude mice. Tumor formation and growth were monitored weekly with in vivo bioluminescence imaging (IVIS) from day 7 to day 28 post-injection. The results show significantly smaller tumors in *NUAK2*-CR mice compared to controls (Fig. 4A,B). Mice injected with *NUAK2*-CR had a median survival of 89 days while mice injected with control U251 cells had no significant deaths even after 150 days (Fig. 4C). Histological analysis further confirmed that tumor sizes were markedly smaller in the *NUAK2*-CR cohort (Fig. 4D), with notably fewer Ki67 and PCNA-expressing cells (Fig. 4E). These findings suggest that *NUAK2* deletion effectively suppresses stable tumor engraftment and expansion in the context of the brain.

### Nuak2 deletion in an IUE model of malignant glioma supports a role for Nuak2 in GBM

Given the limitation that BALB/c nude mice are immunocompromised, we further examined the role of Nuak2 using a piggyBac in utero electroporation (PB-IUE) model of malignant glioma (Zhang and Bordey, 2023; Chen and LoTurco, 2012; Glasgow et al, 2014). PB-IUE tumors are generated in immunocompetent mice and more closely mimic GBM pathophysiology. Using this system, we conducted both Nuak2-GOF and -LOF studies to determine the role of Nuak2 in glioma formation.

For Nuak2 LOF (*Nuak2*-CR) studies, we used a CRISPR-Cas9 approach where dual guide RNAs targeting *Nuak2* were co-electroporated with tumor-generating PB-IUE plasmids (Fig. 5A). Western blot analysis from harvested tumors confirmed deletion of Nuak2 in electroporated tumors (Fig. 5B). Survival studies revealed that the *Nuak2*-CR cohort had significantly prolonged 50% survival rates compared to control tumor-bearing mice (Fig. 5C). Complementary, GOF studies where tumor-generating PB-IUE constructs were co-electroporated with Nuak2 expression plasmid, demonstrated that overexpression of Nuak2 (Nuak2-OE) conferred significantly reduced 50% survival rates compared to control mice (Fig. 5D). Nuak2 expression in GOF and LOF tumors was validated by immunohistochemical analysis at postnatal days P30, P50, and P70 (Figs. 5E,F and Fig EV1A,B). Proliferation in tumors was

analyzed by both Ki67 and PCNA expression levels revealing that tumors with Nuak2 LOF had fewer proliferating cells, while Nuak2 GOF led to enhanced proliferation (Figs. 5E–I and EV1A,B). In addition, we also analyzed CD-44 as a known tumor biomarker for proliferation in cancer (Xu et al, 2020a). Our results showed an increase in CD-44 positive staining in Nuak2-OE tumors compared to control tumor-bearing mice, while Nuak2-CR tumors had significantly fewer Ki67, PCNA, and CD-44 positive cells (Fig. 5E–I). Notably, Nuak2-OE tumors had large areas of necrosis as compared to control and *Nuak2*-CR tumors (Fig. 5E), consistent with Nuak2-OE leading to excessive growth of the cells and poor prognosis. These results are also consistent with survival trends observed in human glioma patients (Fig. 1). Together with our results from orthotopic U251 transplants (Fig. 4), these findings indicate that NUAK2 promotes tumorigenesis in in vivo models of high-grade glioma.

### NUAK2 regulates extracellular matrix (ECM) gene expression

To investigate the mechanistic role of NUAK2 in GBM, we performed bulk RNA-sequencing transcriptomic analysis to compare gene expression profiles of U251-NUAK2-WT and *NUAK2*-CR cell lines (Fig. EV2A). Deletion of NUAK2 resulted in 672 differentially expressed genes (DEGs) using standard cut-offs (Log$_2$ -fold change greater than two, and adjusted $p$ value 0.05) in U251-*NUAK2*-CR cells compared to control cells (Fig. EV2A), with 277 upregulated and 395 downregulated (Dataset EV1). To understand the biological significance of these gene expression changes, we performed gene ontology (GO) analysis of the biological process and cellular component groups using *NUAK2*-CR DEGs (Dataset EV2). This analysis revealed that top dysregulated genes were enriched for neurodevelopmental and extracellular matrix (ECM) genes (Fig. 6A). Next, we used the GlioVis data portal (Bowman et al, 2017) to access TCGA datasets from GBM-NUAK2[high] versus GBM-NUAK2[low] patients. Patients in the top 25% for NUAK2 (GBM-NUAK2[high]) expression were compared to those in the bottom 25% for NUAK2

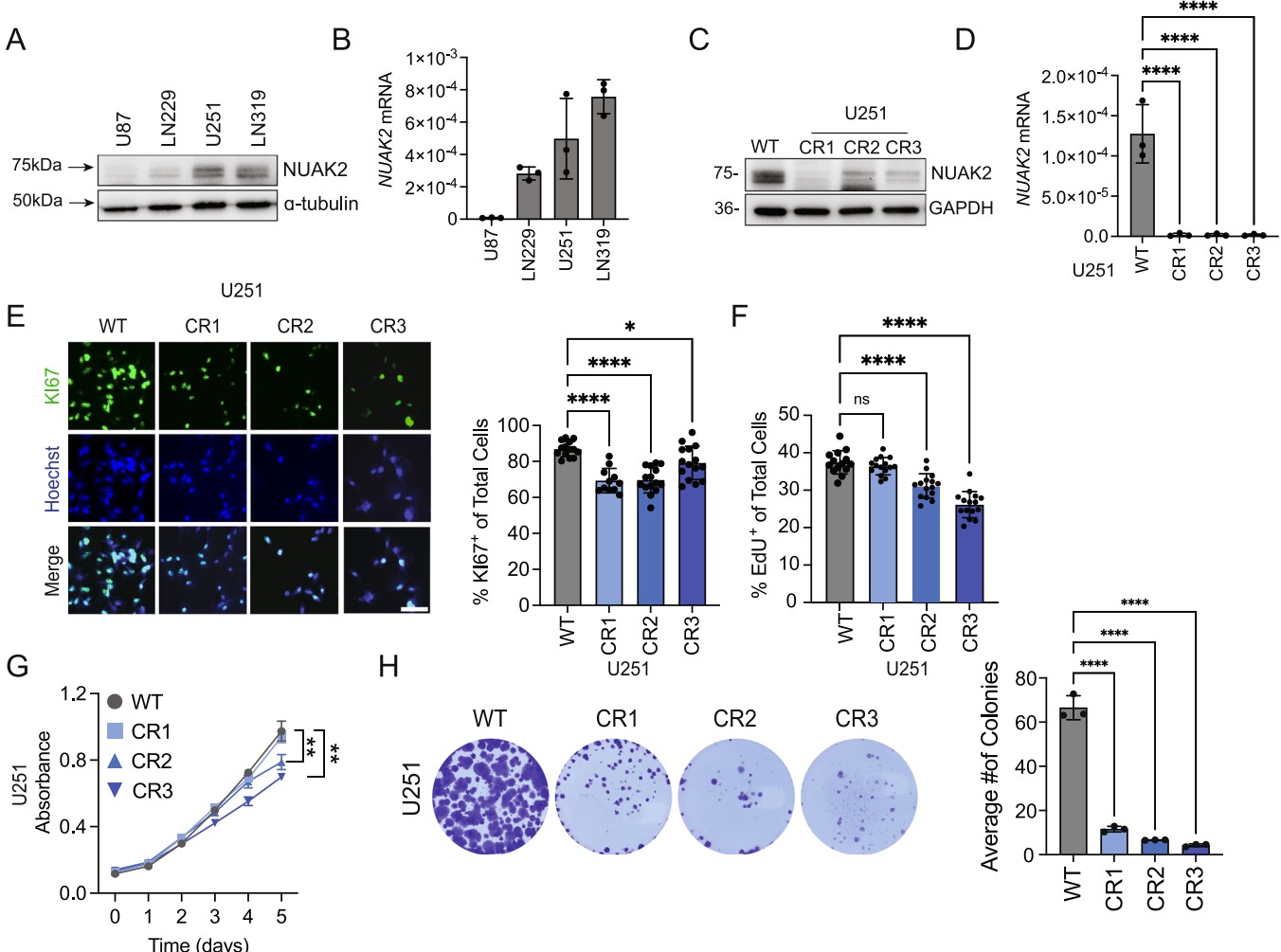

**Figure 2. NUAK2 expression is critical for GBM cell proliferation.**

(A) Representative western blot of NUAK2 protein expression in U87, LN229, U251, and LN319 glioblastoma cell lines. Alpha-tubulin was used as the loading control. Molecular weights are represented by arrows. (B) qRT-PCR analysis of the mRNA levels of NUAK2 in U87, LN229, U251, and LN319 glioma cell lines. GAPDH was used for normalization. Data were represented as mean ± SD ($n = 3$). (C) Western blot analysis of the efficiency of CRISPR-mediated deletion of NUAK2 in U251 cells. GAPDH was used as a loading control. Three independent clones are shown. Molecular weights are demarcated to the left. WT wildtype, NUAK2-CRISPR clone 1 = CR1, CRISPR2 clone = CR2, and CRISPR3 clone = CR3. (D) qRT-PCR analysis of the efficiency of CRISPR-mediated deletion of NUAK2 in U251 cells. Three independent clones are shown. Data were represented as mean ± SD (****$p < 0.0001$; Statistical significance was determined by one-way ANOVA analysis followed by Dunnett's multiple comparison test. Exact $p$ value is reported in Appendix Table S3. (E) Representative images of the proliferation marker Ki67 in NUAK2-deleted U251 cells, Hoechst was used to identify cellular nuclei. Quantification of Ki67 is shown on the right (*$p = 0.0159$, ****$p < 0.0001$; Statistical significance was determined by one-way ANOVA analysis followed by Dunnett's multiple comparison test). Exact $p$ value is reported in Appendix Table S3. Data were represented as mean ± SD. Scale bar = 50 μm. $n = 15$ from three independent coverslips. (F) Quantification of EdU incorporation in U251-control and U251-NUAK2-CR cells. Data were represented as mean ± SD (****$p < 0.0001$; Statistical significance was determined by one-way ANOVA analysis followed by Dunnett's multiple comparison test). Exact $p$ value is reported in Appendix Table S3. $n = 15$. (G) MTT assay evaluating proliferation, as indicated by absorbance, when NUAK2 was deleted in U251 cells. Statistics were evaluated at day 5. WT vs. CR2, WT vs. CR3. Data were represented as mean ± SD (**$p < 0.01$; Statistical significance was determined by two-way RM ANOVA analysis followed by uncorrected Fisher's LSD). Exact $p$ value is reported in Appendix Table S3. $n = 4$. (H) Colony formation assay on WT and NUAK2-deleted U251 cells. Quantification of the average number of colonies per well. Data were represented as mean ± SD ($n = 3$, ****$p < 0.001$; Statistical significance was determined by one-way ANOVA analysis followed by Dunnett's multiple comparison test). Exact $p$ value is reported in Appendix Table S3. Source data are available online for this figure.

(GBM-NUAK2$^{low}$) expression to obtain patient DEGs (Dataset EV1-2). We then compared the U251-CR and GBM-NUAK2 high versus low DEG datasets to define commonly dysregulated DEGs. This analysis revealed that ECM-related genes are a shared group of DEGs between U251-NUAK2-CR and GBM patients (Fig. 6B), suggesting that ECM-related terms are significantly influenced by NUAK2 expression. Gene set enrichment analysis (GSEA) using

the "extracellular matrix organization" GO term (Ashburner et al, 2000; Aleksander et al, 2023; Carbon et al, 2021) confirmed the association of NUAK2 expression with ECM components, with U251-NAUK2WT cells showing a positive enrichment in ECM signature compared to U251-NAUK2-CR (Fig. 6C; Dataset EV3). Similarly, NUAK2$^{high}$-expressing GBM patient samples demonstrated a positive ECM enrichment compared to NUAK2$^{low}$ samples

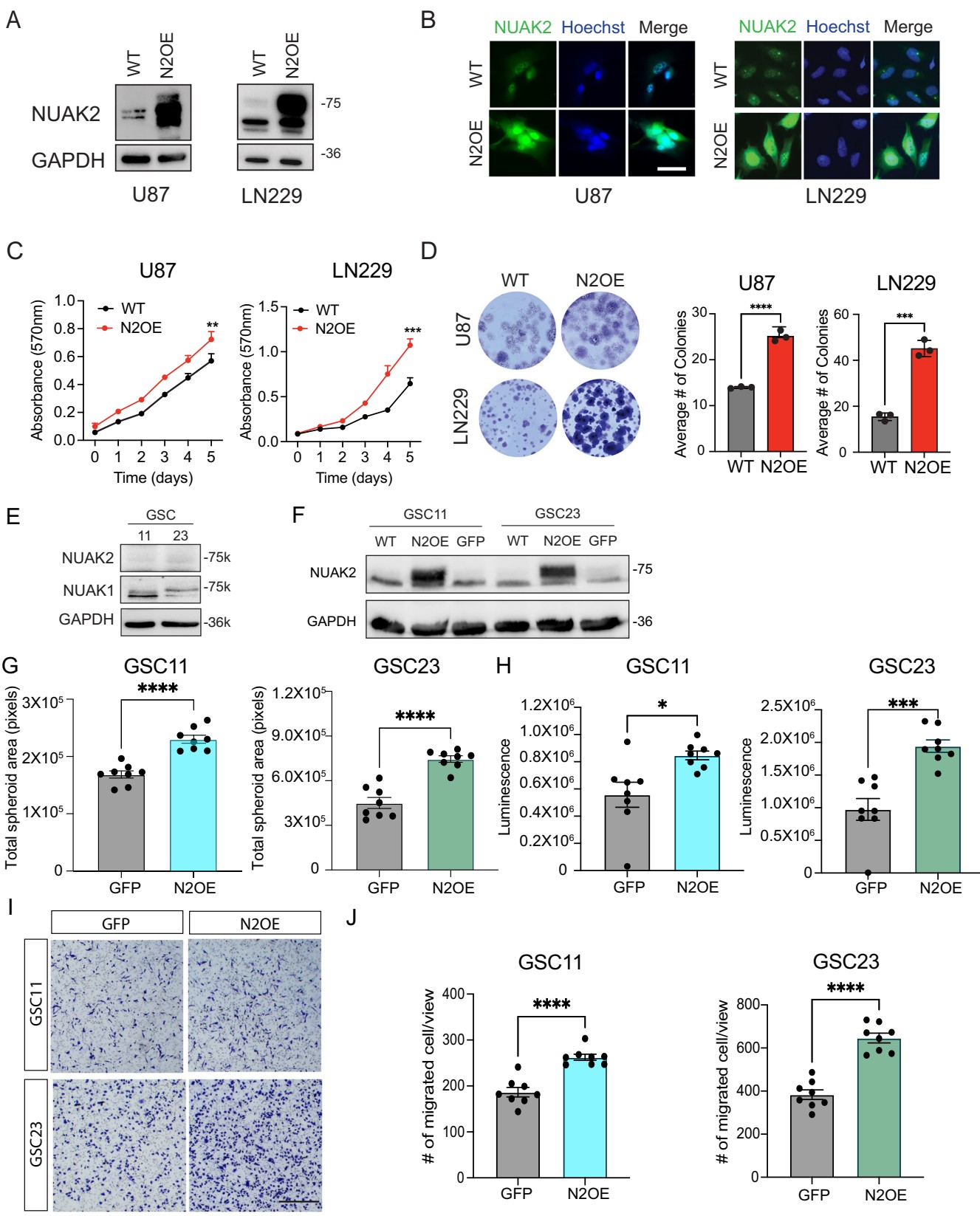

**Figure 3. NUAK2 is sufficient to drive the proliferation and migration of GBM cells.**

(A, B) Western blot and immunocytochemical validation of NUAK2 overexpression (N2OE) in U87 and LN229 WT cells. NUAK2 is in green, and Hoescht is in blue. GAPDH was used as a loading control. Scale bar = 50 µm. (C) MTT assays evaluating the effects of N2OE in U87 and LN229 cells. Data were represented as mean ± SD (**$p$ = 0.0068, ***$p$ = 0.0001; Statistical significance was determined by two-way RM ANOVA analysis followed by uncorrected Fisher's LSD. Exact $p$ value is reported in Appendix Table S3. $n$ = 4. (D) Colony formation assay evaluating the effects of N2OE in U87 and LN229 cells. Data were represented as mean ± SD (***$p$ = 0.0002, ***$p$ < 0.001; Statistical significance was determined by unpaired $t$-test (two-tailed)). Exact $p$ value is reported in Appendix Table S3. $n$ = 3. (E) Representative western blots of NUAK1 and NUAK2 expression in glioma stem cell (GSC) 11 and GSS 23. Alpha-tubulin was used as the loading control. Molecular weights are shown on the left. (F) Western blot validation of N2OE in GSC11 and GSC23. GAPDH was used as a loading control. WT un-transduced cells, N2OE, and control GFP virus are shown. (G) Quantification of total spheroid area after 6 days in culture. Data were represented as mean ± SD ($n$ = 8; ****$p$ < 0.0001; Statistical significance was determined by one-way ANOVA analysis followed by Dunnett's multiple comparison test). Exact $p$ values are reported in Appendix Table S3. (H) Cell Titer Glo 3D cell viability assay quantification of GSC11- and GSC23 control (GFP) or N2OE cells. ($n$ = 8; *$p$ = 0.0165, ***$p$ = 0.0003; Statistical significance was determined by one-way ANOVA analysis followed by Dunnett's multiple comparison test). Data were represented as mean ± SD. (I) Representative images of the transwell migration assay in GSC11- and GSC23 control (GFP) or N2OE cells. Scale bar = 500 µm. (J) Quantification of the transwell assay. Data represent the number of migrated cells per image. Eight images were analyzed per condition. Data were represented as mean ± SD. (****$p$ < 0.0001; Statistical significance was determined by one-way ANOVA analysis followed by Dunnett's multiple comparison test). Exact $p$ values are reported in Appendix Table S3. Source data are available online for this figure.

(Fig. 6D). To identify critical modulators of ECM regulated by NUAK2, we compared the ECM-related DEGs from *NUAK2*-CR U251 cells and NUAK2$^{Low}$ GBM patient samples, finding that eleven genes were consistently altered across both datasets (Fig. 6E). qPCR validated a subset of highly-expressed genes that were dysregulated in U251-NUAK2-CR, recapitulating the expression patterns observed via RNA-seq analysis (Fig. 6F), with an upregulation of genes associated with ECM degradation (MMP7 and ADAMTS20) and a downregulation of genes associated with maintenance of the extracellular matrix (FBLN1). Taken together, these data suggest that ECM-related terms are significantly influenced by NUAK2 expression in both glioma cell lines and human GBM patients.

To further understand the mechanism of NUAK2 activity in glioma cells, we performed Kyoto Encyclopedia of Genes and Genomes (KEGG) pathway enrichment analysis on the U251-NUAK2-CR RNA-seq-derived DEGs to identify significantly dysregulated biological pathways. This analysis revealed several signaling pathways that were dysregulated in the absence of NUAK2, including WNT, TGFβ1, and Hippo pathways (Fig. EV2B). Yes-associated protein (YAP) and transcriptional co-activator with PDZ-binding motif (TAZ) are pivotal downstream effectors of the Hippo signaling pathway, which have been linked not only to cytoskeletal regulation and cell migration, but also to NUAK2 regulation (Li et al, 2016; Pontes and Mendes, 2023; Yuan et al, 2018; Shah et al, 2025; Gill et al, 2018; Mason et al, 2019). GSEA analysis showed there was a positive enrichment of Hippo signaling components in U251-WT compared to U251-NAUK2-CR DEGs (Fig. EV2C).

Within the Hippo signaling pathway, NUAK2 phosphorylates LATS1 (large tumor suppressor) kinase, which normally phosphorylates and inactivates YAP and TAZ by promoting their cytoplasmic retention and eventual degradation (Gill et al, 2018; van de Vis et al, 2021; Yuan et al, 2018; Skalka et al, 2024). To determine if YAP and TAZ activity is affected by modulation of NUAK2 expression, we used Phos-tag western blots to determine the phosphorylation status of YAP and TAZ in cell lysates. CRISPR-mediated deletion of NUAK2 in U251 cells lead to the expected increase in YAP/TAZ phosphorylation (Fig. EV2D). We then tested the ability of NUAK2 to regulate subcellular localization of YAP/TAZ using western blots on LN229 NUAK2-overexpressing lysates, finding nuclear retention of YAP/TAZ in NUAK2-overexpressing cells (Fig. EV2D). Consistent with these results,

overexpression of NUAK2 attenuated phosphorylation of YAP/TAZ (Fig. EV2D). Enhanced phosphorylation of MYPT1 verified that overexpressed NUAK2 was active in these cells (Fig. EV2D). Together, these results suggest that loss of NAUK2 modulates Hippo signaling in glioma cells.

Given the critical role of the ECM in regulating glioma cell migration and invasion (Pontes and Mendes, 2023; Schönthal et al, 2023; Wei et al, 2024) and the enrichment of ECM-related DEGs in U251-NUAK2-CR (Fig. 6A), we investigated how modulation of NUAK2 expression affects glioma cell behavior. Transwell migration assays revealed a marked reduction in migration of U251-NUAK2-CR cells compared to U251-WT (Fig. 6G). In contrast, NUAK2 overexpression in U87 and LN229 cells significantly increased migration, suggesting that NUAK2 promotes cell migration, potentially through modulating the ECM.

## The NUAK2 inhibitor HTH-02-006 attenuates GBM cell proliferation

After identifying a critical role of NUAK2 in GBM proliferation and migration, we investigated whether inhibition of NUAK2 kinase activity could mimic the effects of *NUAK2* gene depletion in GBM cells. We utilized a commercially available NUAK2 inhibitor, HTH-02-006 (Fu et al, 2022; Yuan et al, 2018), across four GBM cell lines. Since HTH-02-006 is a semi-specific NUAK2 inhibitor that could potentially inhibit NUAK1, its homolog, we investigated the potential relevance of NUAK1 to GBM. Our analysis revealed that *NUAK1* expression is relatively stable throughout brain development in both mice and humans (Fig. EV3A–D), distinct from the tightly controlled regulation of *NUAK2*. This suggests that *NUAK1* expression is not developmentally dynamic across stages in the same manner as NUAK2. Additionally, *NUAK1* mRNA levels were significantly lower in both low- and high-grade gliomas than in normal brain tissues (Fig. EV3E). Further examination of glioblastoma subtypes showed no significant differences in *NUAK1* expression across subtypes (Fig. EV3F,G), indicating a lack of strong correlation with tumor grade. Survival analyses from Kaplan–Meier curves also showed no significant relationship between *NUAK1* expression and glioma patient prognosis in either the CGGA and TCGA datasets (Fig. EV3H,I). Furthermore, Spearman's correlation coefficient analysis on TCGA high-grade gliomas revealed that there is no significant correlation between the expression of NUAK1 and NUAK2 in tumors (Fig. EV3J). These

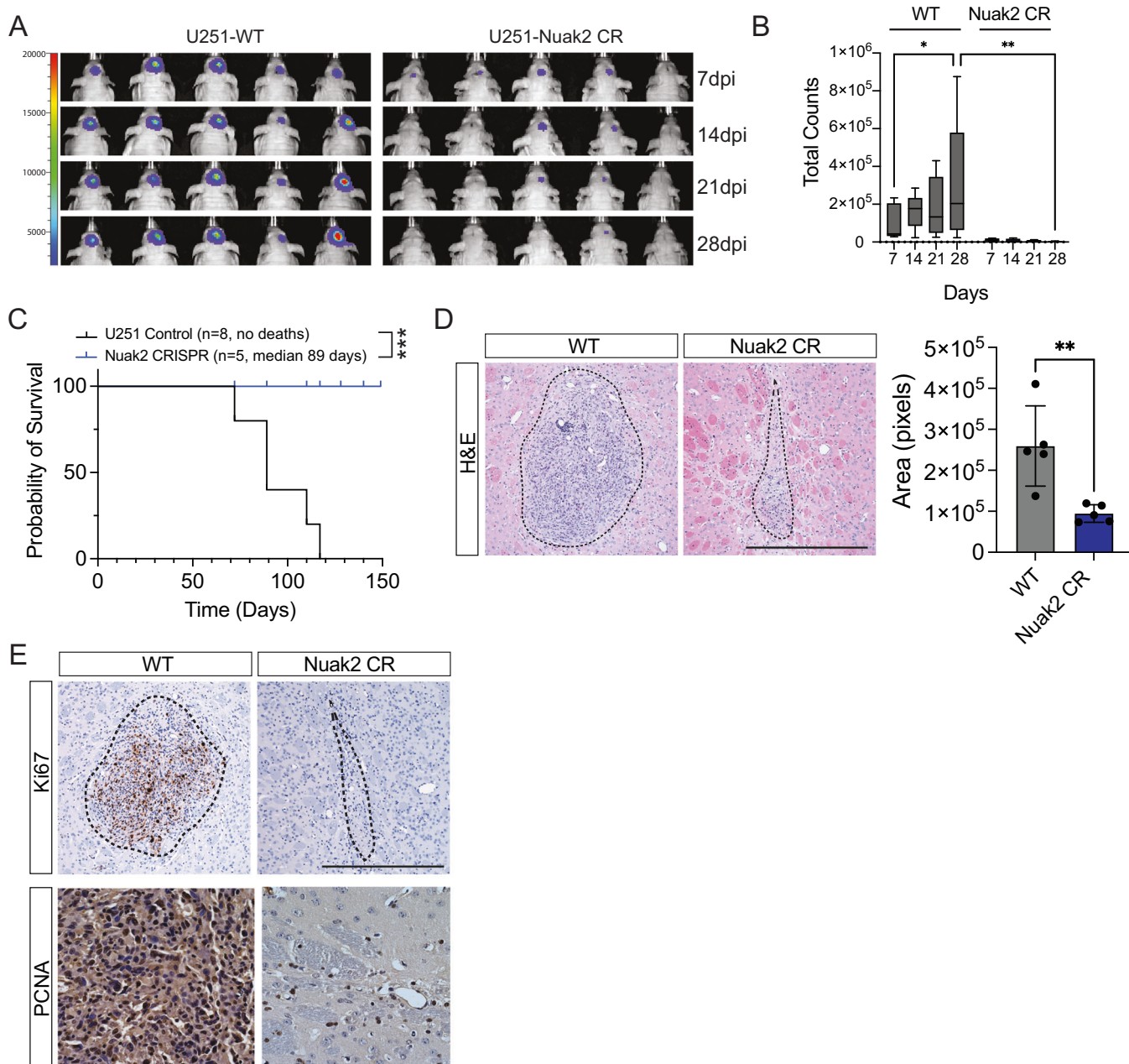

**Figure 4. NUAK2 deletion inhibited tumor growth in in vivo orthotopic xenografts.**

(A) Intracranial orthotopic xenograft in BALB/c nude Mice using U251 cells with NUAK2 deletion (U251-NUAK2-CR) or wild-type (WT) controls ($n = 5$ each). Representative bioluminescent images of tumors at 7, 14, 21, and 28 days post-injection are shown. Fluorescence signal intensity is indicated on the left. NUAK2-CR clone #3 was used in these assays. (B) Quantification of bioluminescent images obtained on the IVIS spectrum imager. Box plots show the median (center line), 25th to 75th percentiles (bounds of the box), and minimum to maximum values (whiskers). ($n = 5$, *$p = 0.036$, **$p = 0.0044$; Statistical significance was determined by two-way RM ANOVA analysis followed by uncorrected Fisher's LSD). (C) Kaplan–Meier survival analysis from NUAK2 loss-of-function (U251-NUAK2-CR) ($n = 5$) and control tumor-bearing mice ($n = 8$). (***$p = 0.0007$; Statistical significance was determined by log-rank (Mantel–Cox) test). (D) Representative images of the end-stage tumor (28 dpi) showing H&E staining. Quantification of the area of the tumor mass is shown. Data were represented as mean ± SD ($n = 5$; **$p = 0.0062$; Statistical significance was determined by an unpaired $t$-test (two-tailed)). Scale bar = 500 μm. (E) Representative images of the end-stage tumor (28 dpi) showing Ki67-positive proliferating cells in the top panels. Scale bar = 500 μm. Bottom panels are images of PCNA+ proliferating cells. Scale bar = 100 μm. Four brains were analyzed per genotype. Source data are available online for this figure.

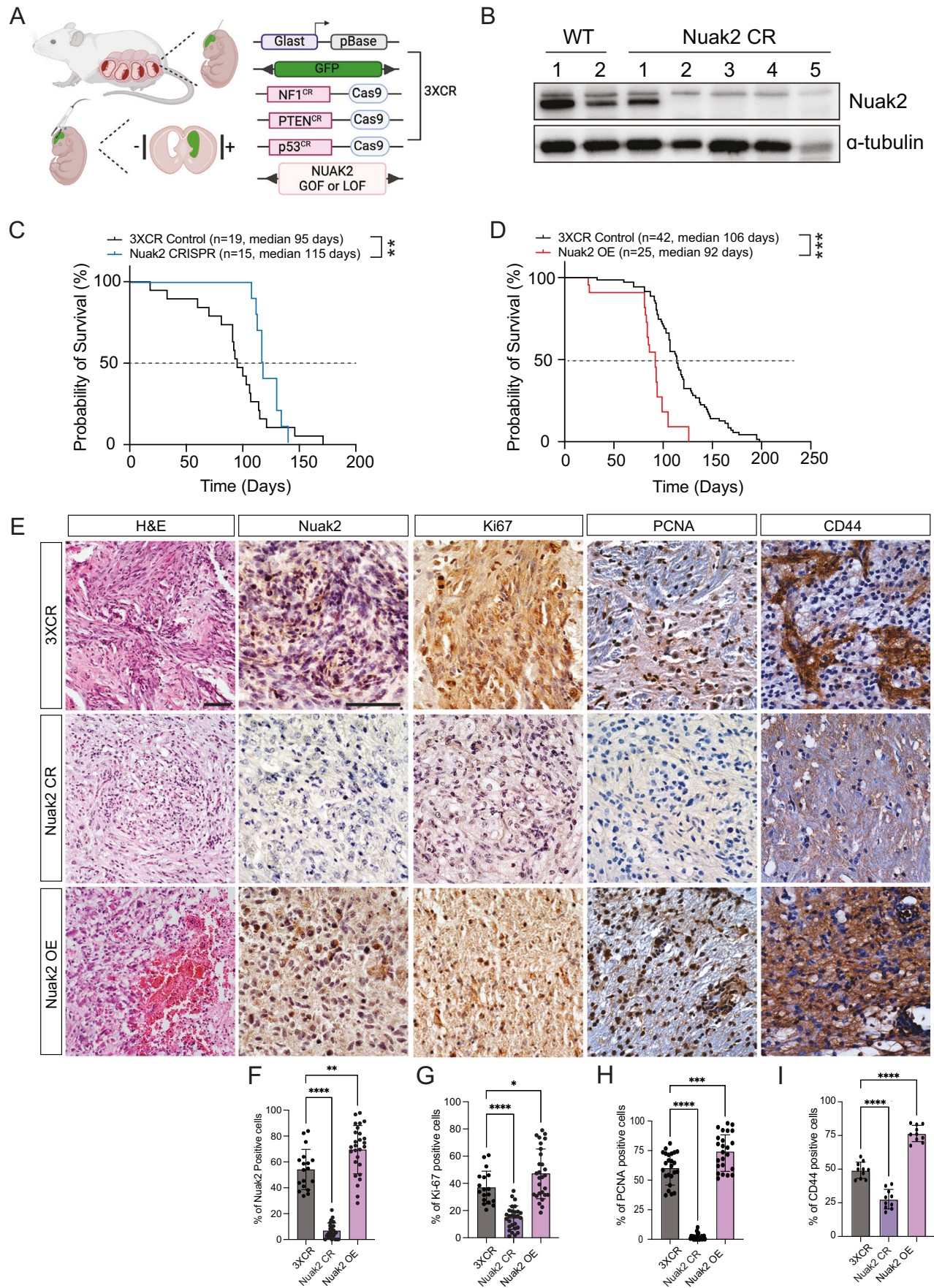

◄ **Figure 5. Modulation of NUAK2 expression in an immunocompetent model of malignant glioma affects tumor growth.**

(A) Cartoon representation of the in utero electroporation (IUE) model of malignant glioma. (B) Representative western blot of the efficiency of CRISPR-mediated deletion of NUAK2 in IUE-generated malignant gliomas. Alpha-tubulin was used as a loading control. (C) Kaplan–Meier survival analysis from NUAK2 loss-of-function (CRISPR) ($n = 15$) and control tumor-bearing mice ($n = 19$; $**p = 0.001$; Statistical significance was determined by log-rank (Mantel–Cox) test). (D) Kaplan–Meier survival analysis from NUAK2 gain-of-function (OE) ($n = 25$) and control tumor-bearing mice ($n = 42$; $***p = 0.004$; Statistical significance was determined by log-rank (Mantel–Cox) test). (E) Representative images of H&E, NUAK2 expressing, and Ki67-positive proliferating cells in control, NUAK2 deleted or OE tumors at P50. Three images from three independent brains were used for quantification. Scale bar = 100 μm. (F) Quantification of NUAK2 expression in control, NUAK2-deleted or OE tumors. Data were represented as mean ± SD ($**p = 0.0015$, $****p = 1.0E-14$; Statistical significance was determined by one-way ANOVA analysis followed by Dunnett's multiple comparison test). Three images from three independent brains were used for quantification. (G) Quantification of Ki67-positive proliferating cells in control, NUAK2-deleted or OE tumors. Data were represented as mean ± SD ($*p = 0.0282$, $****p = 2.4E-06$; Statistical significance was determined by one-way ANOVA analysis followed by Dunnett's multiple comparison test). Three images from three independent brains were used for quantification. (H) Quantification of PCNA-positive proliferating cells in control, NUAK2-deleted or OE tumors. Data were represented as mean ± SD ($***p = 0.0002$, $****p = 1.0E-14$; Statistical significance was determined by one-way ANOVA analysis followed by Dunnett's multiple comparison test). Three images from three independent brains were used for quantification. (I) Quantification of CD-44-positive proliferating cells in control, NUAK2-deleted or OE tumors. Data were represented as mean ± SD ($****p = 1.1E-07$; Statistical significance was determined by one-way ANOVA analysis followed by Dunnett's multiple comparison test). Three images from three independent brains were used for quantification. Source data are available online for this figure.

findings collectively suggest that NUAK2 may have a more prominent role than NUAK1 in glioblastoma tumorigenesis and progression.

Next, we examined the expression of NUAK1 in the four glioma cell lines used in these studies. NUAK1 is nominal in U87, LN319, and LN229, while expression is higher in U251 cells (Fig. EV3K). Immunoblots measuring NUAK1 expression in U87 and LN229 cells overexpressing NUAK2, revealed a modest decrease in NUAK1 expression, although we found endogenous NUAK1 expression in these cells is low (Fig. EV3K). We also evaluated expression of NUAK1 in U251 NUAK2-CR deleted cells and found that there was no significant difference in expression in NUAK2-CR3 cells used for further analysis (Fig. EV3K). Together, the data suggest that modulation of NUAK2 expression does not affect NUAK1, which is consistent with previous reports in other cancers (Yuan et al, 2018; Gill et al, 2018; Fu et al, 2022).

We further assessed the selectivity of the inhibitor across four glioma cell lines, first confirming the inhibitor targets NUAK2 kinase activity in these cells. This was accomplished by measuring phosphorylation of MYPT1—a known target of NUAK2 kinase (Yamamoto 2008)—using phos-tag western blot in U251 cells. We found that HTH-02-006 induced phosphorylation of MYTP1 in U251 cells (Fig. EV4A), verifying that the inhibitor blocks NAUK2 kinase activity in these cells. Additionally, evaluation of NUAK1 and NUAK2 levels in HTH-02-006-treated cells compared to vehicle-treated cells showed no change in NUAK2 or NUAK1 (Fig. EV4B). Second, we treated U251WT and U251-NUAK2-CR#1-3 with HTH-02-006 and measured spheroid formation in serum-free media and found that U251-WT cells responded to treatment in a dose-dependent manner, while U251-NUAK2-CR-treated cells did not (Fig. EV4C). Taken together, these results indicate that the inhibitor does target NUAK2 activity and likely has high specificity to NUAK2 at the tested doses, consistent with previous reports of HTH-02-006 action (Fu et al, 2022; Gill et al, 2018; Yuan et al, 2018).

To determine the effect of HTH-02-006 on GBM cell proliferation, we performed MTT and colony formation proliferation assays. We observed that the inhibitor suppressed cell proliferation in a dose-dependent manner across all four cell lines (Figs. 7A,B and EV4D,E). However, sensitivity to the drug varied depending on the level of NUAK2 expression in the cell line (Figs. 7A,B and EV4A–E). Notably, U87 cells showed a limited

response to the inhibitor, likely due to their lower NUAK2 expression levels. This finding suggests that the growth and propagation of U87 cells may be less dependent on NUAK2 activity. Additionally, scratch assay, which tests for the migration of cells across a specified region, showed that the inhibitor markedly hindered the migration of GBM cells (Figs. 7C and EV4F). To confirm that HTH-02-006 attenuates migration in the scratch assay rather than simply being an effect of decreased proliferation, we performed transwell assays on HTH-02-006 and vehicle-treated cells. Inhibitor-treated cells showed a marked decrease in migration (Figs. 7D and EV4G), suggesting that the observed result in the scratch assay is due in large part due to attenuated migration.

While our analysis using two-dimensional (2D) monolayer cultures provides initial insights into the inhibitor efficacy, this method does not replicate the architecture of tumor masses in vivo. To address this issue, we employed three-dimensional (3D) spheroid analysis to evaluate the effects of HTH-02-006 on serum-free grown glioblastoma cells. The 3D spheroid model is particularly advantageous when drug kinetics are not well understood in organisms, as it incorporates in vivo-like features such as cell-cell interactions, drug penetration, and ECM deposition (Barbosa et al, 2021; Zanoni et al, 2016). HTH-02-006 treatment demonstrated dose-dependent growth inhibition and reduced viability in all four GBM spheroid models (Figs. 7E–G and EV5A–C) underscoring its clinical significance. Altogether, these findings support the conclusion that NUAK2 is a promising therapeutic target for GBM.

## Discussion

A growing number of studies indicate that cancer cells capitalize on embryonic developmental paradigms to promote their development and progression. The parallels between development and cancer most commonly relate to stemness, EMT, and proliferation, which give the cell a selective growth advantage (Cao et al, 2023; Sharma et al, 2022). While these processes are tightly regulated during development, cancers repurpose them to enable unchecked proliferation and invasion (Balachandran and Narendran, 2023; Aiello and Stanger, 2016; Ma et al, 2010). Therapies targeting abnormal developmental pathways in cancer have been explored, but their success hinges on identifying actionable, tumor-specific

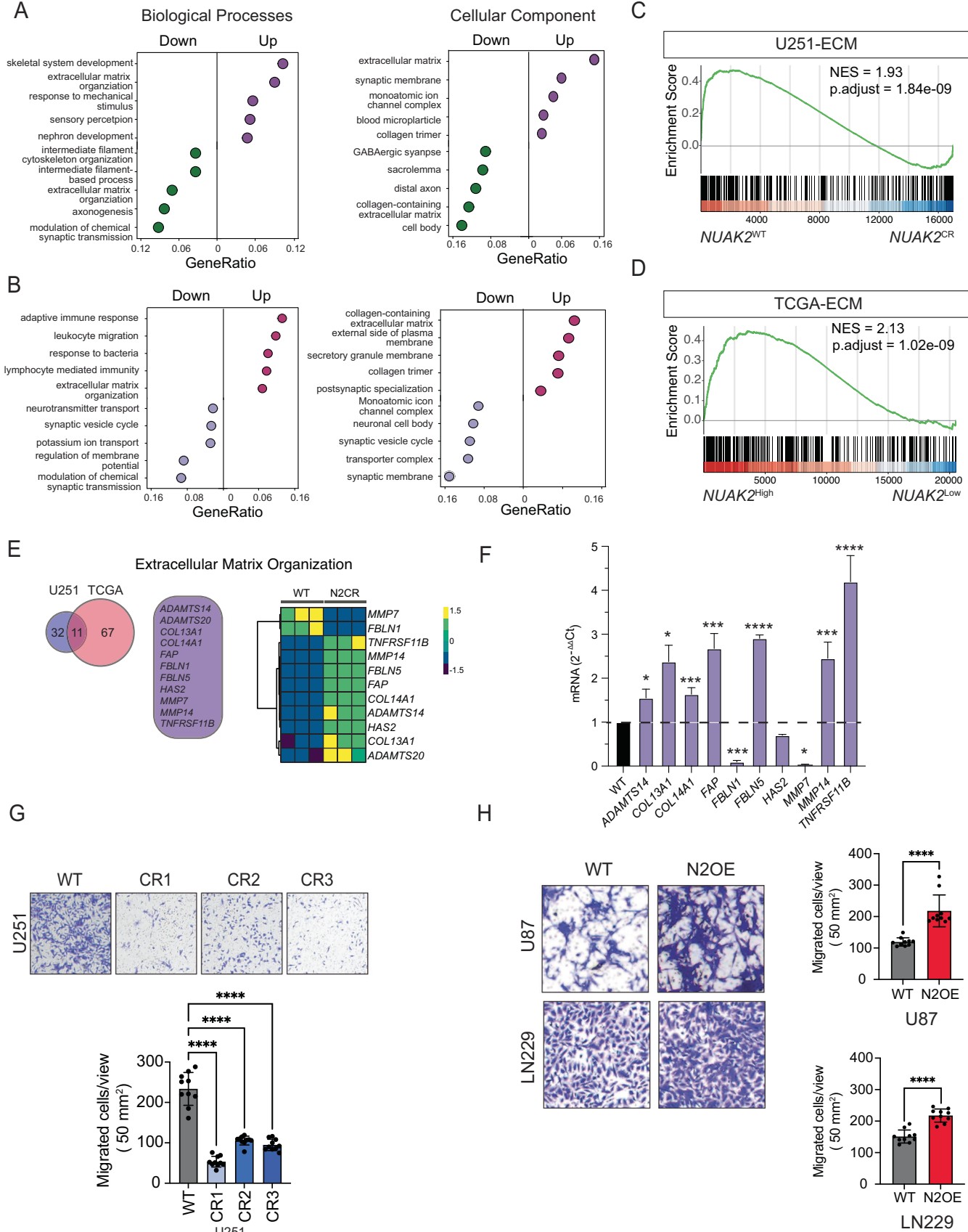

**Figure 6. NUAK2 regulates extracellular matrix (ECM) gene expression.**

(A) Gene ontology (GO) term enrichment analysis of U251 DEGs after NUAK2-CRISPR (NUAK2-CR) mediated deletion. Plots represent DEG categories by Biological process (BP) and cellular component (CC). GO analysis was performed separately on up- and downregulated genes. (B) Gene ontology (GO) term enrichment analysis of TCGA DEGs of NUAK2$^{Low}$ Group compared to NUAK2$^{High}$. Plots represent DEG categories by Biological Process (BP) and Cellular Component (CC). GO analysis was performed separately on up- and downregulated genes. (C) Gene set enrichment analysis (GSEA) of U251WT and U251CR showing enrichment in extracellular matrix (ECM) gene sets. (D) GSEA of TCGA GBM NUAK2-high and NUAK2-low DEGs showing enrichment in extracellular matrix (ECM) gene sets. (E) Venn diagram depicting the correlation between NUAK2-deleted U251 cells and TCGA GBM ECM-associated DEGs. Heatmap of 11 shared genes between U251 NUAK2-deleted and TCGA-GBM ECM DEGs. (F) qRT-PCR of the 11 shared ECM genes from control and U251 NUAK2-deleted cells. Data were represented as mean ± SD ($n = 3$, *$p < 0.01$, ***$p < 0.001$, ****$p < 0.0001$; Statistical significance is determined by unpaired $t$-test (two-tailed). Exact $p$ values are reported in Appendix Table S3. (G) Representative images of the transwell migration assay post NUAK2 deletion in U251 cells. Three independent CRISPR clones are shown. Quantification is displayed to the right as mean ± SD ($n = 10$, ****$p < 0.0001$; Statistical significance was determined by one-way ANOVA analysis followed by Dunnett's multiple comparison test). Exact $p$ values are reported in Appendix Table S3. (H) Representative images and quantification of the transwell migration assay after NUAK2 overexpression in U87 and LN229 cells. Data were represented as mean ± SD ($n = 10$; ****$p < 0.0001$; Statistical significance is determined by unpaired $t$-test (two-tailed)). Exact $p$ values are reported in Appendix Table S3. Source data are available online for this figure.

factors (Dempke et al, 2017; Kiesslich et al, 2012). Therefore, investigating the factors that intersect embryogenesis and tumorigenesis is critical for understanding tumor biology and developing more effective therapeutic strategies. In this study, we identify NUAK2 as a fetal oncogene essential for CNS development and demonstrate its key role in GBM growth and progression. We show that NUAK2 regulates ECM genes likely through activation of the Hippo signaling pathway via YAP and TAZ. Importantly, pharmaceutical inhibition of NUAK2 attenuates GBM cell proliferation and migration, highlighting its therapeutic potential.

Transcriptomic profiling of U251-NUAK2-CR cells revealed enrichment of ECM-associated genes and dysregulation of some EMT-related markers, suggesting NUAK2 contributes to proliferative and migratory phenotypes. Similar dual roles for NUAK2 have been reported in other cancers. For example, in cervical cancer, NUAK2 knockdown reduced cell proliferation, migration, and expression of EMT markers via interaction with CYFIP2 (Li et al, 2021). In hepatocellular carcinoma, NUAK2 silencing or pharmacological inhibition impaired proliferation through modulation of YAP signaling (Yuan et al, 2018), and YAP-driven NUAK2 was shown to reinforce YAP activity via actomyosin cytoskeleton remodeling (Lamouille et al, 2014).

This dual regulation of proliferation and mesenchymal traits is not uncommon in cancer and reflects context-dependent activation of intersecting pathways. Pathways such as TGF-β, WNT, EGFR, and Hippo can simultaneously regulate cell cycle progression and EMT by engaging distinct downstream effectors (e.g., Snail, ZEB1, and Twist). NUAK2 has been implicated in both Hippo and TGF-β signaling (Wang et al, 2024; van de Vis et al, 2021; Kolliopoulos et al, 2019), yet the crosstalk between these pathways and how they intersect with NUAK2 kinase activity remains poorly defined. Currently, known substrates of NUAK2 kinase include MYPT1 and LATS1/2 (Yamamoto et al, 2008; Banerjee et al, 2014; Gill et al, 2018), but additional targets likely exist that could mediate NUAK2's effects on both proliferation and migration. Defining these targets will be key to understanding NUAK2's full oncogenic potential in glioma and other cancers.

In glioblastoma, the ECM is a major regulator of tumor progression, influencing cellular invasion, proliferation, and survival (Brassart-Pasco et al, 2020; Dzobo and Dandara, 2023; Huang et al, 2021; Venning et al, 2015; Zhao et al, 2021; Gkretsi and Stylianopoulos, 2018). Changes in the structure and composition of the ECM can influence tumor behavior by modifying mechanical properties like stiffness, which in turn can trigger signaling cascades that enhance

malignancy (Larsen et al, 2006; Kai et al, 2019). These mechanical cues activate pathways such as Hippo-YAP/TAZ, which are sensitive to matrix stiffness and regulate transcription of genes involved in proliferation and ECM remodeling (Masliantsev et al, 2021; Pontes and Mendes, 2023; Li et al, 2022). Inactivation of the Hippo pathway can lead to increased YAP/TAZ activity, promoting ECM remodeling and potentially increasing stiffness; a recent study suggests that reversing this process by promoting Hippo pathway activity and inactivating YAP/TAZ can lead to a softer ECM, which may improve drug delivery and reduce therapeutic resistance (Feng et al, 2024).

Our data show that Nuak2 modulates YAP/TAZ activity: NUAK2 loss increases YAP/TAZ phosphorylation (inactivation), while its overexpression promotes nuclear localization (activation). NUAK2 interacts with and phosphorylates LATS1/2, thereby inhibiting their activity and preventing YAP/TAZ phosphorylation (Gill et al, 2018). Consistently, our RNA-seq analysis showed broad Hippo pathway dysregulation in NUAK2-deficient cells. These findings position NUAK2 as a key mediator of the ECM–Hippo–YAP/TAZ axis in GBM, offering potential avenues for therapeutic intervention through ECM modulation and targeted pathway inhibition.

NUAK2 and NUAK1 share roughly 60% sequence identity (Skalka et al, 2024; Molina et al, 2021), and have shared functions in cytoskeletal remodeling, metabolic adaption, proliferation, and EMT—key hallmarks of cancer (Skalka et al, 2024; van de Vis et al, 2021; Namiki et al, 2011). However, these two kinases have distinct roles in cancer depending on context, with distinct molecular mechanisms and pathway involvement. For instance, in TGF-β signaling, NUAK1 serves as a negative feedback regulator, dampening SMAD3 activity, EMT, and contractility. Conversely, NUAK2 acts as a feed-forward amplifier, stabilizing SMAD3 and promoting EMT and matrix remodeling (Kolliopoulos et al, 2019). Similar divergence is observed in Hippo signaling regulation. NUAK2 acts as a potent YAP/TAZ activator, inhibiting LATS1/2 kinase activity, leading to nuclear accumulation of YAP and ultimately tumorigenesis in liver and breast cancers (Yuan et al, 2018; Gill et al, 2018). In contrast, NUAK1 did not show the same requirement for YAP-driven proliferation (Gill et al, 2018; Yuan et al, 2018).

Although we centered our analysis on NUAK2, NUAK1—its closely related kinase—has also been implicated in glioma progression, suggesting that both kinases may play complementary or distinct roles in disease pathogenesis. To examine their relationship more closely, we assessed their expression patterns across development and found that NUAK1 did not share the same dynamic expression pattern across development as NUAK2, nor

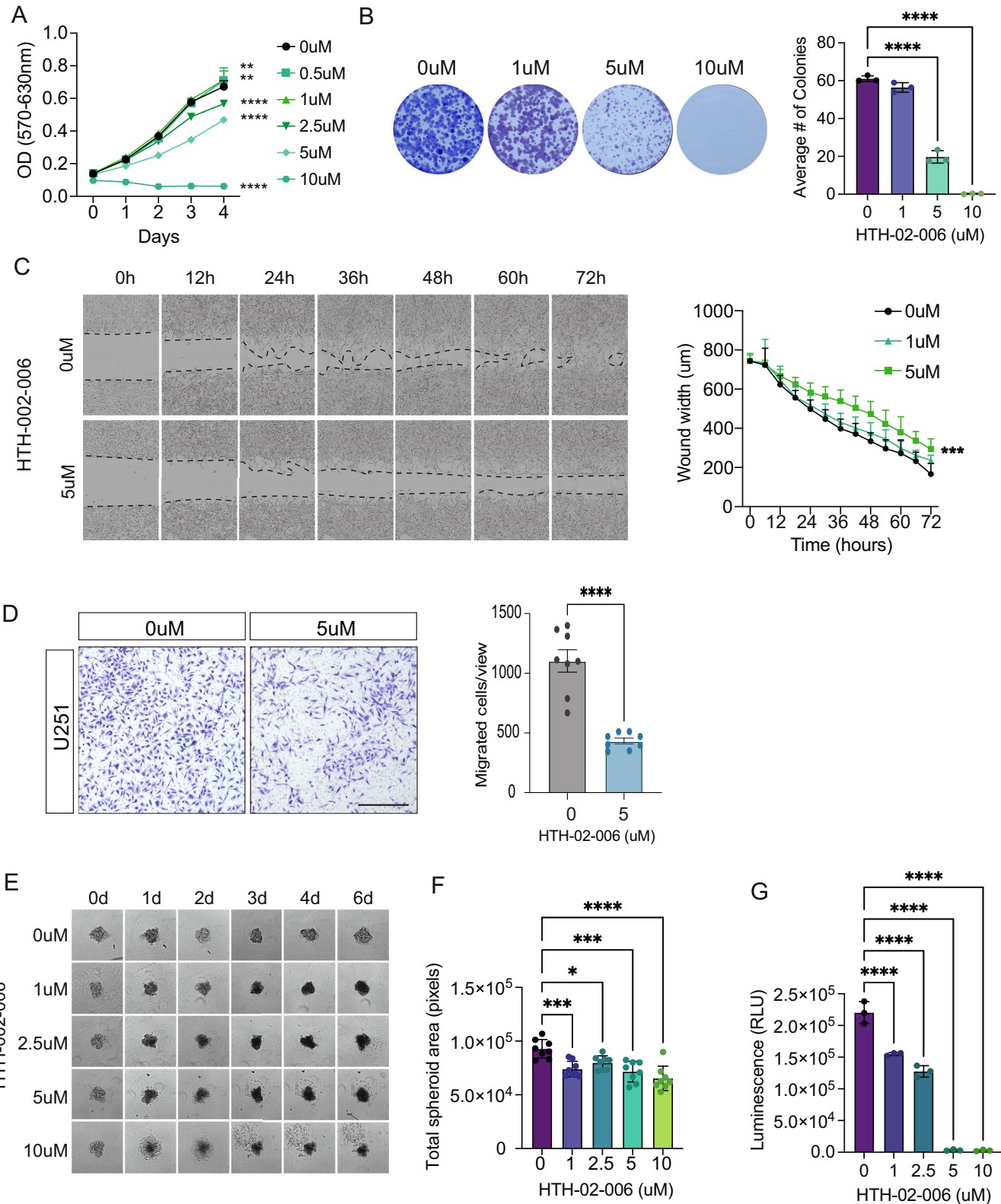

**Figure 7. Pharmaceutical inhibition of NUAK2 suppresses GBM cell progression and expansion.**

(A) MTT assay for proliferation in HTH-02-006-treated U251 cells. Data were represented as mean ± SD ($n = 7$; **$p < 0.001$, ****$p < 0.0001$; Statistical significance was determined by two-way RM ANOVA analysis followed by uncorrected Fisher's LSD. Exact $p$ values are reported in Appendix Table S3. (B) Colony formation assay and quantification on HTH-02-006-treated U251 cells. Data were represented as mean ± SD ($n = 3$; ****$p < 0.0001$; Statistical significance was determined by one-way ANOVA analysis followed by Dunnett's multiple comparison test). (C) Representative phase images and quantification of HTH-02-006-treated U251 cell migration into the wound area. Data were represented as mean ± SD ($n = 7$; ***$p = 0.0008$; Statistical significance was determined by two-way RM ANOVA analysis followed by uncorrected Fisher's LSD). Black dotted lines demarcate the wound boundary. (D) Representative images and quantification of the transwell migration assay after HTH-02-006 treatment of U251 cells. Data were represented as mean ± SD ($n = 8$; ****$p < 0.0001$; Statistical significance is determined by unpaired $t$-test (two-tailed)). Scale bar = 500 µm. (E, F) Representative brightfield images and quantification of total spheroid area of HTH-02-006-treated U251 spheroids over the course of 6 days. Data were represented as mean ± SD ($n = 8$; *$p < 0.05$, ***$p < 0.001$, ****$p < 0.0001$; Statistical significance was determined by one-way ANOVA analysis followed by Dunnett's multiple comparison test. Exact $p$ values are reported in Appendix Table S3. (G) Cell Titer Glo 3D cell viability assay quantification in HTH-02-06-treated U251 spheroids at day 6. Data were represented as mean ± SD ($n = 3$; ****$p < 0.0001$; Statistical significance was determined by one-way ANOVA analysis followed by Tukey's multiple comparison test). Exact $p$ values are reported in Appendix Table S3.

did its expression correlate with glioma tumor grade or patient survival. This is in contrast to a previous finding that NUAK1 is correlated to patient survival and had a role in promoting GBM growth in vitro (Lu et al, 2013). The discrepancies between these findings and our observations likely arise from differences in the study populations, which differ in demographics, size, and environmental and/or clinical contexts. Nonetheless, our findings that there is no correlation between NUAK1 and survival do not preclude a role for NUAK1 in glioma progression. It is interesting that our in-silico analysis revealed that NUAK2 expression was increased in GBM patient samples, while NUAK1 expression was decreased compared to control brains. These differences may be observed because NUAK2 expression is reactivated in the tumor, while NUAK1 expression is more stable across the lifetime of an individual. Moreover, we and others found no evidence that upregulation of NUAK2 directly suppresses NUAK1 expression (Yuan et al, 2018; Gill et al, 2018; Fu et al, 2022). We observed no significant changes in NUAK1 expression when NUAK2 was either depleted or overexpressed. This finding suggests that NUAK1/2 expression are independently controlled and indicates that targeting NUAK2 may offer greater tumor specificity without affecting NUAK1-related functions.

Finally, our data demonstrate that pharmacological inhibition of NUAK2 using HTH-02-006 suppresses growth across four glioma cell lines in both attached and suspension conditions. HTH-02-006 demonstrates semi-selective inhibition with favorable specificity toward NUAK2 (Yuan et al, 2018), although additional refinement is needed to minimize off-target effects. Recently developed analogs designed to enhance specificity have been reported (Hyeong Lee et al, 2024), but their efficacy and specificity have not been thoroughly validated. Our study is the first, to our knowledge, to evaluate both genetic and pharmacologic modulation of NUAK2 in GBM, and it provides preclinical rationale for NUAK2 as a candidate therapeutic target.

# Methods

**Reagents and tools table**

| Reagent/resource | Reference or source | Identifier or catalog number |
|---|---|---|
| **Experimental models** | | |
| CD-1 IGS | Charles River | Cat #022 |
| BALB/c Nude | UCSD in-house breeding | n/a |

| Reagent/resource | Reference or source | Identifier or catalog number |
|---|---|---|
| **Recombinant DNA** | | |
| pBCAG-GFP | Addgene | #40973 |
| pGLAST-PBase | Chen F, LoTurco J. A method for stable transgenesis of radial glia lineage in rat neocortex by piggyBac mediated transposition. J Neurosci Methods. 2012 Jun 15;207(2):172-80. https://doi.org/10.1016/j.jneumeth.2012.03.016. Epub 2012 Apr 11. PMID: 22521325; PMCID: PMC3972033. | |
| pbCAG-Luciferase | Chen F, LoTurco J. A method for stable transgenesis of radial glia lineage in rat neocortex by piggyBac mediated transposition. J Neurosci Methods. 2012 Jun 15;207(2):172-80. https://doi.org/10.1016/j.jneumeth.2012.03.016. Epub 2012 Apr 11. PMID: 22521325; PMCID: PMC3972033. | |
| PX330 crNF1, crPTEN, and crp53 | Yu, K. et al PIK3CA variants selectively initiate brain hyperactivity during gliomagenesis. *Nature* **578**, 166–171 (2020). | |
| Nuak2 pENTR223 | DNASU | HsCD00514317 |
| pBCAG-3xHA-gateway | This study | |
| PX330 | Addgene | #42230 |
| pBCAG-Nuak2 | This study | |
| PX330-Nuak2-CR | This study | |
| pLentipuro3/TO/V5-GW/EGFP-Firefly Luciferase | Addgene | 119816 |
| LentiCRISPRv2GFP | Addgene | #82416 |
| LentiCRISPRv2mCherry | Addgene | #99154 |
| **Antibodies** | | |
| anti-Alpha tubulin | GeneTex | GTX628802 |
| Anti-Beta Actin | Sigma | A5316 |
| Anti-CD-44 | Abcam | Ab189524 |
| anti-E-Cadherin | CST | 3195 |
| Anti-EdU | Thermo Fisher Scientific | C10639 |
| Anti-EFTUD2 | Novus | NBP1-87019 |

| Reagent/resource | Reference or source | Identifier or catalog number |
|---|---|---|
| anti-GAPDH | Millipore | MAB374 |
| Anti-HSP90 | stressMarq | SMC-137 |
| anti-Ki67 | CST | 12202S |
| anti-N-Cadherin | CST | 13116 |
| anti-NUAK1 | CST | 4458S |
| anti-NUAK2 | Abcam | ab224079 |
| Anti-PCNA | Sigma | MAB424 |
| Anti-MYPT1 | ProteinTech | 22117-1-AP |
| Anti-YAP1 | CST | 14074 |
| Anti-TAZ | CST | 4883 |
| Anti-LATS1 | CST | C66B5 |
| anti-Slug | CST | 9585 |
| anti-Snail | CST | 3879 |
| Anti-Stem121 | Takara | Y40410 |
| anti-Vimentin | CST | 5741 |
| anti-ZEB1 | CST | 3396 |
| anti-ZO-1 | CST | 8193 |
| anti-β-Catenin | CST | 8480 |
| Anti-Stem121 | Takara | Y40410 |
| Cell cycle regulation sampler antibody kit | CST | 9932 |
| Hoechst 33258 | Sigma | B2883 |
| **Oligonucleotides and other sequence-based reagents** | | |
| PCR Primers | This study | EV Table 1 |
| sgRNAs | This study | See Methods |
| **Chemicals, enzymes and other reagents** | | |
| U87MG | ATCC | #HTB-14 |
| LN229 | ATCC | #CRL-2611 |
| U251MG | Addexbio | # C0005029 |
| LN319 | Addexbio | #C0005001 |
| GSC11 | Vadla R, Miki S, Taylor B, Kawauchi D, Jones BM, Nathwani N, Pham P, Tsang J, Nathanson DA, Furnari FB. Glioblastoma Mesenchymal Transition and Invasion are Dependent on a NF-κB/BRD2 Chromatin Complex. bioRxiv [Preprint]. 2023 Jul 3:2023.07.03.546613. https://doi.org/10.1101/2023.07.03.546613. PMID: 37461511; PMCID: PMC10349949. | |
| GSC23 | Vadla R, Miki S, Taylor B, Kawauchi D, Jones BM, Nathwani N, Pham P, Tsang J, Nathanson DA, Furnari FB. Glioblastoma Mesenchymal Transition and Invasion are Dependent on a NF-κB/BRD2 Chromatin Complex. bioRxiv [Preprint]. 2023 Jul 3:2023.07.03.546613. https://doi.org/10.1101/2023.07.03.546613. PMID: 37461511; PMCID: PMC10349949. | |

| Reagent/resource | Reference or source | Identifier or catalog number |
|---|---|---|
| GSC6 | Vadla R, Miki S, Taylor B, Kawauchi D, Jones BM, Nathwani N, Pham P, Tsang J, Nathanson DA, Furnari FB. Glioblastoma Mesenchymal Transition and Invasion are Dependent on a NF-κB/BRD2 Chromatin Complex. bioRxiv [Preprint]. 2023 Jul 3:2023.07.03.546613. https://doi.org/10.1101/2023.07.03.546613. PMID: 37461511; PMCID: PMC10349949. | |
| GSC39 | Vadla R, Miki S, Taylor B, Kawauchi D, Jones BM, Nathwani N, Pham P, Tsang J, Nathanson DA, Furnari FB. Glioblastoma Mesenchymal Transition and Invasion are Dependent on a NF-κB/BRD2 Chromatin Complex. bioRxiv [Preprint]. 2023 Jul 3:2023.07.03.546613. https://doi.org/10.1101/2023.07.03.546613. PMID: 37461511; PMCID: PMC10349949. | |
| 0.2-μm Nitrocellulose membrane | VWR | Cat#10119-996 |
| 10 cm Cell Culture Plates | VWR | 10062-880 |
| 2-Mercaptoethanol | Sigma | M6250 |
| 40% Acrylamide/Bis Solution | Bio-Rad | 1610148 |
| Six-well low attach plates | fisher | 29443-030 |
| Accutase | Innovative Cell Technologies, Inc | AT104-500 |
| B27 supplement | Thermo Fisher Scientific | Cat#17504044 |
| Bradford protein assay | Sigma | Cat#B6916 |
| BSA | Sigma | Cat#A7906 |
| Circular glass coverslips | Thermo Fisher Scientific | #50949008 |
| Click-iT™ Plus EdU Cell Proliferation Kit for Imaging, Alexa Fluor™ 594 dye | Invitrogen | C10639 |
| Crystal Violet | VWR | AAB21932-14 |
| D-Luciferin | Perkin Elmer | 122799 |
| DMEM (high glucose) | Gibco | 11995073 |
| DMSO | Sigma | Cat#D8418 |
| Donkey serum | Sigma | Cat#S30-M |
| Dulbecco's Modified Eagle Medium/F12 | Thermo Fisher Scientific | Cat# 11320033 |
| EDTA | Sigma | Cat#EDS-100G |
| Eosin | Thermo Fisher Scientific | NC9077264 |
| Fast Green Dye | Sigma | F7252-5g |
| fetal bovine serum (FBS) | Thermo Fisher Scientific | Cat#16000044 |
| Firefly Luciferase Glow Assay Kit | Thermo Fisher Scientific | Cat#16176 |
| Formaldehyde, 37%, Microfiltered | Electron Microscopy Sciences | 15686 |
| Geltrex™ | Gibco | A1413302 |

| Reagent/resource | Reference or source | Identifier or catalog number |
|---|---|---|
| Glass Coverslips (50X24mm) | Fisher | 12-541-023 |
| Glo 3D Cell Viability Assay reagent | Promega | #G9681 |
| Goat serum | Sigma | Cat#S26-M |
| Halt Protease + Phosphatase Inhibitor Cocktail, EDTA-Free (100X) | Fisher | 78441 |
| Hemataxolin | Thermo Fisher Scientific | NC0242438 |
| HEPES | Sigma | Cat#H3375 |
| Histobond charged slides | VWR | 16004-406 |
| HTH-02-006 | Aobious | #AOB36960 |
| ImmPRESS® Excel Amplified Polymer Staining Kit Anti-Rabbit IgG | Vector Laboratories | #MP-7601 |
| ImmPRESS® Excel Amplified Polymer Staining Kit, Anti-Mouse IgG | Vector Laboratories | MP-7602 |
| iScript cDNA synthesis kit | Bio-Rad | #1708891 |
| LookOut Mycoplasma PCR Detection Kit | Sigma | MP0035 |
| Methanol | VWR | BDH1135-4LP |
| MTT assay kit | Roche | #11465007001 |
| N2 supplement | Thermo Fisher Scientific | 17502048 |
| Ne-Per kit | fisher | 78835 |
| Neutral-buffered formalin 10% | VWR | 10790-714 |
| NP-40 | Sigma | Cat#492016 |
| PBS | Thermo Fisher Scientific | Cat#10010-23 |
| penicillin-streptomycin | Thermo Fisher Scientific | Cat#15070063 |
| PerfeCTa® SYBR® Green FastMix® | Quantabio | #101414-270 |
| Permount | VWR | 100496-500 |
| Phos-Tag Acrylimide | Wako | AAL-107 |
| POLY-L-ornithine | Sigma | P4957 |
| Precision Plus Protein Dual color stds | Bio-Rad | Cat#1610374 |
| **Software** | | |
| GraphPad Prism9 | GraphPad Software, Inc | RRID:SCR_002798 |
| QuPath | https://qupath.github.io/ | RRID:SCR_018257 |
| ImageJ-Fiji | https://imagej.net/software/fiji/ | RRID:SCR_002285 |
| Adobe Photoshop | Adobe Systems | RRID:SCR_014199 |
| Adobe Illustrator | Adobe Systems | RRID:SCR_010279 |
| **Other** | | |

## Animals

CD-1 IGS (Charles River #022) timed pregnant mice were used for IUE studies. BALB/c nude immunocompromised mice were obtained from the University of California, San Diego (UCSD) in-house breeding program for xenograft studies. All animal experiments were performed in accordance with approved protocols and guidelines. Animals were housed in a temperature-controlled environment (22 °C, 30–70% humidity) with a 12-h light/dark cycle, and food and water were available ad libitum. Care of all animals in this study was approved by the UCSD Institutional Animal Care and Use Committee (IACUC, #S18174) and followed NIH guidelines and procedures.

## Cell culture and reagents

Four GBM cell lines (U87MG, LN229, U251MG, and LN319) were used in this study. Authenticated U87MG and LN229 glioblastoma cell lines were purchased from ATCC (#HTB-14, #CRL-2611). Authenticated U251MG and LN319 cell lines were obtained from Addexbio (#C0005029, #C0005001). U87 was grown in Dulbecco's Modified Eagle Medium/F12 (DMEM/F12) supplemented with 10% fetal bovine serum (FBS) and 1% penicillin-streptomycin (PS). LN229, U251MG, and LN319 were cultured in DMEM with 10% FBS and 1% PS. HTH-02-006 (#AOB36960) was purchased from Aobious and dissolved in dimethyl sulfoxide (DMSO). Glioma stem cell neurosphere cultures were maintained in low-attachment plates in GSC media: DMEM/F12 supplemented with N12, B27 (Invitrogen), 10 ng/mL bFGF (cat # 133-FB), 10 ng/mL EGF (cat # 236-EG), R&D Systems and 1% PS. For experiments, neurospheres of five or fewer passages were used. All cells were maintained in humidified incubators at 37 °C and 5% $CO_2$. Cell lines were tested for mycoplasma using the LookOut Mycoplasma PCR Detection Kit (Sigma; MP0035).

## Orthotopic xenograft models

Seven- to eight-week-old male BALB/c nude mice were used to generate cell line xenograft models. U251 wildtype and CRISPR-edited cells were transduced with a lentivirus expressing luciferase (pLentipuro3/TO/V5-GW/EGFP-Firefly Luciferase, Addgene #119816) prior to use in these assays. The luciferase expressing U251 wildtype and CRISPR-edited cells were dissociated with trypsin and resuspended at PBS ($1.7 \times 10^5$ cells/ul). $5 \times 10^5$ cells in 3 ul of PBS were injected into the specific coordinates (x, y, z = 0.5, −2.0, −3.0) from bregma using stereotaxic injection system (RWD Life Science). Bioluminescence-based in vivo imaging of xenograft mice was performed at 7, 14, 21, and 28 days after cell injection using a Perkin Elmer IVIS Spectrum imaging system. Mice were intraperitoneally injected with 10 µL/g body weight of 15 mg/ml D-luciferin, anesthetized, and placed in IVIS Spectrum bioluminescent and fluorescent imaging systems (Perkin Elmer). Luminescence signals were developed and acquired per minute, followed by a 1-min exposure time for 10–15 min. To quantify bioluminescent intensity, a region of interest (ROI) was selected and analyzed using IVIS software. Brains were harvested, fixed, and embedded for histology analysis on the 28th day post-injection.

## In utero electroporation (IUE)

In utero electroporation was used to generate mouse gliomas as previously described (Chen and LoTurco, 2012; Glasgow et al, 2014). In short, the uterine horns of E15 pregnant females were exposed, and the appropriate DNA cocktail containing 1X Fast

Green dye indicator was injected into the lateral ventricles of embryos. The embryos were then electroporated with BTW Tweezertrodes connected to a BTX 8300 electroporator. The settings for electroporation were: 33 V, 55 ms per pulse, conducted six times, at 100 ms intervals. DNA combinations used were the helper plasmid pGLAST-PBase (2.0 μg/μL), pbCAG-GFP, pbCAG-Luciferase, crNF1, crPTEN, and crp53, and either *Nuak2*-expressing or *Nuak2*-targeting sgRNA plasmids, all at a concentration of 1.0 μg/μL each (Chen and LoTurco, 2012; Glasgow et al, 2017; John Lin et al, 2017). Mouse-specific *Nuak2* sgRNAs (Yuan et al, 2018) targeting exon 1 (5′-CCTCGCGGTCCCCGCACCAT-3′ and 5′-CTACGAGTTCCTGGAGACGC-3′) and non-targeting control (5′-ATGTTGCAGTTCGGCTCGAT-3′) were cloned into pX330. Animals were sacrificed at various time points and processed for further analysis. Bioluminescent imaging was performed as described for orthotopic transplants.

## Stable cell line generation

To generate *NUAK2* knockout lines in human glioblastoma cell lines, CRISPR guides targeting human *NUAK2* (5′-TGGAGTCGCTGGTTTTCGCG-3′) were cloned into a GFP- or mCherry-containing lentiviral vectors: LentiCRISPRv2GFP (Addgene#82416) and LentiCRISPRv2mCherry (Addgene #99154), respectively. Hek293T cells were transfected with the *NUAK2* sgRNA expressing GFP and mCherry vectors and the appropriate viral packaging plasmids using Viafect Transfection Reagent (Promega #E981) according to the manufacturer's instructions. The virus was collected over three days, combined, and filtered prior to the transduction of human GBM cell lines. Transduced cells expressing both GFP and mCherry were enriched using fluorescence-activated cell sorting (FACS). After transduction, cells were trypsinized and resuspended in 1 ml of FACS sorting buffer (0.1% BSA, 1% pen/strep, 1% 1 M HEPES pH 7, 25 mg/ml DNase in Leibowitz medium (Fisher, #21083027). Green/Red double-positive cells were sorted into 96-well plates, which was performed by UC San Diego Human Embryonic Stem Cell Core Facility using a BD FACSAriaII. Clones were cultured in a 96-well until 80–90% confluency, then transferred to plates with larger surfaces. From a 24-well plate, clones were screened by PCR and propagated for further experimentation.

For glioma stem cells, lentiviruses containing GFP control or Nuak2 were generated, and the neurospheres were transduced. Forty-eight hours after infection, cells were subjected to puromycin selection for 2–4 days to generate stably expressing populations, which were used for experimentation.

## Cell proliferation and clonogenic assays

MTT assays were conducted for two-dimensional proliferation assays using an MTT assay kit (Roche; #11465007001) following the manufacturer's protocol. About $1 \times 10^3$ cells per well were plated in 96-well plates. To validate the effect of the NUAK2 inhibitor, cells were treated with a complete growth medium containing various concentrations of HTH-02-006 (1, 2.5, 5, 10, 20 uM) for the indicated amount of time. 0.1% dimethyl sulfoxide (DMSO) was used as a control. Upon collection, HTH-02-006-treated cells were labeled with 10 ul of labeling solution per well for 4 h and lysed by adding 100 ul of solubilization reagent, followed by overnight incubation. The 570 and 630 nm absorbance were measured using a spectrophotometer (Perkin Elmer).

For colony formation assay, 500 cells/well were plated in six-well plates and maintained for 10–14 days. To evaluate the effect of HTH-02-006, 1000 cells/well were plated in six-well plates and grown for a week, then the cells were treated with various concentrations of the inhibitor for another week. Next, the colonies were fixed in 100% methanol and stained with 0.05% crystal violet solution. Excessive stains were removed by rinsing the plates with tap water. The plates were air-dried and photographed for quantification. Colony formation quantification was performed in Fiji (version 2.1.6.0) by manually isolating the well from the rest of the image, then thresholding using the default method. The images were then binarised, and the total area of the colonies per well was measured.

For EdU incorporation assays, glioma cells were plated on XX-coated coverslips at 1000 cells per 24-well. For GSCs cells were plated on Geltrex (1:100, Gibco A1413302) at 50,000cells per 24-well. EdU was added at a concentration of 20 uM for 3 h. Cells were then fixed in 4% Paraformaldehyde, washed with 1XPBS and stained using the Click-iT kit (Life Technologies C10639).

## Scratch assay

Cells were seeded at $2 \times 10^4$ cells per well in 96-well plates. After 24 h, uniform wounds were created using IncuCyte 96-well WoundMaker Tool as described in the manufacturer's protocol. After the scratch wound creation, cells were carefully washed twice with 1X PBS, treated, and maintained at indicated concentrations of HTH-02-006. The wound closure process was visualized every 12 h for 3 days and analyzed in real-time with the IncuCyte S3 live-cell imaging system (Sartorius Bioscience).

## Transwell migration assay

Cells were grown in regular media to 60% confluency in 10 cm plates. On the next day, the media was changed to serum-free media to starve the cells overnight. Then, 500 ul of complete media, including FBS, was placed into the wells of a 24-well plate. Transwell inserts (Thermo Fisher; # 07-200-150) were transferred into each well, creating an upper chamber. Serum-starved cells were harvested, and $1.5 \times 10^4$ cells/well were resuspended in 400 μL of serum-free medium and plated onto the upper chamber of the transwell insert. Cells were allowed to migrate while incubating at 37 °C for 40–46 h. Next, the media was gently removed from the inserts and washed with PBS, followed by fixation with 800 μL of 4% paraformaldehyde (PFA) in PBS, split between the lower and upper chambers. After 15 min of fixation at room temperature, inserts were washed twice with PBS. Cells were permeabilized with 100% cold methanol for 10 min, washed twice with PBS, and stained with 0.05% crystal violet for 15 min. Inserts were washed twice with PBS, and non-migrated cells were removed by gently scraping with cotton swabs. Membranes were then cut out, fixed in Permount on a slide, and imaged on an Olympus BX63 Microscope. Image analysis was performed using the ImageJ software 1.54. Images were converted to 8-bit, background was subtracted with a rolling ball radius of 30, and a Gaussian blur with a sigma of 1 was applied. Four individual ROI per image were then thresholded and counted using particle analysis.

## 3D spheroid analysis

To generate 3D spheroids of each GBM cell line, $1 \times 10^3$ cells suspended in serum-free media were plated in each well of ultra-low attachment 96-well plates (Corning; #7007) and briefly spun down by centrifugation. Spheroids were treated with various concentrations of HTH-02-006 after they formed circular masses, then imaged daily until day 6 after the initial drug treatment with ImageXpress MicroXLS (Molecular Devices) from the UCSD screening core laboratory. To determine viable cells in the spheroids, the Cell Titer Glo 3D Cell Viability Assay reagent (Promega; #G9681) was used as described in the manufacturer's protocol.

To determine spheroid sizes, batch image analysis of spheroids was conducted in Fiji (version 2.16.0) using a script to measure spheroid area. External plugins used in the script are Adjustable Watershed, BioVoxxel (version 2.6.0), and MorpholibJ (version 1.64). Spheroid image annotations were manually inspected for quality. Valid spheroid area measurements accurately traced the perimeter of the spheroid while excluding the surrounding cell monolayer. Dissociated spheroids were counted as having an area of zero. Out of the 1152 spheroid images for the four cell lines (U87, U251, LN229, and LN319), 36 images were manually annotated. For manual annotation, the area of the dissociated spheroids was set to zero, or the spheroid was manually traced in yellow, and its contained area was measured. Four images were removed due to poor quality.

## Neurosphere formation assay

Glioma stem cells were plated at a density of 1000 cells per well in an ultra-low attachment 96-well plate. Spheres were grown in serum-free media and growth was monitored every 48 h for 6 days using IncuCyte S3 live-cell imaging system (Sartorius Bioscience). Spheroid size was determined as described in the 3D spheroid section above.

## Quantitative real-time PCR (qRT-PCR)

Total RNA was isolated using Trizol (Invitrogen) solution following the manufacturer's protocol. Trizol reagent was added directly to cells or tissues, and lysates were either immediately processed or stored at −80 °C. The RNA concentrations were measured with a NanoDrop spectrophotometer (Thermo Fisher). cDNAs were generated from 0.5 ug of total RNA per sample by reverse transcription using the iScript cDNA synthesis kit (Bio-Rad; #1708891). Samples were analyzed by CFX384 real-time system (Bio-Rad) using PerfeCTa® SYBR® Green FastMix® (Quantabio; #101414-270) according to the manufacturer's protocol. Gene expression was normalized to the housekeeping gene *GAPDH*. See Appendix Table S1 for the list of qPCR primers used in the study.

## Western blot

Cells were lysed in radioimmunoprecipitation assay (RIPA; 150 mM Sodium chloride, 50 mM Tris-HCl, 1% NP-40, 0.5% sodium deoxycholate, 0.1% SDS) buffer with ethylenediaminetetraacetic acid (EDTA) free protease/phosphatase inhibitor cocktail (Thermo Fisher; #78441) and kept at −20 °C for long-term storage and future analysis. Nuclear fractionation was performed when appropriate using the NE-PER Nuclear and Cytoplasmic Extraction Kit [78833] following the manufacturer's protocol. Protein concentration was determined by performing the Bradford assay (Sigma; #B6916). A total of 20–40 ug of protein lysates were resolved with polyacrylamide gel electrophoresis (8–10% Tris-HCl SDS–PAGE gels) and transferred onto either nitrocellulose or polyvinylidene difluoride (PVDF) membrane based on the molecular weight of the target protein. Membranes were then immersed in 5% bovine serum albumin (BSA) and incubated for 1 h at room temperature, followed by overnight incubation with primary antibodies at 4 °C. Membranes were washed with tris-buffered saline (TBS) with 0.1% Tween and then incubated with secondary antibodies for 1 h. Lastly, the target protein signal was developed using Western Blotting Luminol reagent (Santa Cruz Biotechnology; #sc-2048). For re-probing blots, membranes were stripped in stripping buffer (1% SDS, 62.5 mM Tris-HCl, pH 6.8, 0.8% beta-mercaptoethanol) heated to 50 °C for 45 min; followed by rinsing with water for 2 mins, washing with TBS-T for 5 min, and lastly blocking in BSA for 1 h before addition of antibody. See Appendix Table S2 for primary antibodies and specifications used for the study.

Phos-tag acrylamide was purchased from Wako Pure Chemical Industries (AAL-107). For gel preparation for the detection of Lats and MYPT1, 30 uM Phos-Tag acrylamide was added to a 6% polyacrylamide separating gel containing 60 uM of $MnCl_2$ according to the manufacturer's protocol. For gel preparation for the detection of Yap, 20 uM Phos-Tag acrylamide was added to a 6% polyacrylamide separating gel containing 40 uM of $MnCl_2$. For the detection of Taz, 20 uM Phos-Tag acrylamide was added to a 7.5% polyacrylamide separating gel containing 40 uM of $MnCl_2$. After SDS–PAGE was performed, gels were incubated in transfer buffer containing 10 mM EDTA for 20 min before being equilibrated in transfer buffer and proceeding to transfer to PVDF membrane.

## Immunohistochemistry (IHC-P)

For paraffin embedding, mice were perfused with PBS followed by 10% neutral-buffered formalin for whole-body fixation. Fixed brains were dissected and drop-fixed in 10% neutral-buffered formalin for 16 h, followed by 24 h incubation in 70% ethanol. The brains were processed for paraffin embedding at the UCSD Biorepository and Tissue Technology core. Brains were sectioned at 5 um using a Leica microtome and allowed to dry for analysis. Sections were deparaffinized using xylene and a series of decreasing ethanol concentration washes. Sections were washed with TBS-T, and antigen retrieval was performed using sodium citrate buffer (pH 6.0) at 95 °C for 15 min. Immunohistochemistry was performed using ImmPRESS® Excel Amplified Polymer Staining Kit (Vector Laboratories; #MP-7601). Briefly, sections were washed with TBS-T before using BLOXALL® Endogenous HRP/AP Blocking Solution for 10 min, followed by two washes with TBS-T. Sections were blocked in 2.5% horse serum for 30 min followed by incubation in primary antibodies, either Ki67 (Cell Signaling Technologies; D3B5) 1:500 or NUAK2 (Novus Biologicals; NBP1-81880) 1:50 antibodies overnight at 4 °C. Sections were washed three times with TBS-T before applying Amplifier Antibody (Goat Anti-Rabbit IgG) for 15 min. Sections were washed three times with TBS-T and ImmPRESS Horse Anti-Goat IgG Polymer Reagent was applied for 30 min. Before chromogenic detection, sections were washed twice with TBS-T before the DAB substrate was applied and

allowed to develop for 2 min. The slides were washed three times with TBS-T before counterstaining with hematoxylin and dehydrating through a series of increasing ethanol concentrations and xylene incubations. Sections were mounted and dried for 24 h prior to imaging. The percentage of positively stained cells was analyzed using QuPath software cell detection protocol on 20X images of tumor areas. Three separate $500 \times 500$ pixel squares were counted for each sample.

## Immunocytochemistry (ICC)

Circular glass coverslips (Fisher; #50949008, 12 mm) were coated with 0.01% poly-L-ornithine solution (Sigma; P4957) overnight prior to cell seeding. The next day, an appropriate number of cells were plated onto the coverslips to yield ~70% confluency. After treatment, cells were fixed using cold 4% PFA for 15 min, followed by permeabilization with 0.1% Triton X-100 in PBS for three minutes with gentle agitation. The coverslips were washed with PBS and incubated with 3% BSA blocking buffer (3% BSA in PBS (w/v)) for 1 h at room temperature. After blocking, primary antibodies diluted in the same blocking buffer were added onto coverslips and incubated overnight at 4 °C, Coverslips were then washed with PBS and incubated with fluorescence-conjugated secondary antibodies for 1–2 h at room temperature, washed with PBS, and nuclei stained with Hoechst 33258 (Sigma; #B2883). Coverslips were mounted using an anti-fade mounting medium (Vectashield; H-1400), dried overnight, and imaged using the fluorescence microscope (Olympus). Images were analyzed and quantified using FIJI software.

## RNA-sequencing (RNA-seq)

### Sample preparation
Total RNA was isolated using TRIzol Reagent following the manufacturer's protocol. The quality of total RNA was evaluated using an Agilent Tapestation 4200, and only samples with an RNA Integrity Number (RIN) above 9.0 were selected for RNA-seq library preparation with the Illumina® Stranded mRNA Prep kit (Illumina, San Diego, CA). Library preparation was conducted according to the manufacturer's protocol by the UCSD Institute for Genomic Medicine (IGM) Core Facility. The prepared libraries were multiplexed and sequenced using 100 base pair (bp) paired-end reads (PE100) on an Illumina NovaSeq 6000, achieving a sequencing depth of ~25 million reads per sample. Demultiplexing was performed with the bcl2fastq Conversion Software (Illumina, San Diego, CA).

## Data analysis

For U251 RNA-seq analysis, FASTQ files were processed in Galaxy using Trimmomatic with default parameters. Read alignment was performed with HISAT2 using the hg38 reference genome and default parameters. Raw expression data were obtained using feature Counts with default parameters. Differential expression analysis was performed in R (version 4.4.1) using the DESeq2 package (version 1.46.0). Differentially expressed genes (DEGs) were determined based on a significance threshold adjusted $p$ value of <0.05 and a $\log_2$ fold change (LFC) >2. About 685 DEGs, with 273 upregulated and 385 downregulated, were identified. Differential expression analysis for TCGA GBM was obtained from the open-access web application for data visualization and analysis, GlioVis, which compared the highest 25% and lowest 25% NUAK2-expressing samples. DEGs were determined based on a significance threshold adjusted $p$ value < 0.05

and an LFC >1. About 1494 DEGs, with 807 upregulated and 687 downregulated, were identified.

## Over-representation analysis

Over-representation analysis (ORA) was performed using the clusterProfiler package (version 4.14.0). Gene Ontology (GO) terms for Biological Process (BP) and Cellular Component (CC) categories were identified using the set of 17,767 genes for U251 and 20,501 for TCGA GBM as background. A significance threshold of $p < 0.01$ and $q < 0.05$ was applied. The "simplify" method from clusterProfiler with default parameters was used to remove redundant GO terms. The Benjamini–Hochberg procedure was used for multiple-hypothesis testing correction.

## Gene set enrichment analysis

Gene set enrichment analysis (GSEA) was done using clusterProfiler, with genes ranked by the Wald statistic generated from DESeq2 for U251 and calculated by GlioVis for TCGA_GBM. GO: BP terms were analyzed using default parameters with a minimum gene set size of 50. GSEA using gene sets for the EMT, MES signature, and PN signature were obtained from MSigDb under the systematic names M817, M2122, and M2115, respectively. The EMT gene set is equivalent to the GO Biological Process term "epithelial to mesenchymal transition." Analysis using these gene sets was performed separately with default parameters. A $p$ value cutoff of 0.05 was used for all analyses. Benjamini–Hochberg procedure was used for multiple-hypothesis correction. Kyoto Encyclopedia of Genes and Genomes (KEGG) pathway enrichment analysis was performed using the Galaxy platform Goseq and Pathveiw tools (Luo and Brouwer, 2013; Young et al, 2010).

## Statistical analysis

All data were included in the analysis, and no exclusion criteria were applied. For mouse experiments, sample sizes were estimated based on our previous studies. Statistical analyses were conducted using GraphPad Prism 10 software. The data represent findings from at least three independent experiments. Unpaired $t$-tests were used to assess significance ($p < 0.05$). Kaplan–Meier survival curves were generated, and survival comparisons were evaluated using the log-rank (Mantel–Cox) test. More statistical information is reported in Appendix Table S3. Whenever possible, analysis was performed blind to genotype or condition for all studies.

## Graphics

Figure 4A and the synopsis were created with BioRender.com.

# Data availability

Source data for Fig. 7 has been made available on BioImage Archive accession number S-BIAD2082 (https://www.ebi.ac.uk/biostudies/bioimages/studies/S-BIAD2082). The RNA-sequencing results have been deposited to the Gene Expression Omnibus (GEO) and can be found under the accession number GSE285513 (https://www.ncbi.nlm.nih.gov/geo/query/acc.cgi?acc=GSE285513).

## The paper explained

### Problem

Glioblastoma (GBM) is the most aggressive and common malignant primary brain tumor in adults. Despite advances in treatments, the prognosis for GBM patients remains poor. GBM recapitulates normal neurodevelopmental programs, but the developmental interface with malignancy is poorly defined. Understanding these neurodevelopmental programs is crucial for developing better treatments for GBM.

### Results

Our study identifies NUAK2 as a fetal oncogene important for regulating migration and proliferation in GBM cells. CRISPR-mediated deletion of NUAK2 in glioma cells attenuated proliferation and migration, while NUAK2 overexpression promotes these processes. Similarly, in an in vivo mouse model of malignant glioma, NUAK2 deletion suppressed tumor growth, and NUAK2 overexpression promoted tumor growth. Importantly, pharmacological inhibition of NUAK2 kinase activity inhibited proliferation and migration.

### Impact

Our data demonstrate that targeting NUAK2 kinase markedly inhibited the proliferation and migration of GBM cells, suggesting that NUAK2 is an actionable therapeutic target for GBM treatment.

All scripts and RNA-sequencing analysis code can be found in the following GitHub repository: https://github.com/smglasgowlab/nuak2-2024.

The source data of this paper are collected in the following database record: biostudies:S-SCDT-10_1038-S44321-025-00287-3.

## Peer review information

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

## Acknowledgements

We would like to thank the University of California, San Diego (UCSD) IGEM Core, the UCSD Human Embryonic Stem Cell Core Facility (hESCCF), and the Screening Core Laboratory directed by Dr. Jair Siqueira-Neto of the UCSD Center for Drug Discovery Innovation for their expert assistance. This study was supported by NIH/NINDS 1R01NS123385 and the Hellman Foundation Fellowship grants, both awarded to SMG.

## Author contributions

**Hanhee Jo**: Investigation; Writing—original draft. **Sarah Munoz**: Data curation; Formal analysis; Investigation; Writing—review and editing. **Aneesh Dalvi**: Investigation. **Wenqi Yang**: Investigation. **Elizabeth Morozova**: Investigation. **Stacey M Glasgow**: Conceptualization; Supervision; Investigation; Writing—original draft; Project administration; Writing—review and editing.

Source data underlying figure panels in this paper may have individual authorship assigned. Where available, figure panel/source data authorship is listed in the following database record: biostudies:S-SCDT-10_1038-S44321-025-00287-3.

## Disclosure and competing interests statement

The authors declare no competing interests.

# Expanded View Figures

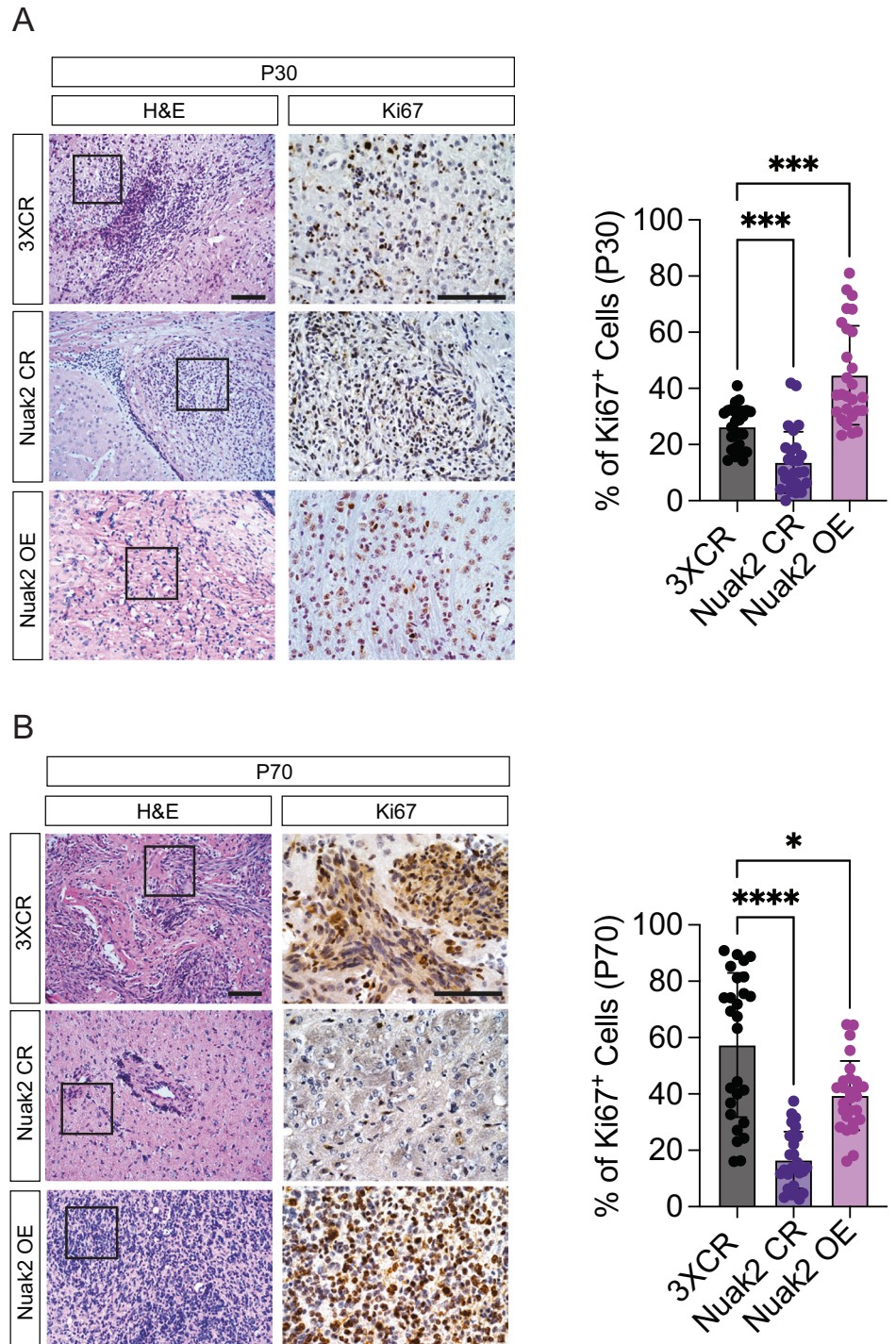

**Figure EV1.  Histological analysis of P30 and P70 IUE.**

(**A**) Representative images of H&E and Ki67 proliferating cells in control, NUAK2-deleted or OE tumors at P30. Scale bar = 100 μm. Quantification analysis of Ki67-positive cells is represented as mean ± SD (***$p$ = 0.0001; Statistical significance was determined by two-way RM ANOVA analysis followed by Dunnett's multiple comparisons test). Three images from three independent brains were used for quantification. (**B**) Representative images of H&E and Ki67 proliferating cells in control, NUAK2 deleted or OE tumors at P70. Scale bar = 100 μm. Quantification analysis of Ki67-positive cells is represented as mean ± SD (*$p$ = 0.016, ****$p$ = 3.5E-10; Statistical significance was determined by two-way RM ANOVA analysis followed by Dunnett's multiple comparisons test). Three images from three independent brains were used for quantification.

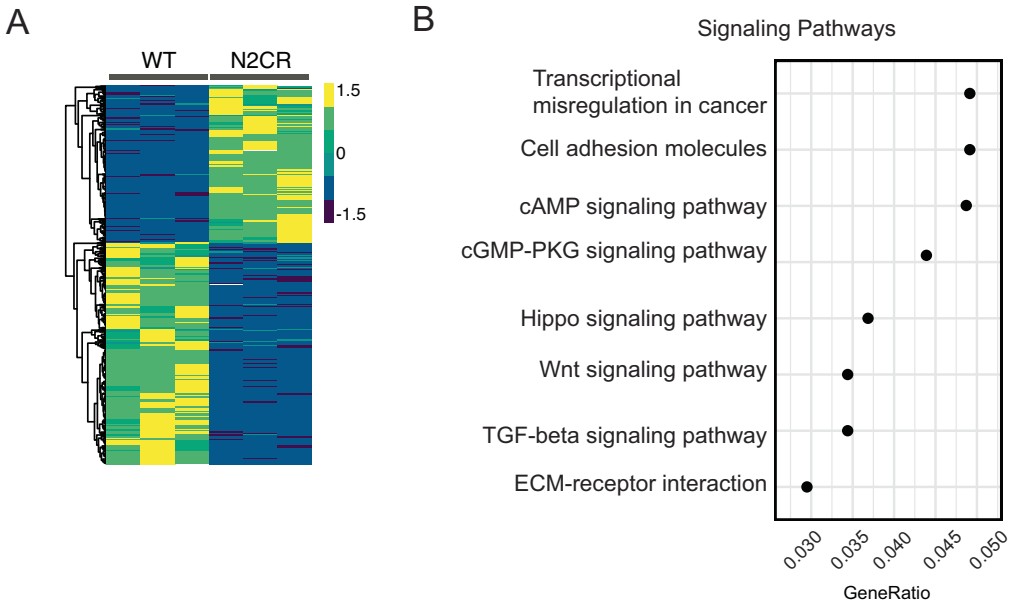

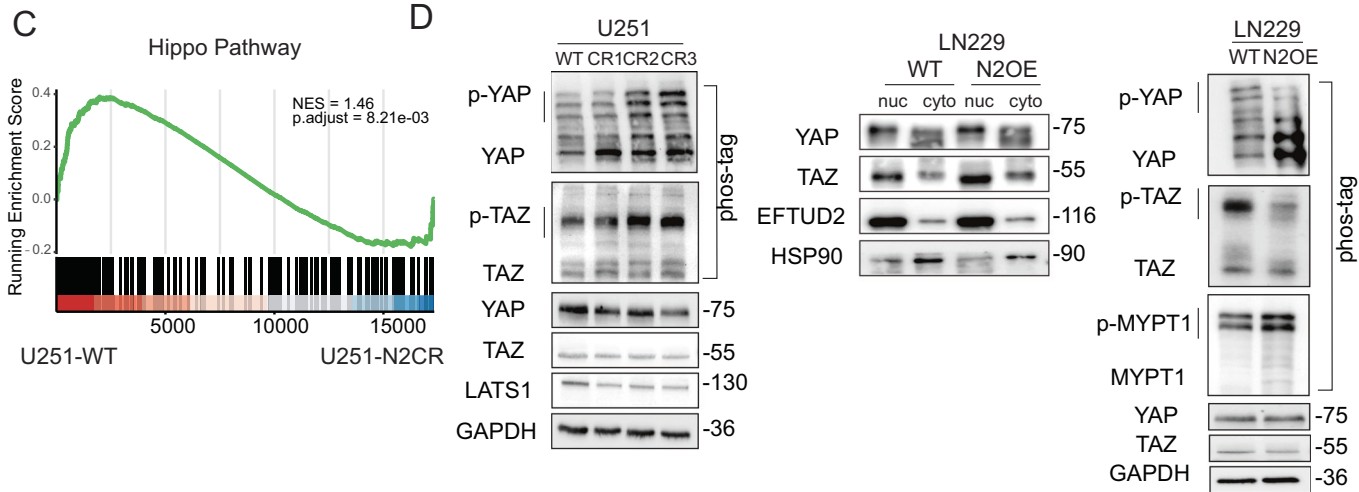

**Figure EV2. NUAK2 association with the Hippo signaling pathway.**

(A) Heatmap of differentially expressed genes in U251-control and U251 NUAK2 CRISPR-deleted cells. (B) KEGG pathway analysis of U251 NUAK2-CR DEGs. (C) GSEA enrichment plots of U251 NUAK2-CR DEGs showing enrichment in genes associated with the Hippo signaling pathway. Enrichment significance was assessed using the false discovery rate (FDR) method, and adjusted p-values (padj) were calculated using the Benjamini–Hochberg procedure. (D) Representative western blots analyzing downstream components of the Hippo pathway. Left panels: phos-tag western blots separating phosphorylated and non-phosphorylated YAP and TAZ in NUAK2-CR clones. Total YAP, TAZ, and LATS1 are also represented using a standard Western blot. GAPDH was used as a loading control. Molecular weights are shown to the right. Middle panels: standard Western blots of nuclear versus cytoplasmic fractions in LN229 NUAK2 overexpression lysates. Nuclear enrichment of YAP and TAZ is shown in the upper blots. Lower blots for EFTUD2 (nuclear) and HSP90 (cytoplasmic) confirm fractions. Right panels: phos-tag western blots separating phosphorylated and non-phosphorylated YAP and TAZ in LN229 NUAK2 overexpression lysates. Phos-tag blot showing enrichment of p-MYPT1 confirms that overexpressed NUAK2 is active in LN229 cells. Total YAP and TAZ are also represented using a standard Western blot. GAPDH was used as a loading control. Molecular weights are indicated to the right of the image.

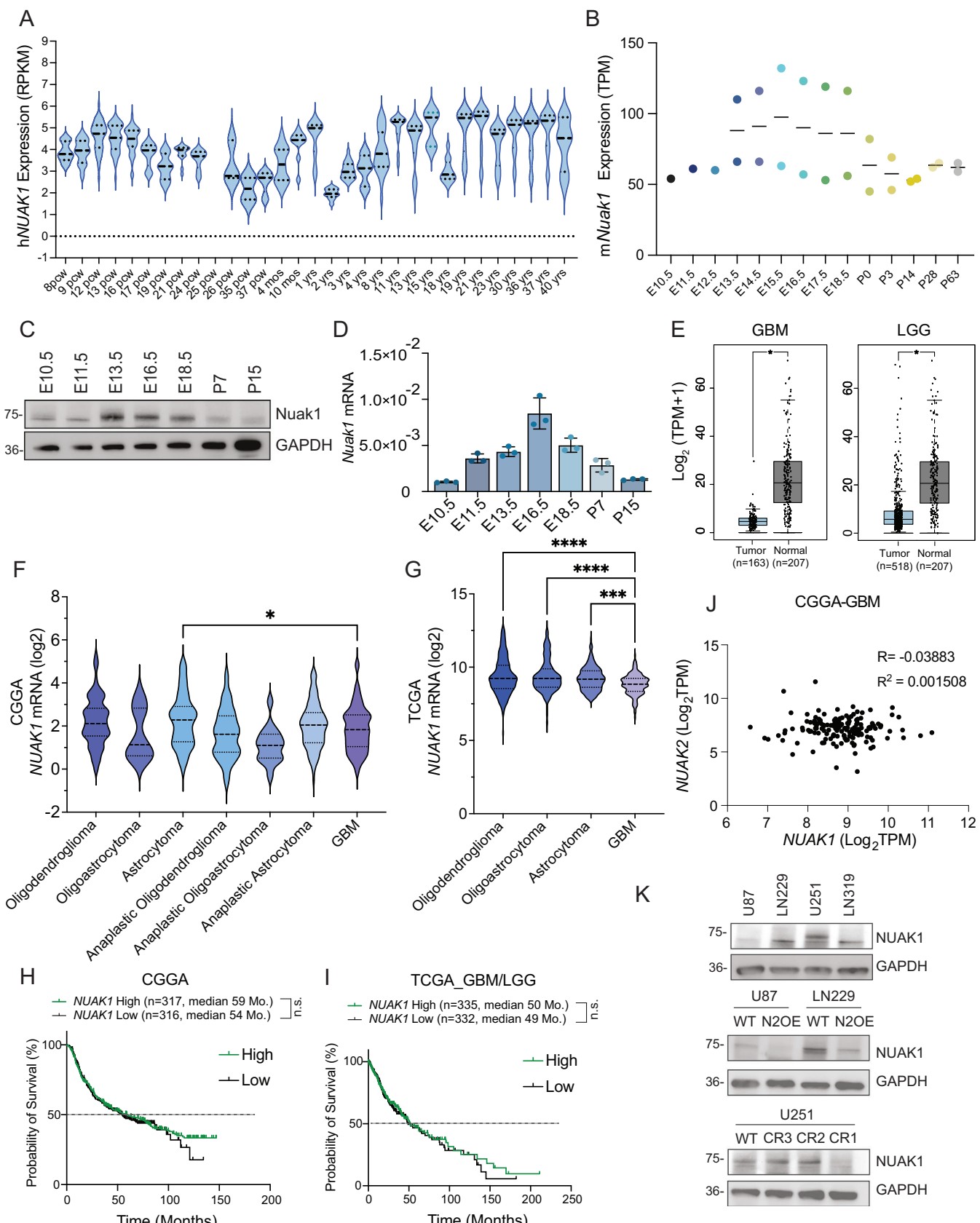

◄ **Figure EV3. NUAK1 is not associated with GBM progression and patient survival.**

(A) RPKM-normalized NUAK1 mRNA expression of specific human brain regions from eight post-conception weeks (pcw) to 40 years of age. Data were obtained from the BrainSpan Atlas. $n = 5$–17. (B) TPM-normalized NUAK1 mRNA expression of mouse forebrain or hindbrain, ranging from embryonic day 10.5 to postnatal day 63. Data was obtained from EMBL's European Bioinformatics Institute (EMBL-EBI; https://www.ebi.ac.uk/). (C) Representative western blot of NUAK1 protein expression in wild-type embryonic brain tissue across seven stages of development. GAPDH was used as the loading control. Nuak1 is 74 kDa and Gapdh is 36 kDa. (D) Representative RT-PCR of NUAK1 mRNA expression in wild-type embryonic brain tissues across developmental stages. (E) Normalized NUAK1 mRNA expression of TCGA GBM ($n = 163$) or LGG ($n = 518$) and GTEx non-tumor ($n = 207$) samples (*$p = 0.01$; Statistical significance is determined by one-way ANOVA). Box plots display the median (center line), interquartile range (box: 25th to 75th percentiles), and whiskers representing the minimum and maximum values. Data were obtained from GEPIA (http://gepia.cancer-pku.cn/). (F) NUAK1 mRNA expression across glioma subtypes in the CGGA dataset. Data were represented as mean ± SD (*$p = 0.0193$; Statistical significance is determined by one-way ANOVA followed by Tukey's multiple comparisons test). $n = 8$–225 depending on the grade of the tumor. Data were obtained from the GlioVis Database. (G) NUAK1 mRNA expression across glioma subtypes in the TCGA dataset. Data were represented as mean ± SD (***$p < 0.001$, ****$p < 0.0001$; Statistical significance is determined by one-way ANOVA followed by Tukey's multiple comparisons test). Exact $p$ values are reported in Appendix Table S3. $n = 130$–194, depending on the grade of the tumor. Data were obtained from the GlioVis Database. (H) Kaplan–Meier survival analysis from CGGA of high (21 days; $n = 317$) and low (145 days; $n = 316$) NUAK1 expressers shows no correlation with survival outcomes ($p = 0.682$; Statistical significance was determined by log-rank (Mantel–Cox) test). (I) Kaplan–Meier survival analysis from TCGA of high (15 days; $n = 335$) and low (134 days; $n = 332$) NUAK2 expressers shows no correlation with survival outcomes ($p = 0.6262$; Statistical significance was determined by log-rank (Mantel–Cox) test). (J) Spearman's rank coefficient plots demonstrating no correlation between NUAK1 and NUAK2 expression in human gliomas. Analyzed data were obtained from the CGGA GBM dataset. ($p = 0.6337$). (K) Top panels: Representative western blots of NUAK1 expression in U87, LN229, U251, and LN219. Middle panels: Western blots of NUAK1 expression in U87-N2OE and LN229-N2OE cells. Bottom panels: NUAK1 expression in U251-CRISPR-deleted clones. GAPDH was used as the loading control. Molecular weights are shown to the left of the blots.

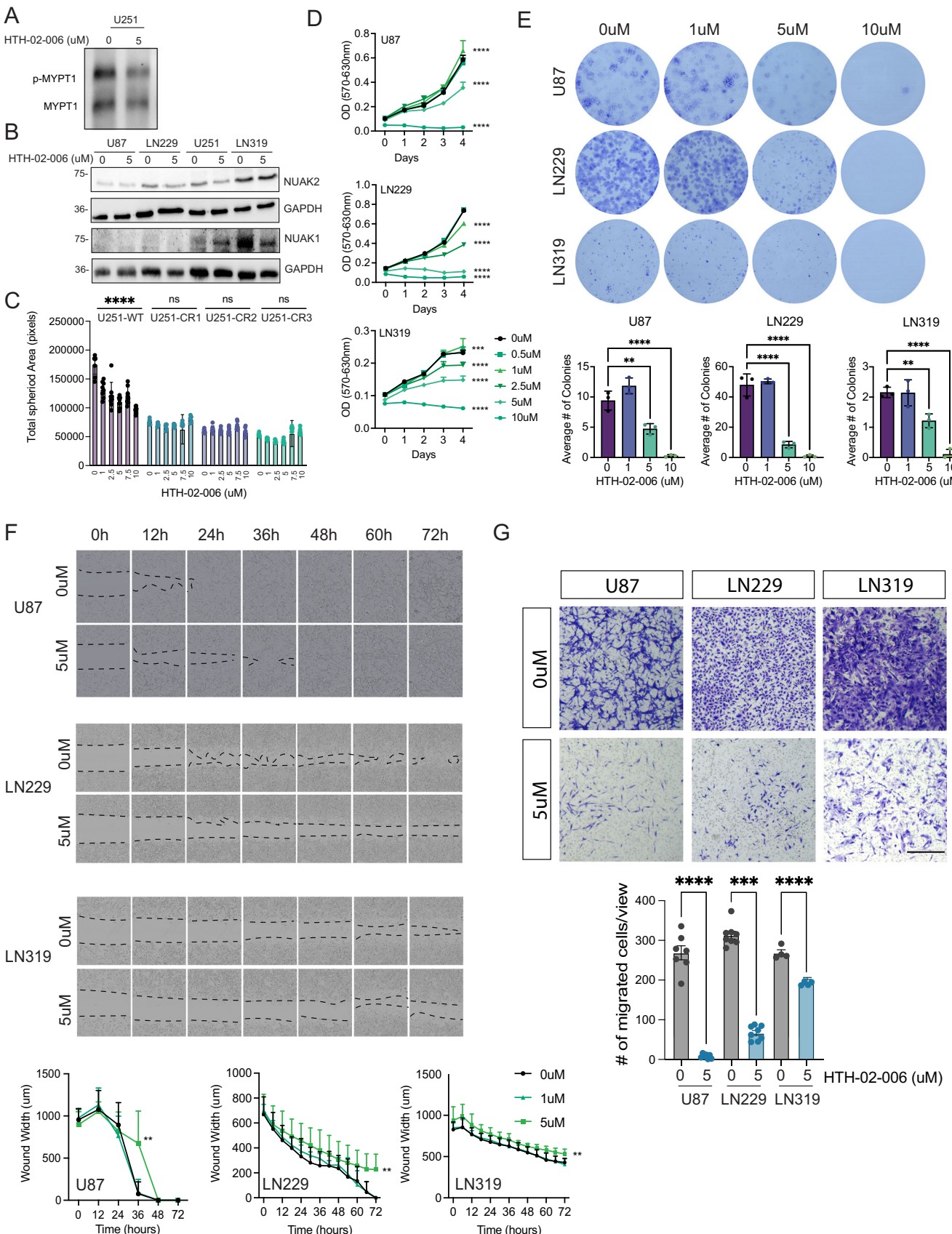

◀

**Figure EV4. NUAK2 inhibitor, HTH-02-006, attenuates GBM cell progression.**

(A) Representative phos-tag western blot of MYPT1 on lysates from U251 HTH-02-006 or vehicle-treated cells. HTH-02-006 creased phosphorylation of MYPT1. (B) Representative western blot of NAUK1 and NUAK2 expression in U87, LN229, U251, and LN319 cells treated with either 5 uM HTH-02-006 or DMSO vehicle control. GAPDH was used as a loading control. Molecular weights are shown to the left of the blot. (C) Quantification of total spheroid area of HTH-02-006-treated U251 spheroids grown for 6 days. Data were represented as mean ± SD ($n = 8$; ****$p < 0.0001$; Statistical significance was determined by one-way ANOVA analysis followed by trend testing. Exact $p$ values are reported in Appendix Table S3. (D) MTT assay for proliferation in HTH-02-006-treated U87 ($n = 6$), LN229 ($n = 6$), and LN219 ($n = 6$) cells. Data were represented as mean ± SD (***$p = 0.0008$, ****$p < 0.0001$; Statistical significance was determined by two-way RM ANOVA followed by Dunnett's multiple comparison test. Exact $p$ values are reported in Appendix Table S3. (E) Representative images of colony formation assay of U87 ($n = 3$), LN229 ($n = 3$), and LN319 ($n = 3$) cells with HTH-02-006 treatment. Quantification of colony formation assay (**$p < 0.01$, ****$p < 0.0001$; Statistical significance was determined by one-way ANOVA followed by Dunnett's multiple comparison test). Exact p values are reported in Appendix Table S3. Data were represented as mean ± SD. (F) Representative images and quantification of HTH-02-06-treated U87, LN229, and LN319 cell migration into the wound area. Data were represented as mean ± SD (**$p < 0.01$; Statistical significance was determined by two-way RM ANOVA followed by Dunnett's multiple comparison test). Exact $p$ values are reported in Appendix Table S3. The white dotted lines demarcate the wound boundary. Scale bar = 100 μm. (G) Representative images and quantification of transwell migration assay after HTH-02-006 treatment of U87 ($n = 7$), LN229 ($n = 8$), LN319 ($n = 4$) cells. Data were represented as mean ± SD (***$p < 0.001$, ****$p < 0.0001$; Statistical significance is determined by unpaired $t$-test (two-tailed)). Exact $p$ values are reported in Appendix Table S3. Scale bar = 500 μm.

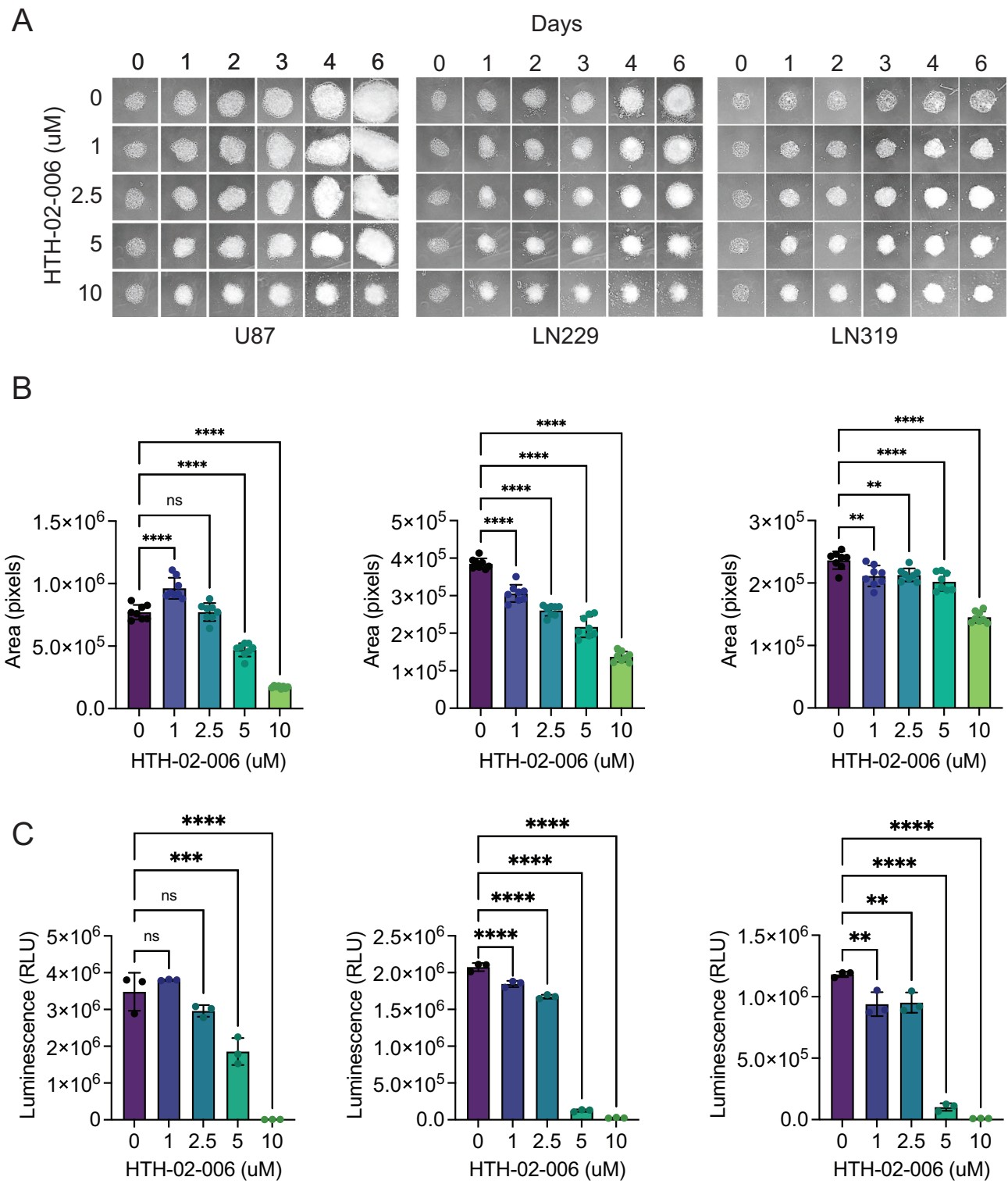

**Figure EV5. Efficacy of HTH-02-006 in 3D GBM spheroids.**

(A) Representative brightfield images of spheroid assay in HTH-02-06-treated U87, LN229, and LN319 cells over the course of 6 days. (B) Quantification of total spheroid area of HTH-02-06-treated U87, LN229, and LN319 cells spheroids. Data were represented as mean ± SD ($n = 8$, **$p < 0.01$, ****$p < 0.0001$; Statistical significance was determined by one-way ANOVA followed by Dunnett's multiple comparison test. Exact $p$ values are reported in Appendix Table S3. (C) Luminescence intensity of viable cells in HTH-02-06-treated U87, LN229, and LN319 spheroids at day 6. Data were represented as mean ± SD ($n = 3$, **$p < 0.01$, ***$p < 0.001$, ****$p < 0.0001$; Statistical significance was determined by one-way ANOVA followed by Dunnett's multiple comparison test. Exact $p$ values are reported in Appendix Table S3.

