## [Peer Review File · EMBO Molecular Medicine]

A fetal oncogene NIAK2 is an emerging therapeutic target in glioblastoma

Hanhee Jo, Sarah Munoz, Aneesh Dalvi, Wenqi Yang, Elizabeth Morozova, and Stacey Glasgow

Corresponding author: Stacey Glasgow (sglasgow@ucsd.edu)

Review Timeline:

Submission Date:	1st Jan 25
Editorial Decision:	7th Feb 25
Revision Received:	18th Jun 25
Editorial Decision:	15th Jul 25
Revision Received:	18th Jul 25
Accepted:	21st Jul 25

Editor: Lise Roth

Transaction Report:

7th Feb 2025

Dear Dr. Glasgow,

Thank you for the submission of your manuscript to EMBO Molecular Medicine, and please accept my apologies for the delay in getting back to you as we were waiting for one referee report. Unfortunately, referee #3 has not yet gotten back to us despite several chasers, but given that both referees #1 and #2 provide similar recommendations, we prefer to make a decision now in order to avoid further delay in the process. Should referee #3 provide a report, we will send it to you, with the understanding that we will not ask you for extensive experiments in addition to the ones required in the enclosed reports from referees #1 and #2.

As you will see from the reports below, the referees acknowledge the interest of the study and are overall supporting publication of your work pending appropriate revisions. Upon further cross-commenting with the referees, we agreed that revisions should focus on the following points:

- Confirm the findings in a more adequate cellular model (could be limited to some of the results, i.e Fig.2)
- Provide a deeper mechanistic insight (including LoF/GoF in Hippo signaling pathway)
- Clarify the potential contribution of NUA1
- Provide additional information, methodological details, statistics, and discussion as detailed in the referees' reports.

Acceptance of the manuscript will entail a second round of review. EMBO Molecular Medicine encourages a single round of revision only and therefore, acceptance or rejection of the manuscript will depend on the completeness of your responses included in the next, final version of the manuscript. For this reason, and to save you from any frustrations in the end, I would strongly advise against returning an incomplete revision.

We are expecting your revised manuscript within three to four months, if you anticipate any delay, please contact us.

We require:

- 1) A .docx formatted version of the manuscript text (including legends for main figures, EV figures and tables). Please make sure that the changes are highlighted to be clearly visible.
- 2) Individual production quality figure files as .eps, .tif, .jpg (one file per figure). For guidance, download the 'Figure Guide PDF' (<https://www.embopress.org/page/journal/17574684/authorguide#figureformat>).
- 3) At EMBO Press we ask authors to provide source data for the main figures. Our source data coordinator will contact you to discuss which figure panels we would need source data for and will also provide you with helpful tips on how to upload and organize the files.
- 4) A .docx formatted letter INCLUDING the reviewers' reports and your detailed point-by-point responses to their comments. As part of the EMBO Press transparent editorial process, the point-by-point response is part of the Review Process File (RPF), which will be published alongside your paper.
- 5) A complete author checklist, which you can download from our author guidelines (<https://www.embopress.org/page/journal/17574684/authorguide#submissionofrevisions>). Please insert information in the checklist that is also reflected in the manuscript. The completed author checklist will also be part of the RPF.
- 6) All Materials and Methods need to be described in the main text using our 'Structured Methods' format. According to this format, the Methods section includes a Reagents and Tools Table (listing key reagents, experimental models, software and relevant equipment and including their sources and relevant identifiers) followed by a Methods and Protocols section describing the methods, ideally using a step-by-step protocol format. The aim is to facilitate adoption of the methodologies across labs. Please download and fill our Reagents and Tools Table template (.docx), which you can find in our author guidelines: <https://www.embopress.org/page/journal/14693178/authorguide#structuredmethods>. When submitting your revised manuscript, please do not include the Reagents and Tools Table in the Methods section of the manuscript but upload it as a separate file choosing the file type "Reagent Table". An example of a Method paper with Structured Methods can be found here: <https://www.embopress.org/doi/10.15252/msb.20178071>

7) Please note that all corresponding authors are required to supply an ORCID ID for their name upon submission of a revised manuscript.

8) It is mandatory to include a 'Data Availability' section after the Materials and Methods. Before submitting your revision, primary datasets produced in this study need to be deposited in an appropriate public database, and the accession numbers and database listed under 'Data Availability'. Please remember to provide a reviewer password if the datasets are not yet public (see <https://www.embopress.org/page/journal/17574684/authorguide#dataavailability>).

9) For data quantification: please specify the name of the statistical test used to generate error bars and P values, the number (n) of independent experiments (specify technical or biological replicates) underlying each data point and the test used to calculate p-values in each figure legend. The figure legends should contain a basic description of n, P and the test applied. Graphs must include a description of the bars and the error bars (s.d., s.e.m.). Please provide exact p values.

10) Our journal encourages inclusion of *data citations in the reference list* to directly cite datasets that were re-used and obtained from public databases. Data citations in the article text are distinct from normal bibliographical citations and should directly link to the database records from which the data can be accessed. In the main text, data citations are formatted as follows: "Data ref: Smith et al, 2001" or "Data ref: NCBI Sequence Read Archive PRJNA342805, 2017". In the Reference list, data citations must be labeled with "[DATASET]". A data reference must provide the database name, accession number/identifiers and a resolvable link to the landing page from which the data can be accessed at the end of the reference. Further instructions are available at .

11) We replaced Supplementary Information with Expanded View (EV) Figures and Tables that are collapsible/expandable online. A maximum of 5 EV Figures can be typeset. EV Figures should be cited as 'Figure EV1, Figure EV2" etc... in the text and their respective legends should be included in the main text after the legends of regular figures.

12) The paper explained: EMBO Molecular Medicine articles are accompanied by a summary of the articles to emphasize the major findings in the paper and their medical implications for the non-specialist reader. Please provide a draft summary of your article highlighting

13) Author contributions: CRediT has replaced the traditional author contributions section because it offers a systematic machine readable author contributions format that allows for more effective research assessment. Please remove the Authors Contributions from the manuscript and use the free text boxes beneath each contributing author's name in our system to add specific details on the author's contribution. More information is available in our guide to authors.

Please also suggest a visual abstract to illustrate your article as a PNG file 550 px wide x 300-600 px high. A cropped portion of this image will serve as thumbnail for the table of content on our webpage.

16) As part of the EMBO Publications transparent editorial process initiative (see our Editorial at <http://embomolmed.embopress.org/content/2/9/329>), EMBO Molecular Medicine will publish online a Review Process File (RPF) to accompany accepted manuscripts.

In the event of acceptance, this file will be published in conjunction with your paper and will include the anonymous referee reports, your point-by-point response and all pertinent correspondence relating to the manuscript. Let us know whether you agree with the publication of the RPF and as here, if you want to remove or not any figures from it prior to publication. Please note that the Authors checklist will be published at the end of the RPF.

EMBO Molecular Medicine has a "scooping protection" policy, whereby similar findings that are published by others during review or revision are not a criterion for rejection. Should you decide to submit a revised version, I do ask that you get in touch after 3-4 months if you have not completed it, to update us on the status.

I look forward to receiving your revised manuscript.

Yours sincerely,

Lise Roth

***** Reviewer's comments *****

Referee #1 (Comments on Novelty/Model System for Author):

In this work cell lines that are not representative for GBM are used, and I have recommended to verify some of their data in more reliable cell models.

Referee #1 (Remarks for Author):

In this study, "A fetal oncogene NUA2 is an emerging therapeutic target in glioblastoma", Jo et al show that the kinase NUA2 is expressed during embryonic brain development and is re-expressed in GBM. NUA2 deletion reduces, while overexpression increases, GBM cell proliferation, and this is supported in vivo. NUA2 also affects cell migration in vitro, and genes involved in extracellular matrix (ECM) and epithelial-to-mesenchymal transition (EMT) are dysregulated in NUA2-deficient cells. Pharmacological inhibition of NUA2 using HTH-02-006 reduces GBM cell growth and migration, and the authors highlight NUA2 as a promising therapeutic target.

Overall, the findings are important and interesting but some additional work is needed to fully support the conclusions made. The paper is generally well-written and easy to follow, however, sometimes critical information is missing, which impacts clarity. My primary concern is the reliance on traditional serum-grown cell lines, which poorly represent GBM biology. For instance, U251MG forms well-defined, non-invasive bulk tumors in orthotopic xenografts GBM (Lee et al. 2006; DOI 10.1016/j.ccr.2006.03.030), making it unsuitable for studying the mesenchymal (MES) phenotype. Mesenchymal transition should be confirmed using GBM cells grown in defined, serum-free media, where well-characterized proneural and mesenchymal-like cell lines are available.

Lastly, it is surprising that NUA2 appears to promote both proliferation and mesenchymal transition, as these processes are typically inversely correlated. MES-like GBM cells usually divide slower than PN-like (w high SOX2, Olig2, PDGFRA). Instead, MES tumors are usually more invasive. This apparent contradiction should be discussed in greater depth.

Below are my comments, both major and minor, figure by figure:

Figure 1

- Mention CGGA earlier, in connection to Fig 1C
- Kaplan-Meier curves, should be months, not days!

- Clarify that all grades of glioma are included in E-F. Since NUA2 is more highly expressed in GBM, the survival difference observed is not surprising when including all grades. Better to focus on GBM here and compare NUA2-high vs -low-expressing patients?

Figure 2

See my comments both above and below that confirmation in better cell model should be done.

E-G. Several proliferation assays (Ki67, MTT, colony forming), but the results are not fully congruent. The strong effect observed in the colony-forming assay should be reflected in others?

- Ki67 poor marker to show proliferation differences in vitro, would have been better to use BrdU or EdU
 - No effect on MTT assay in CR1 cells, the best knock-out (WB blank in 2C), but clear effect on colony formation. MTT assay measures metabolic activity in cells - are cells fewer but increase their metabolic activity? Are CR1 cells larger, senescence-like?
- H. Lower band on NUA2 blots - unspecific or the endogenous? Is the higher MW-band the introduced? Use arrow to indicate NUA2 band, explain lower fat band in U87.

U87MG seems to be the fastest proliferating cell line among the four (see sphere-formation in EV Fig5), but has the lowest NUA2 expression - how can NUA2 overexpression cause an increase in proliferation?

Figure 3

- Which CRISPR cells were used? (CR1-3?)
- For bioluminescence imaging, luciferase-expressing cells are required, but I can't find any information about introducing luciferase into the cells, only GFP/mCherry. Or is it really fluorescence that is measured? If so, as far as I know, detecting fluorescence through the skull can be quite challenging.
- Try to explain the big differences in luminescence intensity already after 7 days; do WT cells proliferate that fast or are they different already at injection? How come no tumor intensity increase is observed over the period in 3 out of 5 WT mice?
- Are there differences in growth and invasion pattern between tumors? Using HE staining only, non-proliferating invading (more mesenchymal?) cells are difficult to detect. Instead, I strongly suggest using staining of a human marker, such as STEM121. But again, U251MG is not a good model for mesenchymal GBM.

Figure 4

Very good that data from this immuno-competent model is included.

B. WB on OE tumors? In CR-tumors, 1 of 5 expressed NUA2 protein - can this be connected to shorter survival of the mouse?

Figure 5

- Gliovis portal reference
- EV datasets (DEGs and GO analyses) can't be presented as pdf files (journal's requirements?) - change to excel. They are now impossible to understand and evaluate.

A-B.

- I find the presentation of pooled DEG analyses (up+down by KO) somewhat confusing. Do I understand it correctly that no "direction" is observed here, that this analysis doesn't show whether ECM-related genes are enriched in either cell types (which is instead clarified in E)? Consider revising this for better clarity.
- Since decreased proliferation is a strong effect in NUA2-CR cells, where do cell cycle-related biological processes end up on this type of analysis? I would be surprised if the EMT- and MES-related WT/NUA2hi-cells showed enrichment of cell cycle-related genes.
- Clarify in body text that you compare NUA2low vs NUA2hi from TCGA (row 167, Fig. 5B)
- Reference for the EMT signature used for GSEA?

G-H. Transwell assays: Typo, row 186: "...whether cell migration is affected by U251 expression".

I-J. These figures are confusing and these cells are bad models for studying mesenchymal transition of GBM. Also, classical EMT markers are often not applicable to GBM cell states or subtypes. For instance, contrary to the conclusions made, the epithelial marker ZO-1 is low in CR-cells and high in OE-cells, and the two OE lines show very different results (E-to-N-cadherin + induction of EMT-related TFs only observed in LN229, whereas in U87MG N-cad goes down, no induction of Snail, Zeb, Slug). In this case, RNAseq analysis and GSEA using GBM cell state and MES transition genesets would be better to verify MES transition upon NUA2 overexpression.

Again, GBM subtype-characterized cell lines grown in serum-free media are available and would be more suitable for proving mesenchymal transition.

EV Fig. 2B, write NUA2-WT/-CR and NUA2-high/-low in Figure.

Figure 6 and EV Fig 4.

- How specific is HTH-02-006 at these pretty high effective concentrations? The NUA2low-expressing U87 cell lines show similar sensitivity as NUA2high lines, indicating unspecific inhibition. Use CR-cells as negative control?
- Can inhibition of NUA2 activity be confirmed by phospho-MYPT1 western blot?

C. scratch-wound healing assay analyses

- Impossible to see cells and wound edge in current pictures. Although blurry photographs, it looks like individual masks for each cell line's scratch-wound healing assay analyses are needed
- Isn't the reduction of wound closure solely due to decreased cell proliferation, not inhibition of active migration?
- Data from U87 can't be used, really bad cell attachment - redo using coating of plates or exclude. Should be less dependent on NUA2, because of its low baseline levels. Row 225: "...distinct growth patten, characterized by convergence and the formation of circular clusters" = sphere-forming, need to coat.

- Row 233: include viability

D.

- If NUA2 is critical for GBM cell proliferation: How can the NUA2high U251MG form small spheres, while the very

NUAK2low-expressing cell line U87MG form massive spheres?

- HTH-02-006 shows the biggest effect on sphere size in the NUAK2low-expressing cell lines line U87MG...
- The effect of this inhibitor should really be confirmed in better GBM cell models.

Referee #2 (Remarks for Author):

This study investigates NUAK2 as a potential therapeutic target for glioblastoma (GBM). Hanhee Jo and colleagues find that NUAK2 is a fetal oncogene highly expressed in embryonic brains and GBM tumors, but minimally expressed in healthy adult brains. Silencing NUAK2 in GBM cells suppressed proliferation and migration, while overexpression enhanced these processes. These results suggest that NUAK2 plays a critical role in GBM progression and is a promising therapeutic target for GBM treatment.

- The LOF and GOF in vitro and in vivo assays described in this study are suitable tools to validate the in-silico findings that NUAK2 over expression contribute to GBM growth and progression.
- The role of NUAK2 on ECM synthesis has been demonstrated based on RNA seq data and validated by Q-PCR in U251 WT and NUAK2-deleted cells. More data and functional assays should have been included to support this observation. The RNA seq data and western blots in Fig 5I-J showing the role of NUAK2 on EMT are not sufficient because not entirely convincing. Overall, all data, text and conclusion related to Figures 5 and EV2 need to be amended (see below more details). Major amendments need to be done, clarifications, more experiments, weak discussion.

The authors do not address the effect of the LoF or GOF on NUAK2 on the Hippo signalling pathway in GBM. LATS1/2 are direct targets of NUAK2, their levels and phosphorylation state should be examined and so does the nuclear to cytoplasmic ratio of endogenous YAP/TAZ be assessed.

A phosphoproteomic-enriched mass spec. analysis of the LoF and GOF on NUAK2 GBM cells should give a clearer view on what are the cellular consequences of the misexpression of this key kinase.

The contribution of NUAK1 is not adequately addressed. Specific primers, antibodies and maybe KO of the paralogous kinase, would inform whether there functional redundancy when NUAK2 levels are altered whether on LoF or GOF scenarios . For instance the effect of the drug HTH-02-006 on NUAK1 also needs to be tested for selectivity.

- Overall, the authors should use official nomenclature when they refer to gene and protein in human / mouse, e.g. italic when gene, transcripts
 - Fig1C & Text line 88-91: is it TCGA or CGGA data?
 - Fig 1H: indicate kDa scale for NUAK2 and GAPDH western blots. What does P7, P14 mean? Please specify in legend
 - Fig 2C&D: please indicate on the figure that the cells used were U251
 - Fig 2E: Does each dot in the graph represent the % calculated for 1 image? How many times this quantification was repeated? n=?
 - Line 858: sentence is not clear - you may need to delete 'and Nuak2 overexpression (N2OE)'
 - Fig2I: may check the scale bar and legend. Is it 50um?
 - Fig 2G & K: how the number of colonies are counted when they are touching to each other? Especially in U251WT, and U87N2OE, LN229N2OE. Should you also calculate the size (surface area) of the colonies? It is unclear what 'average of colonies per well' means - should it be number of colonies per well?
 - Fig3B: we don't see the blue bars, so you may label the graphs with WT and NUAK2CR on top of the graph, instead of using color code.
 - Fig3D: no scale bar. Can you delineate the tumor? As in 3C
 - More histology markers / other markers may be included to show increase/decrease of cell proliferation
 - Fig4A: you may add 'NUAK2' next to 'GOF and LOF'
 - Only Ki-67 staining demonstrate difference in cell proliferation. Would be interesting to know the size of the harvested tumors - tumor growth? Or to show more markers of cell proliferation.
 - Fig4E-G & EV1: how many sections/images were used to quantify?
- Data in Figure 5 and text are not convincing - need to be fully re-arranged, better described and suggestion/conclusion changed.
- Line 168-169: The analysis in Fig 5B suggests that ECM-related terms are significantly influenced by NUAK2 expression in TCGA GBM patient samples, not in 'our samples'. Please rectify the sentence.
 - Line 174 & Fig5C-D: please indicate in the text that 'CGGA' was also used. Correct the labeling of graph 5C: it should be TCGA instead of TGGA.
 - Lines 178 to 185, Fig 5E-F & EV2B: Some labels are missing in EV2B, and what does Verhaak2010 mean? Based on GSEA analysis, authors suggest that 'NUAK2 loss leads to reduced mesenchymal properties' because 'proneural signatures were enriched'. They also mentioned that 'ECM regulation and EMT are interdependent processes' because they were both positively enriched... May need more data/support to suggest this.
 - Line 185-191: 'mesenchymal signature is related to more migratory properties' - please include a reference. The writing is bad and does not correspond to figures 5G-H.

Fig 5G and 5H show cell counts (cell proliferation) and not cell migration...

Fig 5H: cells are U87 and LN229, and not U251 (as mentioned in text).

The effect of loss or over expression of NUAK2 on cell proliferation has been already shown in Fig2.

Western blots in Fig 5I & J are not well described in the text, and do not support 'the role of NUA2 in promoting EMT via ECM modulation'. N2OE in U87 and LN229 gave different results (E-CAD, N-CAD, SNAIL, SLUG, ...), but this is not discussed in the text.

- Title line 159 "NUAK2 mediates mesenchymal transition through ECM regulation" may be changed.

Figure 6:

Line 207-209 & Fig EV3E: As mentioned in the text 'NUAK1 mRNA levels were significantly lower in both low- and high-grade gliomas than in normal brain tissues'. Could we say that NUA2 expression may be correlated to NUA2 expression? While NUA2 increased in GBM & LGG tissues, NUA1 decreased significantly. This could be discussed.

Since NUA1 expression is relatively high and stable throughout brain development in both mice and humans, and in adult brain in humans, we cannot rule out that the significant decrease of NUA1 expression in GBM and LGG may have an impact on brain cancer.

Line 216-219: Please also indicate Fig EV4C.

Line 222: do not mention 'wound healing' - please indicate 'scratch assay' instead of 'scratch wound healing'.

Figures 6C, EV4D: Author may change the brightness of the images as we do not see the cells. No scratch highlight in EV4D, cells U87. So how quantification was done (right graph)?

line 224-225, author mentioned that 'only U87 cells have distinct growth pattern, characterized by convergence and the formation of circular clusters'. However, U87 cells seemed to form colonies as the other cell types (Fig 2K, EV4B). How do we explain that in scratch assay, they do not behave the same as the other cell types?

Fig 6D: weird results - why U251 spheroids do not grow over the course of 6 days without HTH-02-006 (0uM)? as shown for other cell types in Fig EV5A. In fig 6D & EV5B, do the graphs represent area at day 6? Please specify in figure and/or legend.

Other questions/discussion:

Besides Ki-67: Are there any other specific markers of proliferation for uncontrolled growth in cancers compared to tightly regulated proliferation during development?

Transcriptomic analysis of GBM cells and TCGA patient data revealed that NUA2 likely exerts its effects by modulating the ECM. EV2C & D may be shifted in main figure 5

As mentioned before, the data demonstrating the role of NUA2 in mediating EMT is weak, and it is not further discussed in the 'Discussion' part. The authors may only focus on ECM increase.

Line 287: authors do not investigate the effect of 'genetic' modulation of NUA2 but the change of its 'expression'. This should be reformulated.

Line 289: please include 'that': We observed 'that' the growth...

Line 293-303: although authors did not find correlation between expression levels of NUA1 and glioma types and survival rates in available datasets, they cannot conclude that NUA1 does not play a role. The significant decrease of NUA1 expression in tumors (GBM and LGG) compared to normal tissues must be discussed. Authors have not shown any in vitro data that confirm this in-silico finding. They did not quantify NUA1 level in their cell lines, or when NUA2 is overexpressed. NUA1 quantification ought to be added in the revised manuscript if NUA1 is included in FigEV3

Does NUA2 inhibition by HTH-02-006 decrease NUA2 expression? Increase NUA1? The authors should show by Q-PCR and western.

**** Reviewer's comments ****

Referee #1 (Comments on Novelty/Model System for Author):

In this work cell lines that are not representative for GBM are used, and I have recommended to verify some of their data in more reliable cell models.

Referee #1 (Remarks for Author):

In this study, "A fetal oncogene NUA2 is an emerging therapeutic target in glioblastoma", Jo et al show that the kinase NUA2 is expressed during embryonic brain development and is re-expressed in GBM. NUA2 deletion reduces, while overexpression increases, GBM cell proliferation, and this is supported *in vivo*. NUA2 also affects cell migration *in vitro*, and genes involved in extracellular matrix (ECM) and epithelial-to-mesenchymal transition (EMT) are dysregulated in NUA2-deficient cells. Pharmacological inhibition of NUA2 using HTH-02-006 reduces GBM cell growth and migration, and the authors highlight NUA2 as a promising therapeutic target.

Overall, the findings are important and interesting but some additional work is needed to fully support the conclusions made. The paper is generally well-written and easy to follow, however, sometimes critical information is missing, which impacts clarity.

My primary concern is the reliance on traditional serum-grown cell lines, which poorly represent GBM biology. For instance, U251MG forms well-defined, non-invasive bulk tumors in orthotopic xenografts GBM (Lee et al. 2006; DOI 10.1016/j.ccr.2006.03.030), making it unsuitable for studying the mesenchymal (MES) phenotype. Mesenchymal transition should be confirmed using GBM cells grown in defined, serum-free media, where well-characterized proneural and mesenchymal-like cell lines are available.

We thank the reviewer for this thoughtful comment. We fully acknowledge that no glioma cell line perfectly recapitulates all aspects of GBM biology. To mitigate this limitation, we performed our experiments using four widely accepted glioma cell lines—U87, U251, LN229, and LN319—which exhibit distinct molecular, genetic, and physiological characteristics (PMID: 30643153, PMCID: PMC9347152, PMID: 34466804, PMID: 3800350). Notably, U87 cells are associated with a proneural-like expression signature (PMID: 39237744), while U251 cells have been reported to exhibit mesenchymal-like morphology and expression patterns (PMID: 26517510; PMID: 28454322).

Although the majority of our studies were conducted under serum-containing conditions, our 3D spheroid cultures were grown in serum-free media and yielded results consistent with our 2D culture findings. To further strengthen our conclusions beyond *in vitro* models, we performed *in vivo* loss-of-function (LOF) studies using an immune-competent *in utero* electroporation (IUE) model of glioma. These studies supported our *in vitro* data, showing that NUA2 deletion reduced tumor burden and improved survival (Fig. 4). Similarly, in orthotopic U251 xenograft models, NUA2-deficient cells formed smaller tumors and extended survival (Fig. 3). Collectively, these data across 2D, 3D, and *in vivo* models consistently support a role for NUA2 in promoting glioma growth.

We focused on U251 for transcriptomic profiling and xenograft studies due to its robust and reproducible responses across assays. Transcriptomic data revealed enrichment of ECM-related genes following NUA2 deletion. While our initial submission linked these ECM changes to epithelial–mesenchymal transition (EMT)—supported by GSEA results and consistent with studies in other cancers where NUA2 regulates EMT and ECM pathways (PMID: 34558636, PMID: 30446657, PMID: 32755597, PMID: 30622137, PMID: 21460252)—we did not directly test EMT markers in our system. In response to this reviewer's

comment and Reviewer 2's suggestion (comment #37), we have removed all EMT-related data and discussion in the revised manuscript and now focus solely on ECM-related findings.

Regarding the use of serum-free GBM models, we agree that glioma stem-like cells (GSCs) are valuable for modeling mesenchymal and proneural phenotypes. However, GSCs often exhibit patient-specific variability in gene expression, including NUA2. Through collaborations, we screened seven GSC lines and found that all expressed little or no NUA2, which limited our ability to conduct LOF studies. Nonetheless, to address this reviewer's concern, and after consultation with the editor, we performed gain-of-function (GOF) studies by overexpressing NUA2 in GSC11 and GSC23 cells—two well-characterized GSC lines used to study ECM remodeling and migratory behavior (PMID: 37461511). NUA2 overexpression significantly increased proliferation and migration in both GSC lines (Fig. 3G–J), consistent with our findings in traditional glioma cell lines. These results further support a role for NUA2 in promoting GBM progression.

Lastly, it is surprising that NUA2 appears to promote both proliferation and mesenchymal transition, as these processes are typically inversely correlated. MES-like GBM cells usually divide slower than PN-like (w high SOX2, Olig2, PDGFRA). Instead, MES tumors are usually more invasive. This apparent contradiction should be discussed in greater depth.

We thank the reviewer for this insightful comment. We agree that mesenchymal-like (MES) GBM cells are often less proliferative compared to proneural (PN)-like cells, which typically express higher levels of SOX2, OLIG2, and PDGFRA. However, our findings—as well as emerging evidence from other cancers—suggest that NUA2 can promote both proliferation and mesenchymal-like features depending on cellular context. We have added the following paragraph to the Discussion section to address this apparent contradiction in greater depth:

“Our data show that NUA2 promotes both proliferation and migration in glioma cells. Transcriptomic profiling of U251-NUA2-CR cells revealed enrichment of ECM-associated genes and dysregulation of some EMT-related markers, suggesting that NUA2 may contribute to migration properties in addition to driving proliferation. Similar dual roles for NUA2 have been reported in other cancers. In cervical cancer, NUA2 knockdown reduced cell proliferation, migration, and expression of EMT markers such as N-cadherin, Snail, and ZEB1—mediated via interaction with CYFIP2, a known EMT regulator (PMID: 34558636). In liver cancer, NUA2 silencing or pharmacological inhibition suppressed proliferation through modulation of YAP signaling (PMID: 30446657), and YAP-driven NUA2 was shown to reinforce YAP activity via actomyosin cytoskeleton remodeling, a process linked to both migration and mesenchymal phenotypes (PMID: 24556840).

Dual regulation of proliferation and mesenchymal traits is not uncommon in cancer and often reflects context-dependent activation of intersecting pathways. Pathways such as TGF- β , WNT, EGFR, and Hippo can simultaneously regulate cell cycle progression and EMT by engaging distinct downstream effectors (e.g., Snail, ZEB1, Twist). NUA2 has been implicated in both Hippo and TGF- β signaling (PMID: 38510132; PMID: 34282782; PMID: 30622137), yet the crosstalk between these pathways and how they intersect with NUA2 kinase activity remains poorly defined. Currently, only two direct substrates of NUA2 kinase—MYPT1 and LATS1/2—have been described (PMID: 18023418; PMID: 24171924 PMID: 30158528), but it is likely that additional targets exist that could mediate NUA2's effects on both proliferation and migration. Further studies are needed to map the broader substrate repertoire of NUA2 and elucidate how its signaling integrates proliferative and migratory programs in glioma.”

Below are my comments, both major and minor, figure by figure:

Figure 1

1. Mention CGGA earlier, in connection to Fig 1C

We thank the reviewer for this suggestion. We have updated the text to mention CGGA and define its full designation in connection to Fig1C.

2. Kaplan-Meier curves, should be months, not days!

Thank you for the careful reading of the document. We have corrected this typo.

3. Clarify that all grades of glioma are included in E-F. Since NUA2 is more highly expressed in GBM, the survival difference observed is not surprising when including all grades. Better to focus on GBM here and compare NUA2-high vs -low-expressing patients?

To improve clarity of the TCGA and CGGA data we have included additional graphs (Fig 4F) that separate survival by glioma grade. While the differences in survival are more pronounced in lower grades of glioma, there is a statistically significant difference in the survival of GBM patients with high vs low expression of Nuak2.

Figure 2

4. See my comments both above and below that confirmation in better cell model should be done.

Please see our response to GSC experiments above. As discussed with the Editor, we performed experiments in GSC that complement those in glioma cell lines found in Fig 2. These new GSC studies are now incorporated into the updated Fig 3.

5. E-G. Several proliferation assays (Ki67, MTT, colony forming), but the results are not fully congruent. The strong effect observed in the colony-forming assay should be reflected in others?

We thank the reviewer for this important observation. While all the assays performed assess cell proliferation, they measure different aspects over varying time frames and under different conditions, which can explain differences in the magnitude of the observed effects. Ki67 and EdU are short-term markers of proliferation: Ki67 provides a snapshot of the proportion of cells in the active phases of the cell cycle, while EdU incorporates into DNA during a defined 4-hour window of S-phase activity in our assay. MTT, on the other hand, measures metabolic activity over a 5-day period, serving as an indirect proxy for cell number and viability. The colony formation assay (CFA) reflects long-term proliferative capacity and survival over 10–14 days, capturing cumulative effects of both proliferation and cell survival under anchorage-dependent conditions.

We observed statistically significant reductions in all assays following NUA2 deletion, indicating a consistent negative effect on cell proliferation. The more pronounced difference in the CFA likely reflects the compounding nature of long-term proliferation defects. Even modest short-term differences in proliferation or survival, as detected by Ki67, EdU, or MTT, can result in substantial long-term deficits in colony-forming ability. Therefore, the strong CFA phenotype is not inconsistent with the other assays, but rather highlights the extended impact of NUA2 loss over time. Together, these data support the conclusion that NUA2 is a key regulator of glioma cell proliferation, with effects that manifest across both short-term and long-term functional assays.

6. Ki67 poor marker to show proliferation differences in vitro, would have been better to use BrdU or EdU

We appreciate the reviewer's comment regarding the limitations of Ki67 as a proliferation marker *in vitro*. We initially included Ki67 analysis for two main reasons: (1) to maintain consistency between our *in vitro* and *in vivo* experiments, and (2) because Ki67 is a widely used marker of proliferation in cancer biology. However, we acknowledge that Ki67 does not always capture subtle differences in proliferation under *in vitro* conditions. To address this concern, we performed EdU incorporation assays in cultured U251 cells (Fig. 2F), which provide a more direct and time-resolved measurement of DNA synthesis. These results showed a significant reduction in EdU incorporation in U251-NUA2-CR cells compared to controls, corroborating our Ki67 findings. Additionally, to further support our *in vivo* conclusions, we performed PCNA immunostaining on orthotopic and IUE-derived tumors (Figs. 4 and 5), which reinforced the observed proliferation defects upon

NUAK2 deletion. Together, these complementary analyses strengthen the conclusion that NUAK2 promotes glioma cell proliferation in both in vitro and in vivo contexts.

7. No effect on MTT assay in CR1 cells, the best knock-out (WB blank in 2C), but clear effect on colony formation. MTT assay measures metabolic activity in cells - are cells fewer but increase their metabolic activity? Are CR1 cells larger, senescence-like?

We thank the reviewer for this insightful comment. We analyzed the morphology of wild-type U251 cells and compared them to our CR1 and CR3 knockout lines, the latter being used throughout our study. Quantitative assessment of cell area and circularity revealed no significant differences in overall morphology among the three lines (Reviewer Fig 1A-C). To further explore the potential role of senescence, we examined p21 expression as a marker using both Western blotting and immunocytochemistry. In both assays, we observed no increase, as would be expected, in p21 levels, indicating that neither CR1 nor CR3 exhibit increased senescence (Reviewer Fig 1D-I). Therefore, differences observed between the MTT and colony formation assays most likely are not attributed to senescence-related effects.

Reviewer Fig1. (A-C) Brightfield image of U251 WT and CRISPR-deleted cells. Scale bar =100µm. (D) Representative western blot of p21 in U251 WT and CR cell lines. β-actin was used as loading control. (E-J) Immuno-fluorescence of p21(red) and DAPI (blue) staining in U251 WT and CR cell lines. (I) Quantification of p21+ cells normalized to number of total DAPI+ cells. N=6.

8. H. Lower band on NUAK2 blots - unspecific or the endogenous? Is the higher MW-band the introduced? Use arrow to indicate NUAK2 band, explain lower fat band in U87.

We thank the reviewer for this important point. Various studies have reported non-specific bands when using NUAK2 antibodies across different species and cell types (e.g., PMID: 34818445, 30158528, 34558636, 30446657), and some variability between antibody lots and sources has also been noted. In our western blots, the upper band (~75 kDa) corresponds to the predicted molecular weight of NUAK2 and aligns with expression profiles reported in the Human Protein Atlas. This band is reduced or eliminated upon CRISPR-mediated NUAK2 deletion and is strongly enhanced in overexpression experiments (Fig. 2H), supporting its identification as the specific NUAK2 signal. The lower band (~70 kDa) appears to be non-specific, as it persists in NUAK2-CR cells and does not correlate with known NUAK2 expression. However, we cannot fully rule out the possibility that some bands—particularly those appearing as “fat” or diffuse—may represent post-translationally modified forms of NUAK2, especially in cells like U87 where expression is relatively low but modification may alter mobility. To aid interpretation, we have updated the western blot figures to clearly indicate the expected NUAK2 band with an arrow or tick mark and have included molecular weight markers

on each panel.

9. U87MG seems to be the fastest proliferating cell line among the four (see sphere-formation in EV Fig5), but has the lowest NUA2 expression - how can NUA2 overexpression cause an increase in proliferation?

Tumor cell proliferation is a complicated, multifactorial process and cancer cells with distinct genetic backgrounds may rely on different mechanisms to circumvent cell death pathways and increase proliferation and survival. The four cell lines used in this study are genotypically and phenotypically distinct, which likely account for differences in responses to NUA2 overexpression. For instance, U87 is *p53* wildtype whereas U251, LN319, and LN229 harbor *p53* mutations (PMID: 32704022). In addition, previous studies have shown that these cell lines have different gene expression programs and metabolic characteristics (PMID: 26517510 PMID: 27651343; PMID: 30643153; PMID: 35922758). It is likely that even in a highly proliferative cell line like U87, overexpression of NUA2 kinase can activate several pathways that enhance proliferation by accelerating cell cycle progression, increase growth factor signaling or even enhancing metabolic support for proliferation. In other cancers, NUA2 has been implicated in glucose metabolism, the stress response, and cytoskeletal regulation (PMID: 39237744; PMID: 15799806 PMC3834304). It has also been shown to promote glucose uptake, activate glycolysis and enhance mTORC1 signaling (PMID: 39270807; PMID: 39237744), all of which could provide additional metabolic and proliferative advantages—even in a fast-growing line like U87. How NUA2 interfaces with oncogenic signaling networks across different genetic contexts remains an important area for future investigation.

Figure 3

10. Which CRISPR cells were used? (CR1-3?)

CRISPR 3 was used for the transplant studies. We have clarified this in the updated figure legend.

11. For bioluminescence imaging, luciferase-expressing cells are required, but I can't find any information about introducing luciferase into the cells, only GFP/mCherry. Or is it really fluorescence that is measured? If so, as far as I know, detecting fluorescence through the skull can be quite challenging.

Thank you for pointing out that this information was inadvertently excluded from the original submission. We have updated the methods to reflect the inclusion of a luciferase expression plasmid in both the orthotopic transplants and the IUE model.

12. Try to explain the big differences in luminescence intensity already after 7 days; do WT cells proliferate that fast or are they different already at injection? How come no tumor intensity increase is observed over the period in 3 out of 5 WT mice?

Our *in vitro* data demonstrates the U251-NUA2CR cells do indeed grow slower than control cells. The U251 cells were modified and expanded *in vitro* prior to injection, therefore they are already different before they are injected. To generate tumors, we used the protocol from Schulz et al (PMID: 35922758) which yielded observable luminescence (via bioluminescence imaging) as early as 7 days. The reason that the luminescence was different at day 7 is that the tumors failed to grow or grew very slowly compared to controls at that time, therefore the luminescent signal was minimal to undetectable.

In reference to the steady expression of luciferase in some of the control tumor samples, there are several possible reasons that this could have occurred. One explanation is the inherent variability in the luminescence process. While luminescence imaging is a convenient method to measure *in vivo* growth, it has limitations such as tissue attenuation or light scattering, heterogeneous expression of luciferase, tumor depth, or luciferin availability after injection before imaging. While we strived to mitigate any of these issues from occurring in our studies, we cannot rule out that these may have occurred in some mice. Therefore, we cannot rely on this

as the sole measure of tumor growth. To address this, we also examined the IUE tumor tissue at different time points using histology and included analysis of proliferation and other tumor markers (Fig 6 and Fig EV1). For the orthotopic transplants we initially included analysis at day 28. We now include histology and survival curve analysis at terminal time points (Fig 4C and E). Importantly, all WT U251 tumor-bearing mice (including the 3 of 5 mentioned) generated tumors in our studies. Similarly, for IUE studies we analyzed several time points in the original submission to provide a more holistic view of tumor progression.

13. Are there differences in growth and invasion pattern between tumors? Using HE staining only, non-proliferating invading (more mesenchymal?) cells are difficult to detect. Instead, I strongly suggest using staining of a human marker, such as STEM121. But again, U251MG is not a good model for mesenchymal GBM.

We appreciate the reviewer's comment regarding tumor invasion patterns and the suggestion to use human-specific markers such as STEM121. As noted, U251 cells are not considered a representative model for studying mesenchymal GBM or tumor invasion, but can be used to study proliferation. Prior studies, including Shultz et al. (2022), have shown that U251 and U87 cells form compact, non-invasive tumors when transplanted into

immunocompromised mice, which limits their utility for assessing infiltration or mesenchymal behavior *in vivo*. In our orthotopic transplantation experiments, U251 control cells formed well-defined tumors within 7 days post-implantation (Fig. 6), consistent with prior reports. In contrast, U251-NUAK2-CR cells failed to form visible tumors at the same time point. To verify this, we performed STEM121 staining, which showed robust human cell labeling in control tumors but minimal signal in NUAK2-CR-transplanted brains (Reviewer Fig. 2). These results support the conclusion that NUAK2 deletion substantially impairs tumor formation in this model.

Reviewer Fig 2. Immunofluorescence staining of STEM121 (human) and DAPI (nuclei) in control and U251 orthotopic transplants. Scale bar =50µm.

We have also added a Kaplan–Meier survival curve (Fig. 4C), demonstrating that mice implanted with U251-WT cells succumbed to tumor burden over time, whereas mice implanted with U251-NUAK2-CR cells had significantly prolonged survival. These data further support a role for NUAK2 in promoting glioma growth *in vivo*. Given the limitations of the U251 model for studying invasion and mesenchymal transition, we confirmed key findings using an *in utero* electroporation (IUE) model of malignant glioma, which better recapitulates GBM histopathology. In parallel, we used *in vitro* transwell migration assays to assess the effects of NUAK2 on glioma cell motility under defined conditions (Fig. 6G–H). Together, these complementary models provide robust evidence for the role of NUAK2 in promoting glioma cell proliferation and migration.

Figure 4

15. Very good that data from this immuno-competent model is included.

We thank the reviewer for acknowledging the rigor of our study.

16. WB on OE tumors? In CR-tumors, 1of 5 expressed NUAK2 protein - can this be connected to shorter survival of the mouse?

The mice used for western blot analysis were from a separate cohort than those used for survival and histological studies. We collected these animals independently to allocate tissue specifically for biochemical and RNA-based analyses. As a result, NUA2 expression was assessed at various timepoints by immunohistochemistry in a distinct set of animals (Fig. 5E, Fig. EV1). Regarding variability in NUA2 expression across CRISPR-targeted mice, this is a known limitation of the *in utero* electroporation (IUE) technique, which—while generally efficient—can vary between animals. To account for this, we analyzed a larger number of animals per IUE experiment. It is also possible that animals with less efficient NUA2 deletion developed more aggressive tumors and died earlier than those with more efficient deletion, potentially affecting survival outcomes. However, we were unable to directly test this, as several brains from the survival cohort were not analyzable due to post-mortem tissue degradation.

Figure 5

17.B. Gliovis portal reference

We have added this information below and included the citation in the manuscript.

Robert L. Bowman, Qianghu Wang, Angel Carro, Roel G.W. Verhaak, Massimo Squatrito, Gliovis data portal for visualization and analysis of brain tumor expression datasets, *Neuro-Oncology*, Volume 19, Issue 1, 1 January 2017, Pages 139–141, <https://doi.org/10.1093/neuonc/now247>.

18. EV datasets (DEGs and GO analyses) can't be presented as pdf files (journal's requirements?) - change to excel. They are now impossible to understand and evaluate.

We apologize that this information was not accessible to the reviewer. We uploaded the original files to the submission system as Excel files, but they were converted to PDFs by the system. The resubmission process will allow for the files to be submitted and maintained as Excel files, therefore the files should now be represented in a more reader friendly manner.

19. A-B. I find the presentation of pooled DEG analyses (up+down by KO) somewhat confusing. Do I understand it correctly that no "direction" is observed here, that this analysis doesn't show whether ECM-related genes are enriched in either cell types (which is instead clarified in E)? Consider revising this for better clarity.

We regret that original figure 5 (now figure 6) and the accompanying text were unclear. We have significantly revised the figure and Fig EV2. In the update manuscript we performed GO enrichment analysis on DEGs and represent them as either upregulated or downregulated. GO enrichment analysis of the DEGs showed enrichment of the extracellular matrix organization term in both the upregulated and downregulated DEG sets, leading us to perform a GSEA analysis to clarify the overall directionality of all genes in the extracellular matrix organization term (Fig 6A and 6C). The GSEA revealed a significant upregulation of ECM genes in both the U251 and TCGA GBM data when compared to NUA2 CR and NUA2-low respectively (Fig 6C, D).

20. Since decreased proliferation is a strong effect in NUA2-CR cells, where do cell cycle-related biological processes end up on this type of analysis? I would be surprised if the EMT- and MES-related WT/NUA2hi-cells showed enrichment of cell cycle-related genes.

GSEA analysis of NUA2-CR DEGs revealed modest enrichment of genes associated with cell proliferation (Reviewer Fig. 3), although the total number of differentially expressed genes was limited. To further investigate changes in cell cycle regulation, we performed immunoblotting using a commercially available panel of cell cycle proteins. This analysis showed decreased expression of CDK2 and Cyclin D3 in NUA2-CR cells, while levels of Cyclin D1 and CDK6 remained unchanged. The reduction in CDK2 and Cyclin D3 aligns with the observed decrease in proliferation and supports the conclusion that NUA2 contributes to cell

cycle progression. While the exact mechanisms linking NUAK2 to specific cell cycle regulators remain unclear, our data suggest a potential regulatory role that warrants further investigation in future studies.

Reviewer Fig 3. (A) GSEA analysis of cell proliferation associated U251-N2CR DEGs. (B) Western blots from cell cycle antibody sampler. Included antibodies for CyclinD1, CDK6, CDK2, CyclinD3. β -actin was used as loading control.

21. Clarify in body text that you compare NUAK2^{low} vs NUAK2^{hi} from TCGA (row 167, Fig. 5B)

We have made this edit in the revised manuscript.

22. Reference for the EMT signature used for GSEA?

We have updated the main text to include the reference for all signatures used in the study and placed the information here for your review.

MSigDb: (EMT)

A. Subramanian, P. Tamayo, V.K. Mootha, S. Mukherjee, B.L. Ebert, M.A. Gillette, A. Paulovich, S.L. Pomeroy, T.R. Golub, E.S. Lander, & J.P. Mesirov, Gene set enrichment analysis: A knowledge-based approach for interpreting genome-wide expression profiles, *Proc. Natl. Acad. Sci. U.S.A.* 102 (43) 15545-15550, <https://doi.org/10.1073/pnas.0506580102> (2005).

Gene Ontology: (ECM)

Ashburner et al. Gene ontology: tool for the unification of biology. *Nat Genet.* 2000 May;25(1):25-9. DOI: [10.1038/75556](https://doi.org/10.1038/75556)

The Gene Ontology Consortium. The Gene Ontology knowledgebase in 2023. *Genetics.* 2023 May 4;224(1):iyad031. DOI: [10.1093/genetics/iyad031](https://doi.org/10.1093/genetics/iyad031)

KEGG: (Hippo Pathway)

Kanehisa, M., Furumichi, M., Sato, Y., Matsuura, Y. and Ishiguro-Watanabe, M.; KEGG: biological systems database as a model of the real world. *Nucleic Acids Res.* 53, D672-D677 (2025).

Kanehisa, M. and Goto, S.; KEGG: Kyoto Encyclopedia of Genes and Genomes. *Nucleic Acids Res.* 28, 27-30 (2000).

23. G-H. Transwell assays: Typo, row 186: "...whether cell migration is affected by U251 expression".

Thank you for the careful reading of the document. We have corrected this typo in the text.

24. I-J. These figures are confusing and these cells are bad models for studying mesenchymal transition of GBM. Also, classical EMT markers are often not applicable to GBM cell states or subtypes. For instance, contrary to the conclusions made, the epithelial marker ZO-1 is low in CR-cells and high in OE-cells, and the two OE lines show very different results (E-to-N-cadherin + induction of EMT-related TFs only observed in LN229, whereas in U87MG N-cad goes down, no induction of Snail, Zeb, Slug). In this case, RNAseq analysis and GSEA using GBM cell state and MES transition genesets would be better to verify MES transition upon NUA2 overexpression.

We regret that the figure and main text were confusing and have heavily edited the text and figure. As mentioned previously, we have removed references to EMT and MES for the manuscript and instead focus on ECM associated terms.

25. Again, GBM subtype-characterized cell lines grown in serum-free media are available and would be more suitable for proving mesenchymal transition.

Please see responses to previous mention of this issue.

26. EV Fig. 2B, write NUA2-WT/CR and NUA2-high/low in Figure.

We have made this adjustment in the figure.

Figure 6 and EV Fig 4.

27. How specific is HTH-02-006 at these pretty high effective concentrations? The NUA2low-expressing U87 cell lines show similar sensitivity as NUA2high lines, indicating unspecific inhibition.

We appreciate the reviewer's comment regarding the specificity of HTH-02-006 at higher concentrations. HTH-02-006 is a semi-selective NUA2 inhibitor derived from WZ4003, which targets both NUA1 and NUA2, and was designed to enhance selectivity toward NUA2 (PMID: 24171924). While HTH-02-006 exhibits greater specificity for NUA2 at lower concentrations, it may inhibit NUA1 or other kinases at higher doses. Previous studies have demonstrated the effective use of HTH-02-006 to target NUA2 in liver and prostate cancer models (PMID: 20411723; PMID: 34818445). These studies report an IC₅₀ of approximately 125 nM and provide mechanistic validation: for example, Yuan et al. showed that HTH-02-006 directly targets a conserved A236 residue in NUA2, with the A236T mutation significantly reducing inhibitor efficacy—strongly suggesting on-target activity.

In our study, we used concentrations within the 1–10 μM range, consistent with prior literature (PMID: 20411723; PMID: 34818445; PMID: 30158528), and observed significant inhibition of glioma cell proliferation and migration. To further assess specificity, we evaluated NUA1 and NUA2 expression following HTH-02-006 treatment and found no significant changes in expression (Fig. EV4B). Additionally, we treated both U251-WT and U251-NUA2-CR cells with HTH-02-006 and measured spheroid formation (Fig. EV4C). The U251-WT cells exhibited a clear dose-dependent response, whereas the NUA2-CR cells showed no response, supporting that the inhibitor's effects are likely mediated through NUA2. That said, we acknowledge that at higher concentrations, off-target effects cannot be entirely ruled out. However, our data—combined with previous validation studies—support that NUA2 is the primary target of HTH-02-006 under our experimental conditions.

28. Can inhibition of NUA2 activity be confirmed by phospho-MYPT1 western blot?

The inhibition of kinase activity by previous iterations of this inhibitor WZ4003 and the HTH-002-006 itself have been demonstrated by various publications (PMID: 20411723; PMID: 34818445; PMID: 30158528). NUA2 phosphorylates MYPT1 at serine 445 (S445) and can be used to verify that the inhibitor is targeting NUA2 activity. While there are several phosph-MYTP1 antibodies, S445 is not a commercially available

antibody and must be obtained from a consortium in Great Britain. To acquire the antibody, we needed to obtain an import permit from the USDA. Given the current political climate obtaining this permit was not feasible in the revision timeline, not to mention the associated cost. Therefore, we were unable to obtain phosphor-MYPT1 S445 antibody. However, we were able to optimize the phos-tag technique in the lab, which separates protein based on their phosphorylation state, allowing for the detection of both unphosphorylated and phosphorylated versions of a protein. Using this method, we confirmed that treatment of HTH-02-006 lead to overall less phosphorylation and overall degradation of MYPT1 (Fig EV4A).

C. scratch-wound healing assay analyses

29. Impossible to see cells and wound edge in current pictures. Although blurry photographs, it looks like individual masks for each cell line's scratch-wound healing assay analyses are needed

Thank you for bringing this issue to our attention. We have increased the contrast and brightened the images in the revised manuscript.

30. Isn't the reduction of wound closure solely due to decreased cell proliferation, not inhibition of active migration?

It is possible that the reduction in wound closure is in part due to a decrease in cell proliferation. We concluded that the differences observed in wound closure are likely due in part to migration reductions for the following two reasons: 1) the MTT assay does not show significant differences in proliferation at early time points (before 48 hrs) however, significant differences in wound closure are observed as early as 24hrs; and 2) we performed transwell migration assays on HTH-02-006 treated U251 cells and found reduced migration (Fig 7D). Together, these data indicate that the reduction in wound closure is at least in part due to impaired migration.

31. Data from U87 can't be used, really bad cell attachment - redo using coating of plates or exclude. Should be less dependent on NUA2, because of its low baseline levels. Row 225: "...distinct growth patten, characterized by convergence and the formation of circular clusters" = sphere-forming, need to coat.

We regret that these assays were not clear. We optimized the assay for this cell line and have repeated the U87 experiments to improve cell attachment. Consistent with previous results, wound closure was impaired in this cell line (Fig EV4F).

32. Row 233: include viability

We have included this in the updated manuscript.

33. D. If NUA2 is critical for GBM cell proliferation: How can the NUA2high U251MG form small spheres, while the very NUA2low-expressing cell line U87MG form massive spheres? HTH-02-006 shows the biggest effect on sphere size in the NUA2low-expressing cell lines line U87MG...

The growth rates that we observed are consistent with those in prior studies using these cell lines (PMID: 30643153; PMID: 26517510; PMID: 27651343). There are some inherent genetic and phenotypic differences between the cell lines that likely account for the difference in neurosphere-forming capacity between U87 and U251 cells. First, as we mentioned previously, U87 cells are *p53* wildtype while U251, LN319, and LN229 cell lines are *p53* mutant. In addition, U87 carry mutations in several other cell cycle regulators. Second, these cell lines have been shown to have distinct molecular profiles: U87 cells have been classified as having a more proneural-like signature (PMID: 39237744); while, U251 cells have been attributed a more

mesenchymal-like morphology and gene expression profile (PMID: 26517510), which are reflected in differences in protein expression (PMID: 39640533, PMID: 27651343, PMID: 39473408). Together, these studies demonstrate that there are differences between the cell lines that reflect their distinct molecular profiles and signaling activity, both of which could influence proliferation outside of NUA2 expression or be influenced by NUA2 expression in distinct ways. Phenotypically we and others observed differences in proliferation and neurosphere formation. For instance, U87 cells have a faster initial cell aggregation, but have reduced sustainability; while U251 cells are slower to form neurospheres, but are stable across multiple passages (PMC3834304, PMID: 31704823, PMID: 24260059, PMID: 26517510). This is thought to be at least in part due to the metabolic flexibility of U251 cells (PMID: 24260059, PMID: 31704823, PMID: 24260059). It is possible that U87 cells have adapted to circumvent cell cycle checkpoints differently than U251 cells, which are not dependent on NUA2 activity.

34. The effect of this inhibitor should really be confirmed in better GBM cell models.

We appreciate that the reviewer requested that the experiments be repeated in a different serum-free GSC line. However, in all of the GSC cell line that we tested, there was little to no NUA2 expression, therefore testing the inhibitor in these cell lines would not address whether inhibition of NUA2 activity regulates proliferation.

Referee #2 (Remarks for Author):

This study investigates NUA2 as a potential therapeutic target for glioblastoma (GBM). Hanhee Jo and colleagues find that NUA2 is a fetal oncogene highly expressed in embryonic brains and GBM tumors, but minimally expressed in healthy adult brains. Silencing NUA2 in GBM cells suppressed proliferation and migration, while overexpression enhanced these processes. These results suggest that NUA2 plays a critical role in GBM progression and is a promising therapeutic target for GBM treatment.

- The LOF and GOF in vitro and in vivo assays described in this study are suitable tools to validate the in-silico findings that NUA2 over expression contribute to GBM growth and progression.
- The role of NUA2 on ECM synthesis has been demonstrated based on RNA seq data and validated by Q-PCR in U251 WT and NUA2-deleted cells. More data and functional assays should have been included to support this observation. The RNA seq data and western blots in Fig 5I-J showing the role of NUA2 on EMT are not sufficient because not entirely convincing. Overall, all data, text and conclusion related to Figures 5 and EV2 need to be amended (see below more details).

Major amendments need to be done, clarifications, more experiments, weak discussion.

1. The authors do not address the effect of the LoF or GOF on NUA2 on the Hippo signalling pathway in GBM. LATS1/2 are direct targets of NUA2, their levels and phosphorylation state should be examined and so does the nuclear to cytoplasmic ratio of endogenous YAP/TAZ be assessed. A phosphoproteomic-enriched mass spec. analysis of the LoF and GOF on NUA2 GBM cells should give a clearer view on what are the cellular consequences of the misexpression of this key kinase.

We thank the reviewer for this insight. Our RNA-seq data revealed that Hippo-associated genes were dysregulated in the absence of NUA2 and GSEA analysis showed a positive enrichment of Hippo signaling components in U251 WT compared to U251-N2-CR DEGs (Fig EV2B-C). Therefore, we analyzed Hippo signaling pathway in our cell lines. We performed phos-tag immunoblotting to evaluate the phosphorylation state of YAP and TAZ in NUA2 GOF and LOF conditions. We found that when compared to controls, U251-N2-CR cells showed an increase in YAP and TAZ phosphorylation (Fig EV2D). In contrast, NUA2 overexpression lead to a decrease in YAP and TAZ phosphorylation (Fig EV2D). Consistent with these

findings, NUAK2 overexpression lead to the nuclear accumulation of YAP/TAZ. Together, the data suggests NUAK2 modulates Hippo signaling in GBM cells.

2. The contribution of NUAK1 is not adequately addressed. Specific primers, antibodies and maybe KO of the paralogous kinase, would inform whether there functional redundancy when NUAK2 levels are altered whether on LoF or GOF scenarios . For instance the effect of the drug HTH-02-006 on NUAK1 also needs to be tested for selectivity.

We thank the reviewer for this comment and the opportunity to address this concern. We assessed the relationship between NUAK1 and NUAK2 in several ways. First, we used immunoblotting to measure the levels of NUAK1 in the four glioma cell lines, finding low to no expression of NUAK1 in these cells. (Fig EV3K). Next, we examined the NUAK1 levels in U251-N2CR deleted cells finding no significant change in NUAK1 expression in the absence of NUAK2. Notably, N2CR#3 the cell line we used for studies throughout the paper had no change in NUAK1 expression (Fig EV3K). We also measured NUAK1 levels in NUAK2 GOF context and found little to no change compared to controls (Fig EV3K). In regard to patient samples, we performed Spearman's correlation analysis using the GlioVis portal to measure the correlation between NUAK1 and NUAK2 expression in GBM. This analysis showed no significant correlation between NUAK1 and NUAK2 expression, suggesting that they are not regulating each other's expression in a significant way. Lastly, we have added a section to the discussion section that addresses the potential role of NUAK1 in glioma.

Please see Reviewer 1# 27 for our response to HTH-02-006 selectivity.

3. Overall, the authors should use official nomenclature when they refer to gene and protein in human / mouse, e.g. *italic when gene, transcripts*

Thank you for catching this error. We have gone through the manuscript and made the appropriate changes.

4. Fig1C & Text line 88-91: is it TCGA or CGGA data?

We have corrected this typo.

5. Fig 1H: indicate kDa scale for NUAK2 and GAPDH western blots. What does P7, P14 mean? Please specify in legend

We thank the reviewer for pointing this out. We have included this information in the figures and legends.

6. Fig 2C&D: please indicate on the figure that the cells used were U251

We have updated the figure to reflect the cell line used.

7. Fig 2E: Does each dot in the graph represent the % calculated for 1 image? How many times this quantification was repeated? n=?

The data is representative of five images from 3 independent coverslips (15 total images). The percentage is from each individual image. This information has been added to the figure legend.

8. Line 858: sentence is not clear - you may need to delete 'and Nuak2 overexpression (N2OE)'

Thank you for this suggestion. We have edited the sentence to improve readability.

9. Fig2I: may check the scale bar and legend. Is it 50um?

Yes, we have verified that these images were taken at 40X magnification and the scale bar is 50uM.

10. Fig 2G & K: how the number of colonies are counted when they are touching to each other? Especially in U251WT, and U87N2OE, LN229N2OE. Should you also calculate the size (surface area) of the colonies? It is unclear what 'average of colonies per well' means - should it be number of colonies per well?

We have updated the labeling and included a summary of how the cells were analyzed in the methods.

11. Fig3B: we don't see the blue bars, so you may label the graphs with WT and NUA2CR on top of the graph, instead of using color code.

We thank the reviewer for this suggestion. We have incorporated it into the figure.

12. Fig3D: no scale bar. Can you delineate the tumor? As in 3C

We have now included a scale bar and outlined the tumor.

13. More histology markers / other markers may be included to show increase/decrease of cell proliferation

We have included an additional marker of proliferation PCNA (Fig 4E, Fig 5E) for *in vivo* histology studies and performed EdU incorporation assays for *in vitro* U251 experiments (Fig 2F). Both the PCNA and EdU studies corroborate our original finding with Ki67 that NUA2 OE enhances proliferation while NUA2 deletion inhibits proliferation. Please see response to a similar comment by reviewer 1. These results support a role for Nuak2 in glioma proliferation.

14. Fig4A: you may add 'NUAK2' next to 'GOF and LOF'

We have made this adjustment to figure in the updated submission.

15. Only Ki-67 staining demonstrate difference in cell proliferation. Would be interesting to know the size of the harvested tumors - tumor growth? Or to show more markers of cell proliferation.

Since we harvested tumor-bearing brains for histological analysis, we were limited to 2D measurements, which we believe do not accurately represent the full 3D tumor volume. Therefore, we cannot make definitive conclusions about total tumor size. However, as mentioned in reply to previous comments, we have now included additional data using PCNA as a second marker of cell proliferation (Fig. 4E, Fig. 5E), which shows results consistent with our Ki67 analysis.

16. Fig4E-G & EV1: how many sections/images were used to quantify?

For each of the stainings, three sections/images from three independent animals were quantified. We now include this information in the figure legend (Fig4).

17. Data in Figure 5 and text are not convincing - need to be fully re-arranged, better described and suggestion/conclusion changed.

We regret that Figure 5 (now Figure 6) and the accompanying text were not clear. We have significantly amended the text and re-structured the figure to improve clarity. As this reviewer suggested below (R2#37) we now have streamlined the analysis to focus on solely on ECM related associations.

18. Line 168-169: The analysis in Fig 5B suggests that ECM-related terms are significantly influenced by NUA2 expression in TCGA GBM patient samples, not in 'our samples'. Please rectify the sentence.

We have made this edit in the document.

19. Line 174 & Fig5C-D: please indicate in the text that 'CGGA' was also used. Correct the labeling of graph

5C: it should be TCGA instead of TGGA.

This panel has been removed from the updated submission.

20. Lines 178 to 185, Fig 5E-F & EV2B: Some labels are missing in EV2B, and what does Verhaak2010 mean? Based on GSEA analysis, authors suggest that 'NUAK2 loss leads to reduced mesenchymal properties' because 'proneural signatures were enriched'. They also mentioned that 'ECM regulation and EMT are interdependent processes' because they were both positively enriched... May need more data/support to suggest this.

We appreciate and agree with the comment that the connection between ECM and EMT in our analysis is premature. As suggested by this Reviewer we have focused on ECM-related associations and removed EMT (and MES) analysis from the manuscript.

21. Line 185-191: 'mesenchymal signature is related to more migratory properties' - please include a reference. The writing is bad and does not correspond to figures 5G-H.

We have removed this text from the document and significantly revised the results section for this figure.

22. Fig 5G and 5H show cell counts (cell proliferation) and not cell migration...

We apologize for the confusion: Fig 5G and Fig 5H (now in Fig 6) show measures of migrated cells. We have updated the labeling of the axis to provide more clarity that we are quantifying cell migration in these panels, not cell proliferation.

23. Fig 5H: cells are U87 and LN229, and not U251 (as mentioned in text).

We thank the reviewer for their careful reading of the manuscript. The text has been updated to reflect the data accurately.

24. The effect of loss or over expression of NUAK2 on cell proliferation has been already shown in Fig2.

We apologize for the confusion. Functional data shown in Figure 5 (now Figure 6) are transwell migration assays not proliferation assays as shown in Figure 2. We have updated the text to make this clearer.

25. Western blots in Fig 5I & J are not well described in the text, and do not support 'the role of NUAK2 in promoting EMT via ECM modulation'. N2OE in U87 and LN229 gave different results (E-CAD, N-CAD, SNAIL, SLUG, ...), but this is not discussed in the text.

We have removed references to EMT in the figures and text as stated in other question responses.

26. Title line159 "NUAK2 mediates mesenchymal transition through ECM regulation" may be changed.

We have revised the results section title to "NUAK2 regulates extracellular matrix (ECM) gene expression" in the updated manuscript.

Figure 6:

27. Line 207-209 & Fig EV3E: As mentioned in the text 'NUAK1 mRNA levels were significantly lower in both low- and high-grade gliomas than in normal brain tissues'. Could we say that NUAK1 expression may be correlated to NUAK2 expression? While NUAK2 increased in GBM & LGG tissues, NUAK1 decreased significantly. This could be discussed.

Please see response to reviewer 2#2 above.

28. Since NUA1 expression is relatively high and stable throughout brain development in both mice and humans, and in adult brain in humans, we cannot rule out that the significant decrease of NUA1 expression in GBM and LGG may have an impact on brain cancer.

We regret that the original submission did not adequately address NUA1. We have added new data as described above (comment#27) and amended the discussion to address NUAs in glioma.

29. Line 216-219: Please also indicate Fig EV4C.

We have edited the document to include this.

30. Line 222: do not mention 'wound healing' - please indicate 'scratch assay' instead of 'scratch wound healing'.

Thank you for the careful reading of the document. We have made this change in the text.

31. Figures 6C, EV4D: Author may change the brightness of the images as we do not see the cells. No scratch highlight in EV4D, cells U87. So how quantification was done (right graph)?

Thank you for bringing this issue to our attention. We have increased the contrast and brightened the images in the revised manuscript.

32. line 224-225, author mentioned that 'only U87 cells have distinct growth pattern, characterized by convergence and the formation of circular clusters'. However, U87 cells seemed to form colonies as the other cell types (Fig 2K, EV4B). How do we explain that in scratch assay, they do not behave the same as the other cell types?

Under standard adherent conditions (with serum), U87 cells usually grow as a monolayer but can form partial clusters, especially at higher densities. They form more defined spheroids in non-adherent or 3D cultures as we stated in the original text. The presence of the clusters in the scratch assay reflected an initial seeding density that was too high for the six-day assay, therefore we observed clusters. We optimized the seeding density and repeated the assay which is now included in the updated Fig EV4F. These updated results reveal that with 5uM HTH-02-006 the scratch region closes more slowly than vehicle through 36 hours. This is consistent with U87 cells having a higher proliferative rate than the other cell lines used in these studies.

33. Fig 6D: weird results - why U251 spheroids do not grow over the course of 6 days without HTH-02-006 (0uM)? as shown for other cell types in Fig EV5A. In

Previous studies showed that U251 cells have lower cell viability starting as early as day two and demonstrate earlier stagnation in growth than other glioma cell lines (<https://doi.org/10.1016/j.onano.2022.100116>). This may explain why the cells did not form and maintain spheres as readily as the other cell lines.

34. fig 6D & EV5B, do the graphs represent area at day 6? Please specify in figure and/or legend.

We thank the reviewers for pointing this out. The graphs represent total spheroid area and we have updated the figure to reflect this.

Other questions/discussion:

35. Besides Ki-67: Are there any other specific markers of proliferation for uncontrolled growth in cancers compared to tightly regulated proliferation during development?

Many of the markers expressed in proliferating cells are common to both cancer and developing cells. Those markers that are more specific to cancer cells are related to uncontrolled proliferation that are linked to oncogene activation or tumor suppressor loss. This becomes complicated when analyzing multiple cell lines that have different genetic background. Therefore, we used the standard markers for cancer proliferation employed across cancer biology: Ki-67 and PCNA. The new data obtained from PCNA IHC is included in Fig 4E and Fig5E which show PCNA staining recapitulates our results with Ki67—NUAK2CR decreases proliferation while NUA2OE enhances proliferation. We have also performed EdU incorporation analysis on U251-WT and U251CR cells as an additional marker of proliferation. This was also consistent with previously included data that showed decreased Ki67 in U251CR cells. This new data can be found in Fig 2F.

36. Transcriptomic analysis of GBM cells and TCGA patient data revealed that NUA2 likely exerts its effects by modulating the ECM. EV2C & D may be shifted in main figure 5

We appreciate that the reviewer drew the conclusion that NUA2 likely exerts its effects by modulating ECM. Per the Reviewers suggestion we have restructured Figure 5 (now Figure 6) to include the ECM qPCR data, updated Fig EV2 with YAP/TAZ analysis, and revised the main text to reflect the focus on ECM rather than EMT as suggest in comment #37 below.

37. As mentioned before, the data demonstrating the role of NUA2 in mediating EMT is weak, and it is not further discussed in the 'Discussion' part. **The authors may only focus on ECM increase.**

We thank the reviewer for this insightful suggestion. We have restructured both the text and the original Figure 5 (now figure 6) to exclude EMT (and MES) associated data and now only focus on ECM and the YAP/TAZ pathway.

38. Line 287: authors do not investigate the effect of 'genetic' modulation of NUA2 but the change of its 'expression'. This should be reformulated.

Thank you for bringing this to our attention. We have edited the sentence.

39. Line 289: please include 'that': We observed 'that' the growth...
We have corrected this typo.

40. Line 293-303: although authors did not find correlation between expression levels of NUA1 and glioma types and survival rates in available datasets, they cannot conclude that NUA1 does not play a role. The significant decrease of NUA1 expression in tumors (GBM and LGG) compared to normal tissues must be discussed. Authors have not shown any in vitro data that confirm this in-silico finding. They did not quantify NUA1 level in their cell lines, or when NUA2 is overexpressed. NUA1 quantification ought to be added in the revised manuscript if NUA1 is included in FigEV3

Please see the response to the R2#2 above and addition to discussion.

41. Does NUA2 inhibition by HTH-02-006 decrease NUA2 expression? Increase NUA1? The authors should show by Q-PCR and western.

Please see response to R2#2 above.

15th Jul 2025

Dear Dr. Glasgow,

Thank you for submitting your revised study. Unfortunately, referee #2 was unavailable to evaluate your revised manuscript; however, referee #1 reviewed your responses to both referees. As you will see below, this referee is satisfied with the revisions. I will therefore be able to accept your manuscript once the following editorial issues have been addressed.

1/ Manuscript text:

- Please indicate in track changes mode any new modification in the text.
- The manuscript sections should be in the following order: Title page - Abstract & Keywords - Introduction - Results - Discussion - Methods - Data Availability - Acknowledgments - Disclosure Statement & Competing Interests - References - Figure Legends - Expanded View Figure Legends.

- Please provide up to 5 keywords.

- Methods:

- o Animals: please provide the housing and husbandry conditions.

- o Cell lines: please indicate whether the cells were authenticated.

- o Western blots: if membranes were stripped, please describe in the methods.

- o Statistics: please provide a statement on inclusion/exclusion criteria, sample size, blinding and randomization.

- o BioRender should be acknowledged at the end of the Methods section in the following way:

Graphics:

(some of the... OR Figure #... OR synopsis) Graphics were created with BioRender.com (and removed from the acknowledgements).

- Data Availability: please provide the specific URLs for S-BIAD2082 and GSE285513 datasets.

- Competing interests should be renamed "Disclosure and Competing Interests Statement" and should contain only one section.

- Ethics statement should be included in the Methods.

- Author contributions: CRediT has replaced the traditional author contributions section because it offers a systematic machine readable author contributions format that allows for more effective research assessment. Please remove the Authors Contributions from the manuscript and use the free text boxes beneath each contributing author's name in our system to add specific details on the author's contribution. More information is available in our guide to authors.

- Funding should be merged with Acknowledgements, and the heading "Funding" should be removed.

- Check the references labeled [PREPRINT], as they seem to be published manuscripts.

- Our journal encourages inclusion of *data citations in the reference list* to directly cite datasets that were re-used and obtained from public databases. Data citations in the article text are distinct from normal bibliographical citations and should directly link to the database records from which the data can be accessed. In the main text, data citations are formatted as follows: "Data ref: Smith et al, 2001" or "Data ref: NCBI Sequence Read Archive PRJNA342805, 2017". In the Reference list, data citations must be labeled with "[DATASET]". A data reference must provide the database name, accession number/identifiers and a resolvable link to the landing page from which the data can be accessed at the end of the reference. Further instructions are available at .

2/ Figures:

- Please provide smaller size figures (the current merged pdf is 990MB).

- Please make sure that all figures and figure panels are referenced in the manuscript text (a callout is currently missing for Fig. 2C).

- The nomenclature in EV figure should be Figure EV1, instead of Expanded View Figure 1, etc.

- Datasets: the source file names and titles in the submission system need to be corrected to Dataset EV1-EV3.

- The Appendix file should be in PDF format.

- Please address the requests from our data editors:

1. Please note that the figure 7F, G is mislabeled as figure 7E, F in the manuscript. This needs to be rectified.

2. Please note that the legend for figure 7E is missing in the manuscript. This needs to be rectified.

3. Please define the annotated p values ****/****/**/ as well as provide the exact p-values for the same in the legend of figure 1B, EV3 E as appropriate.

4. Please note that the exact p values are not provided in the legends of figures 1C, D, E, F; 2D, E, F, H; 5F, G, H, I; 6G, H; 7B, D, G; EV1 B, EV3 E.

5. Please indicate the statistical test used for data analysis in the legend of figure EV2 C

6. Please note that in figures 4C there is a mismatch between the annotated p values in the figure legend and the annotated p values in the figure file that should be corrected.

7. Please note that the box plots need to be defined in terms of minima, maxima, centre, bounds of box and whiskers, and percentile in the legends of figures 1B, 4B, EV3 D.

8. Please note that information related to n is missing in the legends of figures 1A, 2F, G; 3C, D; EV3 A, EV3 F, G; EV3 F, G; EV4 D, F, G.

9. Please note that the error bars are not defined in the legends of figures 1I, 2E, F; 3H, J; EV4 E.

3/ Thank you for providing Source Data. Please also provide a file for Figure 7 Source Data.

4/ Checklist:

- Experimental study design and statistics: please fill in the subsection "inclusion/exclusion criteria".
- Ethics: please fill in the subsection "Studies involving experimental animals"

5/ Thank you for providing The paper explained. Please include it in the manuscript file.

6/ Thank you for providing a nice visual abstract. I have cropped a small portion of this image to serve as thumbnail for the table of content on our webpage (attached). Please let me know if you agree, or provide an alternative image (115 x 70 pixels).

7/ As part of the EMBO Publications transparent editorial process initiative (see our Editorial at <http://embomolmed.embopress.org/content/2/9/329>), EMBO Molecular Medicine will publish online a Review Process File (RPF) to accompany accepted manuscripts.

This file will be published in conjunction with your paper and will include the anonymous referee reports, your point-by-point response and all pertinent correspondence relating to the manuscript. Let us know whether you agree with the publication of the RPF and as here, if you want to remove or not any figures from it prior to publication.

I look forward to receiving your revised manuscript.

With kind regards,

Lise Roth

***** Reviewer's comments *****

Referee #1 (Remarks for Author):

The authors have responded to most of my questions and concerns, and have improved the manuscript accordingly.

The authors addressed the remaining editorial issues.

21st Jul 2025

Dear Stacey,

Thank you for sending your revised files. I am pleased to inform you that your manuscript is accepted for publication and is now being sent to our publisher to be included in the next available issue of EMBO Molecular Medicine!

Please note that I found a discrepancy between Figure 3E Source data (GAPDH) and Figure 3E in the manuscript (tubulin). Please clarify and send us the corrected figure or source data via email, and we'll upload it in the system.

With kind regards,

Lise
